# Multitissue H3K27ac profiling of GTEx samples links epigenomic variation to disease

Lei Hou [1,2,4], Xushen Xiong [1,2,3,4], Yongjin Park[1,2], Carles Boix [1,2], Benjamin James [1,2], Na Sun[1,2], Liang He[1,2], Aman Patel[1,2], Zhizhuo Zhang[1,2], Benoit Molinie[2], Nicholas Van Wittenberghe[2], Scott Steelman[2], Chad Nusbaum[2], François Aguet [2], Kristin G. Ardlie[2] & Manolis Kellis [1,2] ✉

Genetic variants associated with complex traits are primarily noncoding, and their effects on gene-regulatory activity remain largely uncharacterized. To address this, we profile epigenomic variation of histone mark H3K27ac across 387 brain, heart, muscle and lung samples from Genotype-Tissue Expression (GTEx). We annotate 282 k active regulatory elements (AREs) with tissue-specific activity patterns. We identify 2,436 sex-biased AREs and 5,397 genetically influenced AREs associated with 130 k genetic variants (haQTLs) across tissues. We integrate genetic and epigenomic variation to provide mechanistic insights for disease-associated loci from 55 genome-wide association studies (GWAS), by revealing candidate tissues of action, driver SNPs and impacted AREs. Lastly, we build ARE–gene linking scores based on genetics (gLink scores) and demonstrate their unique ability to prioritize SNP–ARE–gene circuits. Overall, our epigenomic datasets, computational integration and mechanistic predictions provide valuable resources and important insights for understanding the molecular basis of human diseases/traits such as schizophrenia.

Genome-wide association studies (GWAS) have identified more than 200 k variants spanning over 3,000 human complex traits and diseases[1]. However, over 93% of lead single-nucleotide polymorphisms (SNPs) in GWAS loci lie in noncoding regions and are coinherited with many other variants in regions of linkage disequilibrium (LD). Thus, it is challenging to recognize the context and gene-regulatory circuits through which these disease-associated genetic variants act[2–5], such as (1) the specific contexts including tissue/cell types where the variants function[6–9], (2) the causal genetic variants that drive the alterations[10–14] and (3) the impact of genetic variants on regulatory elements and target genes at varying genomic distances[15–19].

Tissue-specific functional genomics data characterize molecular effects that may mediate the impact of genetic variants on human diseases/traits, enabling us to address the above challenges. The Genotype-Tissue Expression (GTEx) project[20–22] and other efforts[23,24] identified *cis*-acting genetic variants affecting gene expression and splicing, that is, expression of quantitative trait loci (eQTLs) and splicing QTL (sQTLs), in multiple human tissues. The enhancing GTEx (eGTEx) project[16] has expanded the GTEx map with protein[25], telomere-length[26], DNA methylation[27,28] and the epitranscriptomic mark m6A[29]. These resources connect genetic and phenotypic variations via rich molecular phenotypes. The rationale for eGTEx is twofold—it enables understanding the impact of genetic variation on molecular traits and the

[1]Computer Science and Artificial Intelligence Lab, Massachusetts Institute of Technology, Cambridge, MA, USA. [2]The Broad Institute of Harvard and MIT, Cambridge, MA, USA. [3]Present address: Liangzhu Laboratory, Zhejiang University, Hangzhou, China. [4]These authors contributed equally: Lei Hou, Xushen Xiong. ✉e-mail: manoli@mit.edu

diverse selective forces acting on the human genome, and it elucidates the functions of GWAS variants, providing more stepping stones when they are also captured by eQTLs or indicating context of action when their impacts are condition-/cell-type-specific and missed by eQTL in steady-state postmortem tissues.

Notably, the current eGTEx resource lacks information on promoter/enhancer activity variation. Multiple studies have shown that tissue-specific enhancers are also strongly enriched for genetic variants associated with tissue-relevant complex traits and diseases[3,9,18,19,30–32], and interindividual enhancer variation can help interpret disease-associated loci[33–38]. The epigenomic mark most directly associated with enhancer activity is H3K27ac, found primarily at both active enhancers and promoters.

Here we expand the eGTEx collection with H3K27ac chromatin immunoprecipitation followed by high-throughput sequencing (ChIP–seq) of 387 samples across brain, heart, muscle and lung from 256 GTEx participants and characterize promoter/enhancer variation associated with tissue/cell types, sex, genetic variants, transcription variation and clinical traits (Fig. 1a). We detect 282 k active regulatory elements (AREs) and classify them into 14 groups based on coactivity patterns across 240 reference epigenomes[18,19]. We also identify 2,436 sex-biased AREs and 130 k distinct *cis*-acting histone acetylation quantitative trait loci (haQTLs) targeting 5,397 distinct genetically influenced AREs (gAREs) across tissues. Leveraging these findings, we investigate the genetics of complex diseases and traits with 55 GWAS. We predict candidate tissues/cell types of action for diseases/traits and prioritize 614 distinct GWAS–haQTL-colocalized gAREs. Finally, we calculate ARE–gene linking scores based on genetic evidence (gLink scores) and prioritize 228 target genes for 161 GWAS–haQTL-colocalized gAREs across four tissues.

## Results

### H3K27ac ChIP–seq profiles in brain, heart, muscle and lung
We selected the following four tissues for epigenomic profiling: brain (prefrontal cortex), heart (left ventricle), muscle (skeletal) and lung. These tissues show strong relevance for multiple diseases, including psychiatric, cardiac, metabolic and respiratory traits, and also capture developmental diversity as they span all three germ layers of ectoderm (brain), mesoderm (heart and muscle) and endoderm (lung). We performed H3K27ac ChIP–seq, preprocessed the data with the ENCODE pipeline[39], selected higher-quality samples and generated a peak set in each tissue using the highest-quality samples (Methods).

We reported a total of 387 genome-wide H3K27ac profiles, including 113 brain, 100 heart, 108 muscle and 66 lung samples selected for analysis after QC (Fig. 1a and Supplementary Table 1). They outnumbered the H3K27ac experiments previously undertaken by the Roadmap Epigenomics[31] (*n* = 124), ENCODE[18] (*n* = 98) and Genomics of Gene Regulation[40] (*n* = 18) consortia (total: *n* = 240, combined in the EpiMap resource[19]; Fig. 1b, inset), providing an important resource to the field of epigenomics and gene regulation.

To understand the relationship between our sample profiles and those of other consortia, we projected all 627 H3K27ac profiles into a lower-dimensional embedding space[41] (Fig. 1b and Methods). We found that our H3K27ac profiles from the same tissues are clustered closely to tissue-matched samples from Roadmap, ENCODE and GGR (Fig. 1b and Supplementary Note). We also confirmed that tissue-matched pairs of the H3K27ac profiles from this study and those from previous studies show high correlations (Extended Data Fig. 1 and Methods).

### ARE activity across tissues, cell types and sex
We identified AREs from brain (190 k), heart (132 k), muscle (143 k) and lung (107 k) samples based on H3K27ac ChIP–seq peaks (Methods). We then aggregated overlapping AREs across tissues as a reference set of 282 k distinct AREs, of which 14% (39 k) are fully shared across all four tissues, 24% (68 k) are partially shared in two or three tissues and 62%

are tissue-specific (brain, 86 k; heart, 31 k; muscle, 37 k and lung, 21 k; Fig. 2a and Extended Data Fig. 2a). Fully shared AREs were enriched for promoter annotations (54–80% in promoters versus 26–44% in background across tissues), while tissue-specific AREs were enriched for enhancers (79–93% enhancers versus 56–74% expected).

To distinguish brain/heart/muscle/lung-specific AREs, broadly active AREs or AREs that are primarily active in other tissues, we leveraged the diversity of tissues and cell types of H3K27ac profiles in Roadmap/ENCODE/GGR and clustered our 282 k AREs based on their H3K27ac activity patterns across 240 reference epigenomes, resulting in 127 modules, which we further clustered into 1,413 submodules and also aggregated into 14 groups (G1–G14; Fig. 2b, Extended Data Fig. 2b and Methods). We found that 19 modules (74 k AREs, 26%) were broadly or multitissue active. Only 25 of the 127 modules (67 k, 24%) were active primarily in our four tissues, while 82 modules of these AREs (112 k, 40%) were primarily active in other tissues from the reference samples, such as those in blood/immune (G3). Lastly, one module of AREs (G14, 29 k, 10%) did not show activity in the previous reference epigenomes and was newly detected here.

We next annotated these 127 modules from ARE groups based on enrichments of chromatin states, genomic annotations, transcription factors (TFs) binding sites and proximal gene function annotations (Fig. 2b and Extended Data Figs. 2c and 3a; Methods). These annotations are consistent with their primary tissue of activity. For example, broadly active AREs (G1) were enriched for transcription start site (TSS)-proximal regions, promoter/enhancer chromatin states and housekeeping genes ($P < 2.2 \times 10^{-16}$); brain/neuron AREs (G12) were enriched for brain enhancers/promoters, *trans*-synaptic signaling pathways and neuronal-function TFs binding sites, such as NEUROG2 (ref. 42). Blood/immune AREs (G3) were enriched for immune functions and immune-related TFs binding sites, including IRF5 and IRF8. Newly detected AREs (G14), showing weak activities in both datasets (Extended Data Fig. 2b), required more samples to detect than other AREs (Extended Data Fig. 3b). G14 were enriched for brain-specific AREs, intergenic regions, synaptic-related pathways and neuronal-function TFs binding sites such as NFATC1 (ref. 43), indicating the brain as the primary detection tissue.

We then identified 192 (brains), 1,211 (hearts), 1,214 (muscles) and 23 (lung) sex-biased AREs after controlling for unwanted variations, including tissue-archetype fractions (estimated by our deconvolution approach; Extended Data Fig. 4a,b) and phenotypic variables (Extended Data Fig. 5a, Supplementary Table 2 and Supplementary Note). These sex-biased AREs, enriched near previously identified sex-biased genes (Extended Data Fig. 5b), show consistent directionality with those sex-biased genes in the matched tissues from GTEx[44] (one-sided proportion test, $P = 2.8 \times 10^{-10}$ for female-biased and $1.9 \times 10^{-5}$ for male-biased AREs; heart as an example in Extended Data Fig. 5c), validating our results across the modalities of transcriptomes and epigenome. Gene Ontology (GO) annotation shows that genes near the sex-biased AREs were enriched for housekeeping processes such as endoplasmic reticulum-unfolded protein response and tissue-specific processes such as sensory perception of smell for the brain (Extended Data Fig. 5b), suggesting that sex can affect both types of programs across tissues.

### Genetic drivers of ARE activity in each tissue
We next discover *cis*-acting haQTLs associated with variations of enhancer/promoter activity (genetically influenced AREs and gAREs) across individuals in each tissue. Following the GTEx eQTL calling pipeline, we removed top peer factors[45] and other covariates and mapped haQTL using FastQTL[46] within 100 kb of AREs (Methods and Extended Data Fig. 6a–d). We found a total of 83 k brain haQTLs (targeting 2,162 gAREs), 44 k heart haQTLs (1,311 gAREs), 62 k muscle haQTLs (1,816 gAREs) and 8 k lung haQTLs (537 gAREs; Fig. 3a and Supplementary Table 3).

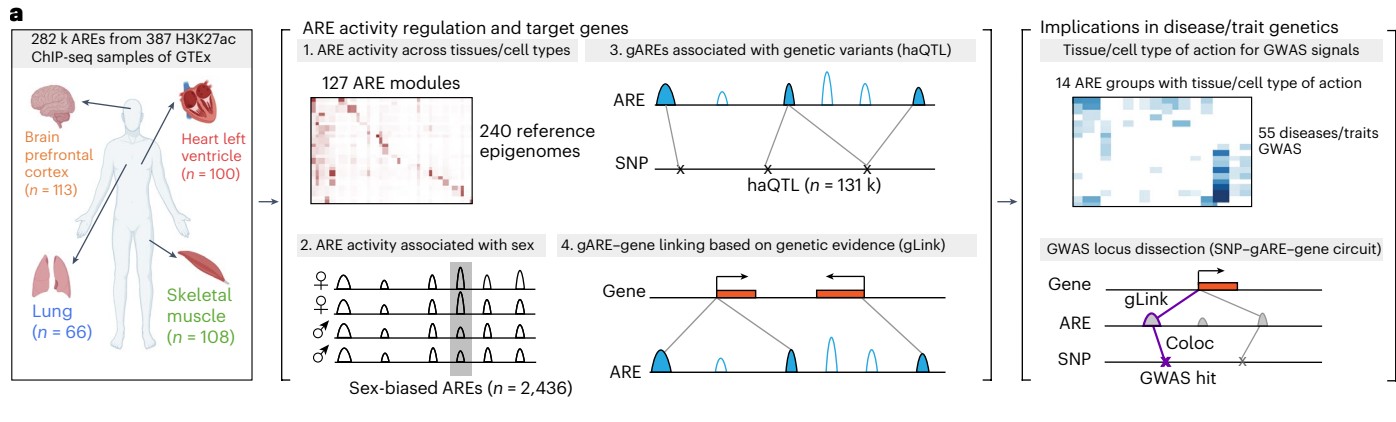

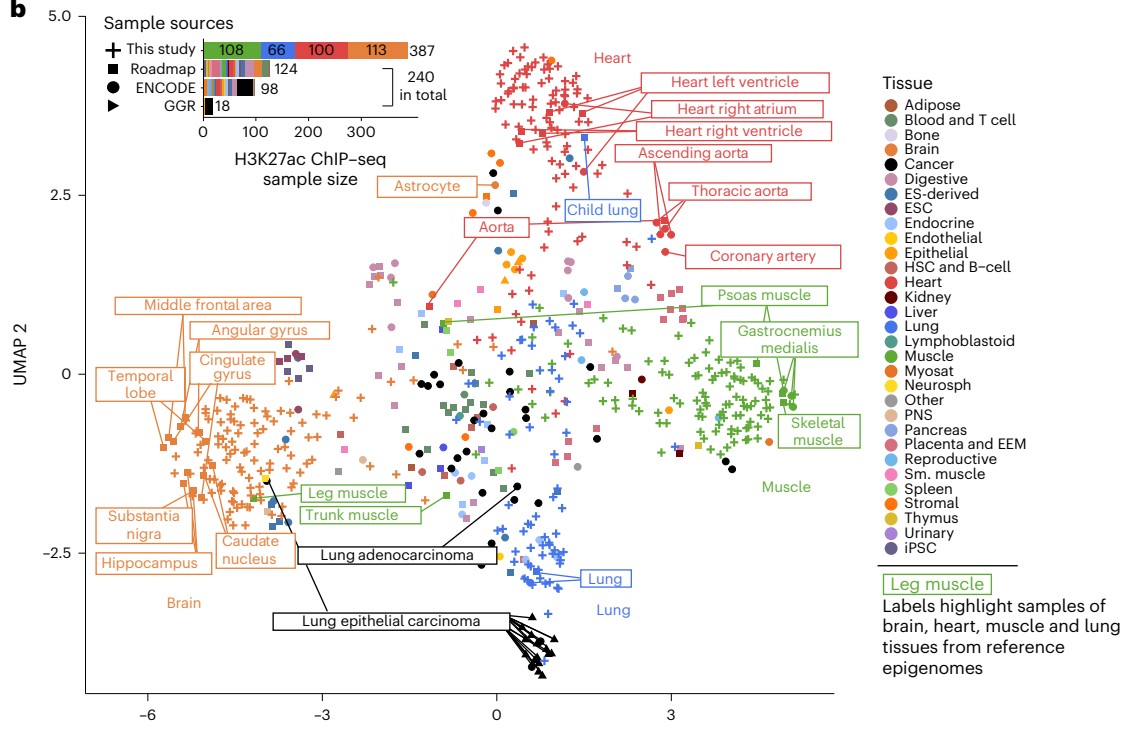

**Fig. 1 | Study overview and lower-dimensional projection of H3K27ac profiles. a**, Study overview. Left, H3K27ac samples in this study; middle, regulation of AREs and target genes; right, functional implications in human disease/traits genetics. Organ sketch plots were created with BioRender.com. **b**, Lower-dimensional projection of 387 samples from this study and 240 samples from the reference epigenomes. Different colors represent different tissues or cell types, and different shapes represent different studies; boxed texts label the samples of brain, heart, muscle and lung from reference epigenomes; the top left inset shows the sample sizes of the H3K27ac experiments from various sample sources.

To understand the tissue specificity of genetic effects on epigenomic regulation, we next studied the sharing of haQTLs and gAREs between tissues. We distinguished three types of gAREs, which are as follows: (1) haQTL-shared gARE, where both gARE and its haQTL are detected in another tissue (examples 1 and 2 in Fig. 3b); (2) haQTL-specific gARE, where the ARE is present in another tissue but the haQTL effect is tissue-specific (example 3 in Fig. 3b) and (3) ARE-specific gARE, where the ARE is tissue-specific and so is the haQTL (examples 4 and 5 in Fig. 3b).

We assessed pairwise haQTL tissue sharing for the shared gAREs by evaluating the directionality consistency between 'discovery' and 'replication' tissues (Methods; Fig. 3c and Extended Data Fig. 7a), a strategy that mitigates limitations due to QTL detection power[29]. We evaluated this consistency at varying levels of haQTL significance in the replication tissue, which are as follows: strongly replicated ($P < 10^{-5}$; examples 1 and 2 in Fig. 3b), medium-replicated ($10^{-5} < P < 10^{-3}$),

weakly replicated ($10^{-3} < P < 0.1$) and no-effect ($P > 0.1$; example 3 in Fig. 3b). Strongly and medium-replicated haQTLs showed over 98% directionality consistency. Even weakly replicated haQTLs showed 82% consistency, suggesting an underestimation of haQTL effect sharing if applying haQTL discovery threshold for both tissues (Fig. 3c). Furthermore, weakly replicated haQTLs show higher effect size similarity with the discovery tissue than those from the no-effect bin (Extended Data Fig. 7b,c; Methods), confirming the tissue sharing of subthreshold haQTLs in the replication tissue.

We then calculated the proportion of haQTL tissue sharing for each pair of tissues based on directionality consistency and found 18% to 49% haQTL tissue sharing (Fig. 3d; Methods). The haQTL tissue sharing between muscle and heart is higher compared to other tissue pairs (Fig. 3d), consistent with shared germ layer origin. For individual gAREs, we defined haQTL-shared gAREs based on the nominal $P$-value threshold ($P \le 0.02$) in the replication tissues

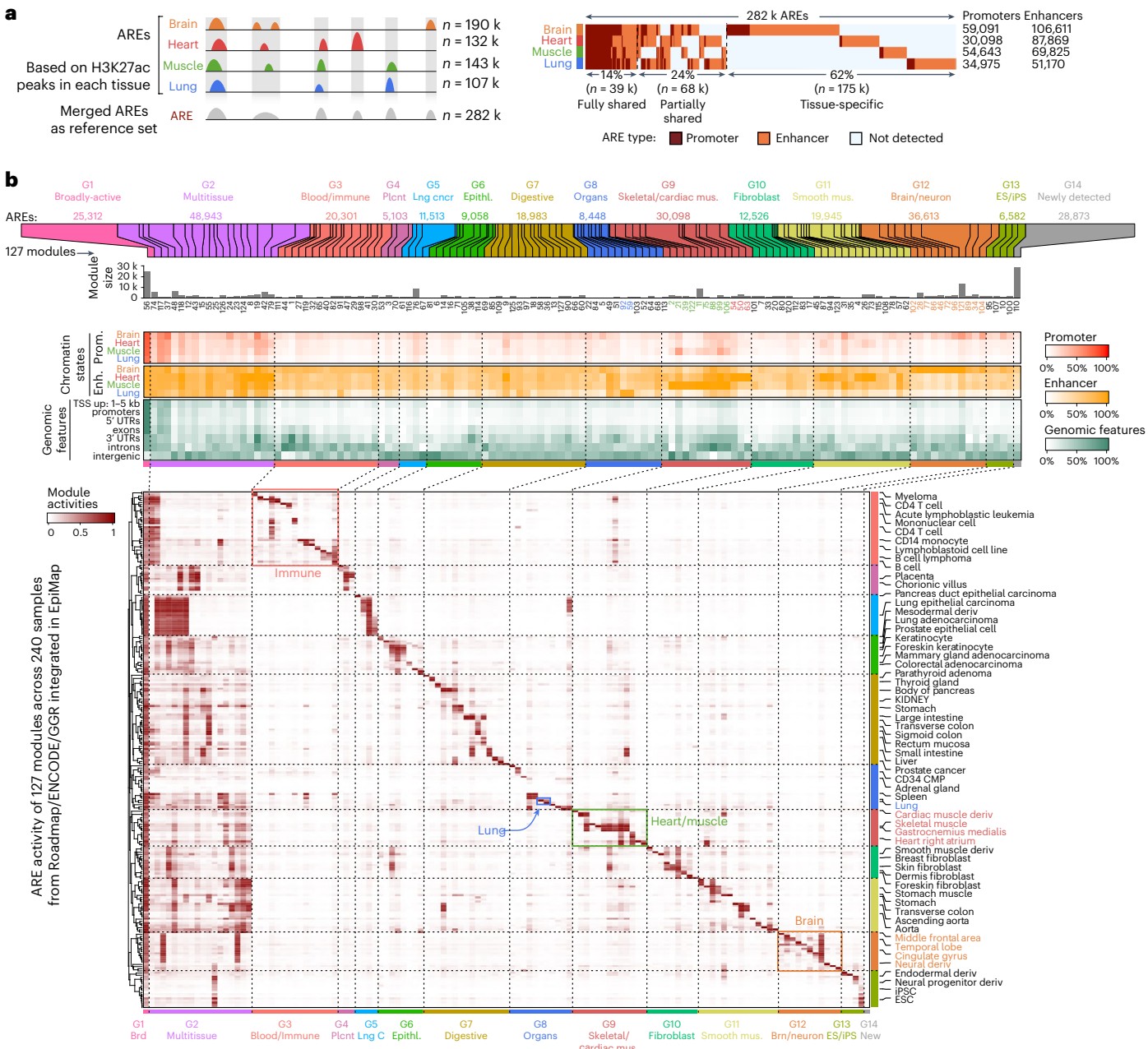

**Fig. 2 | ARE activity across tissues and cell types. a**, ARE detection from brain, heart, muscle and lung in this study. Left, schematic diagram showing the AREs detected from each tissue and how they are merged as a reference set, with the number of AREs shown on the right; right, ARE-sharing and specificity across the four tissues, with promoter and enhancer AREs in different colors. **b**, 282 k AREs identified in this study forming 14 groups of 127 modules based on coactivity across 240 reference epigenomes. Top, 127 ARE modules and module size; middle, proportion of AREs annotated as promoter/enhancer-related chromatin states and different genomic features for each module; bottom, average ARE activity of each module (by column) across 240 reference epigenomes (by row). Representative sample names labeled for each sample cluster on the right; 14 ARE groups labeled at the bottom; red, green, orange and blue boxes showing blood/immune (G3), skeletal/cardiac muscle (G9), brain/neuron (G12) modules and lung-specific modules.

(Extended Data Fig. 7d and Supplementary Table 4; Methods), and then looked at the proportions of all three gARE types in each tissue. The haQTL-specific gARE is the most common type except in the brain, while ARE-specific gAREs comprise a much larger proportion in the brain than in other tissues (Fig. 3e), consistent with a larger proportion of tissue-specific AREs in the brain (Fig. 2a).

We then tested the enrichment of ARE groups for each of the three gARE types (Fig. 3f). The haQTL-shared and haQTL-specific gAREs were primarily enriched in broadly active AREs (G1; examples 1 and 3).

haQTL-specific gAREs were particularly enriched in multitissue AREs (G2) for brain and muscle. ARE-specific gAREs were enriched in newly detected AREs (G14; except lung) and tissue-specific ARE groups, such as G12 for brain (example 5), G9 for heart and muscle, and G7 and G8 AREs for lungs (example 4). These results showed that haQTL-specific and ARE-specific gAREs provide tissue-specific genetic regulation with distinct mechanisms. Separately, we also revealed significantly positive associations of tissue specificity between haQTL and eQTL in all tissues except for lungs (Extended Data Fig. 7e and Supplementary Note).

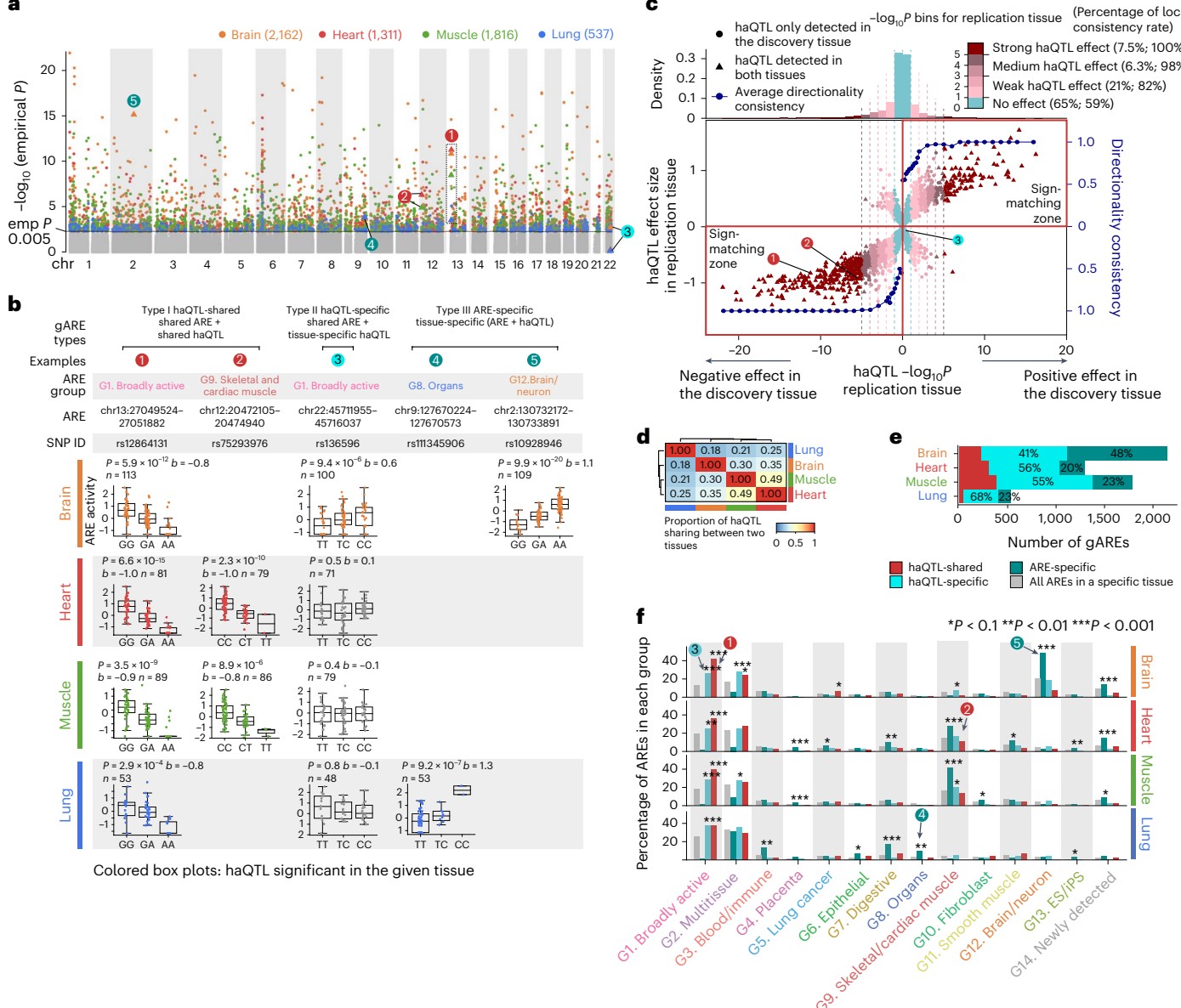

**Fig. 3 | Genetic drivers of ARE activity and their tissue specificity. a**, Overlaid Manhattan plot of lead haQTLs from brain, heart, muscle and lung; *x* axis shows the genomic position and *y* axis shows the -log₁₀(empirical *P* value); empirical *P* value is haQTL *P* value adjusted for multiple SNPs for each loci as described in Methods; examples 1–5 in **b** are marked with numbers in colored circles. **b**, Examples for type I (1–2), type II (3) and type III (4–5), along with the information of ARE group, ARE, lead haQTL, gARE activity and SNP alleles for each sample across tissues and number of samples (*n*); boxes = 25th to 75th percentile (that is, IQR); line = median; whiskers = 1.5× IQR; haQTLs *P* values shown in **b** and **c** are all nominal *P* values based on linear regression (two-sided) implemented in FastQTL as described in Methods. **c**, Directionality consistency-based haQTL tissue-sharing analysis. Lower, between-tissue directionality consistency (*y* axis on the right) of the lead haQTL effect size (*y* axis on the left) increases as the *P* value significance of the haQTLs in the replication tissue

(*x* axis) increases, separated by positive effect (right half-plane) and negative effect (left half-plane) of the matched haQTL–ARE pair in the discovery tissue; the examples 1–3 (shared ARE between tissues) from **a** are labeled. Upper, density of lead haQTLs in each of the nominal *P* value bins; the proportion and the corresponding consistency rates of each group are shown in parentheses in the figure legend. **d**, The estimated proportion of haQTL tissue sharing for each tissue pair. **e**, The number of types I, II and III gAREs in each tissue. Colors for different types of gAREs are shown in the legend below the panel. **f**, Enrichment of different types of gAREs in 14 ARE groups (defined in Fig. 2b). *x* axis, each ARE group; *y* axis, percentage of different types of AREs in each ARE group; asterisk shows the significance level of enrichment (two-sided Fisher's exact test, only for odds ratio > 1) compared to all AREs in each tissue (exact *P* value shown in Supplementary Table 4); colors for different types of gAREs are shown in the legend above the panel. IQR, interquartile range.

## ARE regulation reveals mechanisms for disease/trait genetics

Next, we leveraged ARE regulation associated with tissues/cell types and genetics to gain insight into disease genetics. We first studied the enrichment of common genetic variants associated with diverse traits from 55 GWAS (Supplementary Table 5) in 14 ARE groups with distinct primary tissues of activity using LD score regression analysis[9].

Cardio-metabolic traits (Fig. 4a, top group), including coronary artery disease and hypertension, were enriched in broadly active (G1) and multitissue (G2) AREs detected in multiple tissues in our study, and AREs from other groups mainly detected in our heart samples[47]. Allergy/immune-related traits (Fig. 4a, second group) were enriched exclusively in broadly active (G1), multitissue (G2) or blood/immune (G3)

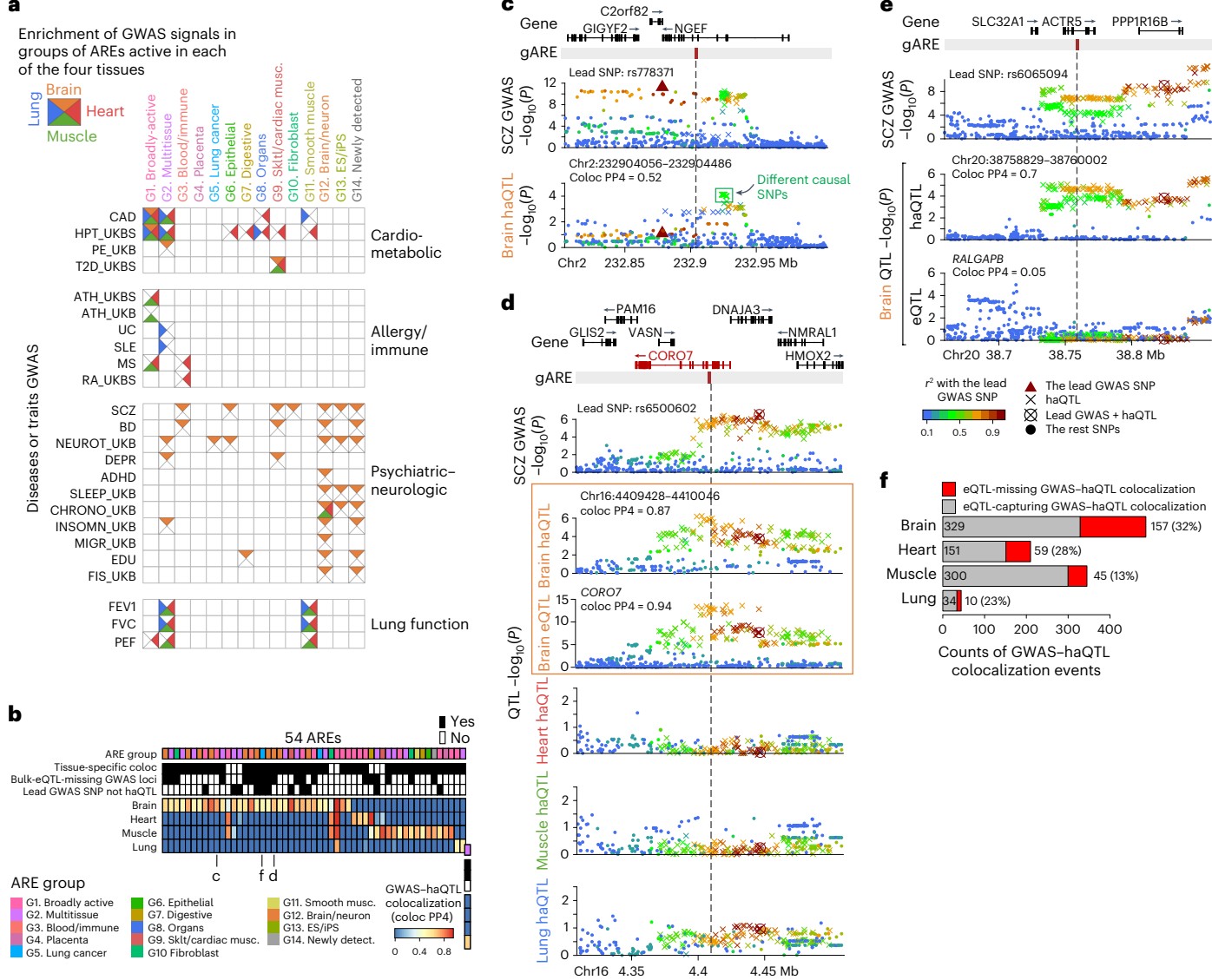

**Fig. 4 | ARE regulation reveals mechanisms for disease/trait genetics.**
**a**, Enrichment of GWAS signals for diseases and traits in 14 ARE groups, partitioned by the tissue presence for each ARE. Rows represent GWAS studies (full names in Supplementary Table 5) from four categories (right label); columns represent different ARE groups; different triangles in each square denote whether the ARE in a group from a tissue is substantially enriched for the corresponding trait. **b**, Overview of 54 GWAS–haQTL-colocalized gARE loci for schizophrenia. Top annotations include the ARE group that each gARE belongs to, whether it is tissue-specific colocalization (an example in **d**), whether the GWAS signal is missed by eQTL from top GWAS–haQTL colocalization tissue (an example in **e**), and whether there is a different causal SNP from the lead SNP suggested by haQTL (an example in **b**). **c**, An example of different GWAS causal SNPs suggested by colocalization. $x$ axis represents genomic location; $y$ axis for each panel from top to bottom, schizophrenia GWAS signal ($-\log_{10}(P)$) and haQTL signals ($-\log_{10}(P)$) for SNPs in brain; $P$ values in Fig. 4 are all nominal $P$ (two-sided) for either GWAS (chi-squared) or QTL (linear regression); the colors of data points represent the correlation ($r^2$) with lead GWAS SNP and symbols denote different types of SNPs as shown in the legend underneath **e**; the inset indicates the alternative causal SNPs. **d**, An example of tissue-specific regulation based on colocalization. Panels similar as **c**; panels for both haQTL and eQTL signals in brain for tissue-specific colocalization are boxed in orange. **e**, An example of bulk-eQTL-missing GWAS–haQTL-colocalized gARE loci. Bottom, the strongest GWAS–eQTL colocalization detected for *RALGAPB*; other panels similar as **c**. **f**, The counts of bulk-eQTL-missing and -capturing GWAS–haQTL-colocalized gAREs for each tissue.

AREs, consistent with the immune-related functions for these ARE groups (Extended Data Fig. 2c). Most psychiatric–neurologic-associated traits (Fig. 4a, third group) were enriched in brain/neuron (G12) and newly detected (G14) AREs active in our brain samples, consistent with brain-related functions for these groups (Extended Data Fig. 2c). Schizophrenia (SCZ) and bipolar disorder (BD) also showed enrichment for blood/immune (G3) AREs, consistent with increasingly recognized immune roles in the diseases[48,49]. Lastly, lung-function-associated traits (Fig. 4a, bottom group) were enriched in broadly active (G1), multitissue (G2) and smooth muscle (G11) AREs only detected in our

nonbrain samples, highlighting the importance of smooth muscle in respiratory traits[50,51].

We then applied coloc, a Bayesian framework of colocalization analysis[52], to estimate the sharing of causal genetic variants between GWAS and haQTL signals within our gARE loci and to recognize potentially impacted AREs (Supplementary Note). We also used the same analysis for GTEx eQTLs within our gARE loci, to reveal the disease loci that show epigenomic and/or transcriptional effects (Extended Data Fig. 8a). We found 1,070 GWAS–haQTL colocalized events (614 unique AREs, coloc PP4 ≥ 0.5) across our 4 tissues and 46 traits, 403 of which

(38%) are also confirmed by Mendelian randomization (MR, adjust $P < 0.2$; Supplementary Table 6 and Supplementary Note).

As a proof of principle, we focused specifically on the 54 GWAS–haQTL-colocalized gAREs for schizophrenia (Fig. 4b), finding that most of them show tissue-specific colocalization (46, 85%) primarily in brain tissue (30, 56%), consistent with the expected tissue of action. For 11 gARE loci, haQTL pinpoints causal SNPs different from the lead GWAS SNPs. For example, the GWAS lead SNP rs77831 was not detected as a significant haQTL in brain tissue (Fig. 4c), while the SNPs identified as significant haQTLs (green box) may impact gene-regulatory alterations in the brain related to schizophrenia etiology.

For 34 of 54 gARE loci (63%), haQTLs–GWAS-colocalized variants also impacted gene expression in the matched tissue. For example, a brain/neuron (G12) ARE, haQTL detected only in the brain (type II 'haQTL-specific' gARE), shows GWAS–haQTL colocalization in the brain, together with GWAS–eQTL colocalization only in the brain (orange box, Fig. 4d). These results together reveal a brain-specific SNP–ARE–gene regulatory circuit involved in schizophrenia.

For the other 20 of 54 gARE loci (37%), we did not observe any eQTL colocalization effect (coloc PP4 < 0.1, 'bulk-eQTL-missing GWAS loci' in Fig. 4b). For example, the lead schizophrenia GWAS SNP rs6065094, also an haQTL of a brain/neuron (G12) ARE, does not show eQTL effect on any genes nearby (Fig. 4e). Using SuSIE followed by colocalization[53], we confirmed that the majority of these eQTL-missing loci are not due to multiple independent eQTLs confounding the analysis (Extended Data Fig. 8b and Supplementary Note). We then carried out colocalization analysis using cell-type-level eQTLs from a brain single-nucleus RNA-seq dataset[54], to check the possibility that the bulk-eQTL-missing may arise from lower power to capture cell-type-specific effect. In total, 10 out of 13 brain bulk-eQTL-missing loci could be explained by cell-type-level eQTLs (Extended Data Fig. 8c), implying that bulk haQTLs may have better power than bulk eQTLs to reveal cell-type-level function for these disease genetic variants. For all GWAS–haQTL colocalization events identified across traits, we found that 13–32% of them were missed by bulk eQTL across four tissues (Fig. 4f).

## gLink scores prioritize gARE–eGene pairs based on genetic variations

We next identified potential target genes for gAREs to uncover gene-regulatory circuits underlying disease genetics. We first linked a gARE to a gene if it is proximal to the gene's fine-mapped eQTLs (FMeQTLs), which are more likely to pinpoint causal SNPs than eQTLs after incorporating the LD structure. As expected, gAREs tend to be proximal to eQTLs (2 kb window) compared to various backgrounds, and this enrichment is even higher for FMeQTLs (Fig. 5a, Extended Data Fig. 9a and Supplementary Note). FMeQTLs located in gAREs are also more likely to interrupt TF binding sites than those at other AREs across tissues ($P < 1 \times 10^{-20}$), providing independent evidence for FMeQTLs functioning within gAREs (Extended Data Fig. 9b and Supplementary Note). These results affirmed the functional dependency between FMeQTL and gARE, suggesting that we can use FMeQTLs to pinpoint gAREs for target genes. We thus used the genomic distance between an FMeQTL and its proximal gARE (gARE-dist-to-FMeQTL) and the most significant $P$ value of gARE-proximal eQTLs of a target gene (gARE-proximal-eQTL) as metrics to quantify the gARE–gene linkage.

We also noticed the limitation of this approach—a substantial part of these gARE–gene links do not show shared genetic regulation (26–61% across four tissues) at either the locus level (coloc PP4 < 0.1) or the SNP level (haQTL $P > 0.1$; Extended Data Fig. 9c). Alternatively, we directly inferred gARE–gene linking based on shared genetic regulation[33] with four linking scores (Supplementary Note) as follows: (1) coloc PP4, the posterior probability of sharing the same causal variant for gARE–gene pair determined by coloc analysis[52]; (2) coloc PP4/PP3, the ratio of the posterior probability of the pair sharing the same causal variant (PP4) versus having different causal variants (PP3) based on coloc; (3) MR, the causal effect of gARE activity on gene expression based on haQTL and eQTL and (4) ExpPGS-gARE-corr., the correlation between the gARE activity and the genetically inferred gene expression. These scores can capture additional gARE–eGene linkages that are missed by the first approach (Fig. 5b).

We next compiled gLink scores with six gARE–eGene linking scores from the two approaches above to analyze all candidate gARE–eGene (genes with eQTL) pairs, where gAREs are located between 2 kb and 1 Mb from the eGene's TSS (Supplementary Table 7 and Supplementary Note). We then tested their performance by using each of the gLink scores as the benchmark. We found that scores from the second approach show higher consistency among each other compared to the scores from the first approach, and gLink scores show overall higher performance than the score based on the distance between ARE and gene (ARE-gene-dist; Extended Data Fig. 9d). We also compared gLink scores with the following two state-of-art linking scores: (1) EpiMap score[19], which is based on the correlation between gene expression and ARE activity across different tissues and cell types and the genomic distance between the pair and (2) ABC score[17], which is based on enhancer–gene chromatin interaction frequency weighted by the enhancer activities. Coloc PP4 and gARE-distance-to-FMeQTL consistently perform better than EpiMap and other gLink scores based on area under precision–recall curve (AUPRC) across tissues with ABC score as the benchmark (Fig. 5c). However, all gLink scores showed poor performance with EpiMap score as the benchmark (Extended Data Fig. 9e). These results suggest that gLink and EpiMap scores may prioritize different subsets of enhancer–gene links.

## Downstream ARE–gene circuits for GWAS variants

As gLink scores identify gARE–eGene pairs based on genetic variation, we hypothesized that gLink scores might prioritize gAREs associated with disease/trait impactful genetic variants. This was confirmed in 19 GWAS results, where at least one tissue-matched gLink score substantially prioritized higher fractions of GWAS–haQTL-colocalized gAREs based on the linking scores to target genes compared to the background, showing consistently higher performance than ARE-gene-dist, EpiMap and ABC score (Fig. 6a, Extended Data Fig. 10a and Supplementary Note).

**Fig. 5 | gLink scores as a novel framework to prioritize gARE–eGene linking.** **a**, gLink approach 1, quantifying gARE–eGene linking based on eQTL proximity to ARE. Top, a schematic view of the approach. Bottom, enrichment of the QTL-proximal regions of gAREs over other regions; x axis, percentage of QTL-proximal regions among all target regions; y axis, fold enrichment of QTL-proximal regions in target regions over those in shuffled genomic regions (the fourth background genomic region in the legend); colors denote types of QTL and symbols denote different target regions as shown in the legend. **b**, gLink approach 2, quantifying ARE–gene linking based on shared genetic regulation. Top, a schematic view of the approach. Middle, solid dots denote the gAREs linked to *MADCAM1* supported by different scores (by row) in different colors; gLink scores are grouped by two approaches with label on the left; the gARE labeled by black dashed vertical line (right one) is supported by scores from both approaches; the ARE labeled by green dashed vertical line (left one) is supported by scores only from approach 2. Bottom, correlation of polygenic gene expression of *MADCAM1* (y axis) and activity of the ARE only captured by scores from approach 2 (x axis) across individuals, with $r^2$ and $P$ value (two-sided based on linear regression) labeled, and regression line (solid red) and its 95% confidence interval (dashed blue) shown. **c**, Comparison of gLink scores and other scores to the ABC score as the benchmark dataset. Top, a schematic view of candidate gAREs and different linking scores considered; Bottom, truncated PRC (x axis: 0–0.2) for gLink scores and EpiMap across four tissues; the inset shows the AUPRC (x axis) for different scores and background; the black dashed line shows the background proportion of the positive links based on the ABC score.

We then integrated GWAS–haQTL-colocalized gAREs (PP4 ≥ 0.1) and different gARE–gene link scores to prioritize target genes. For brain-related traits, gLink scores perform better in identifying disease target genes supported by GWAS–eQTL-colocalization (Fig. 6b). We then compared the genes we prioritized to the target genes predicted previously for brain-related traits[55], which were based on GWAS summary statistics and gene features, independent of eQTL.

For five brain-related traits where any of the scores show substantially enriched overlap with the predictions from the literature, we found that at least one of our gLink scores showed significant and higher enrichment of benchmark genes than ABC and EpiMap scores (Extended Data Fig. 10b). In addition, we compared target genes for brain-related GWAS predicted based on different linking scores. Results from gLink scores approach 2 are more consistent with each other compared

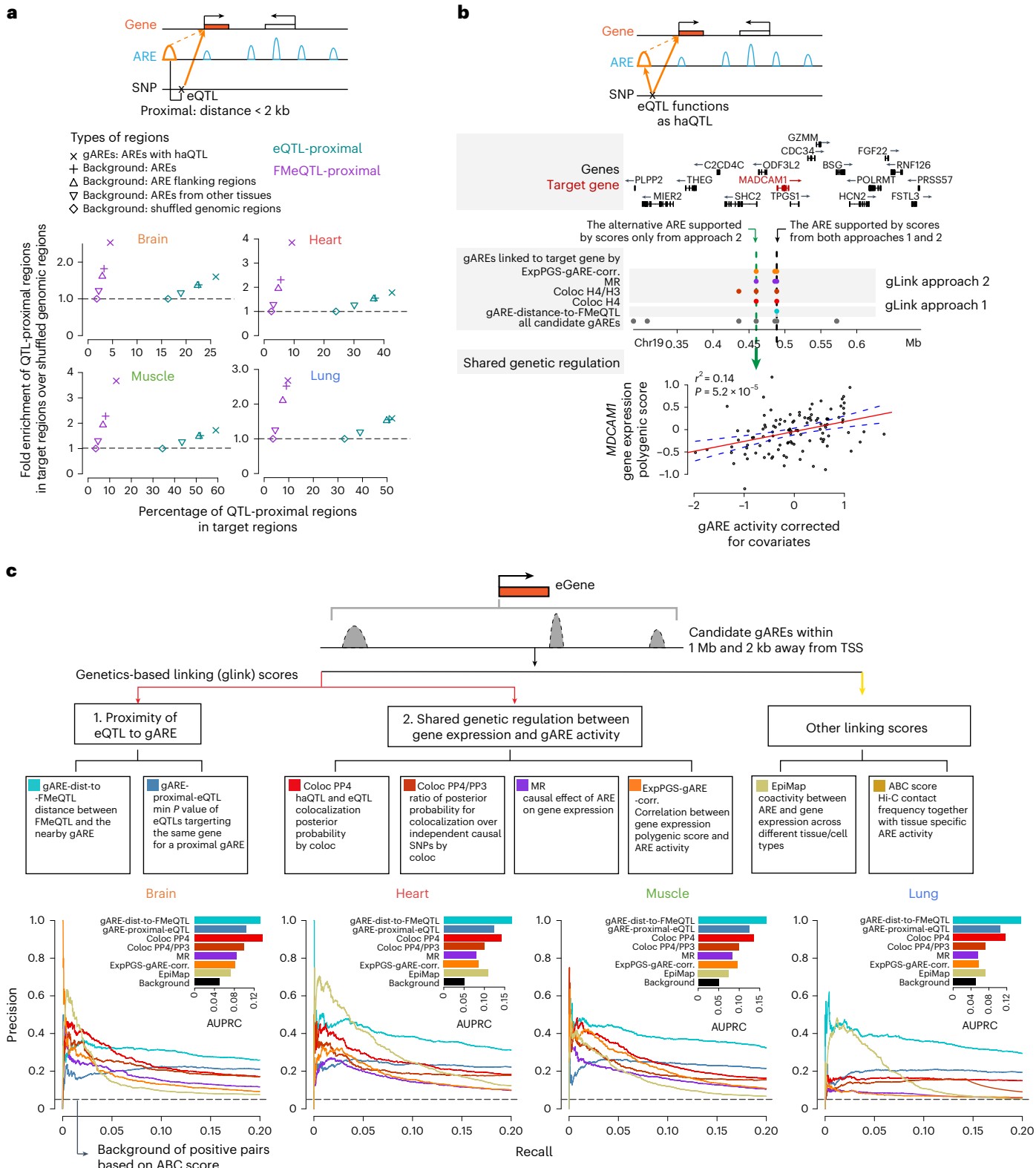

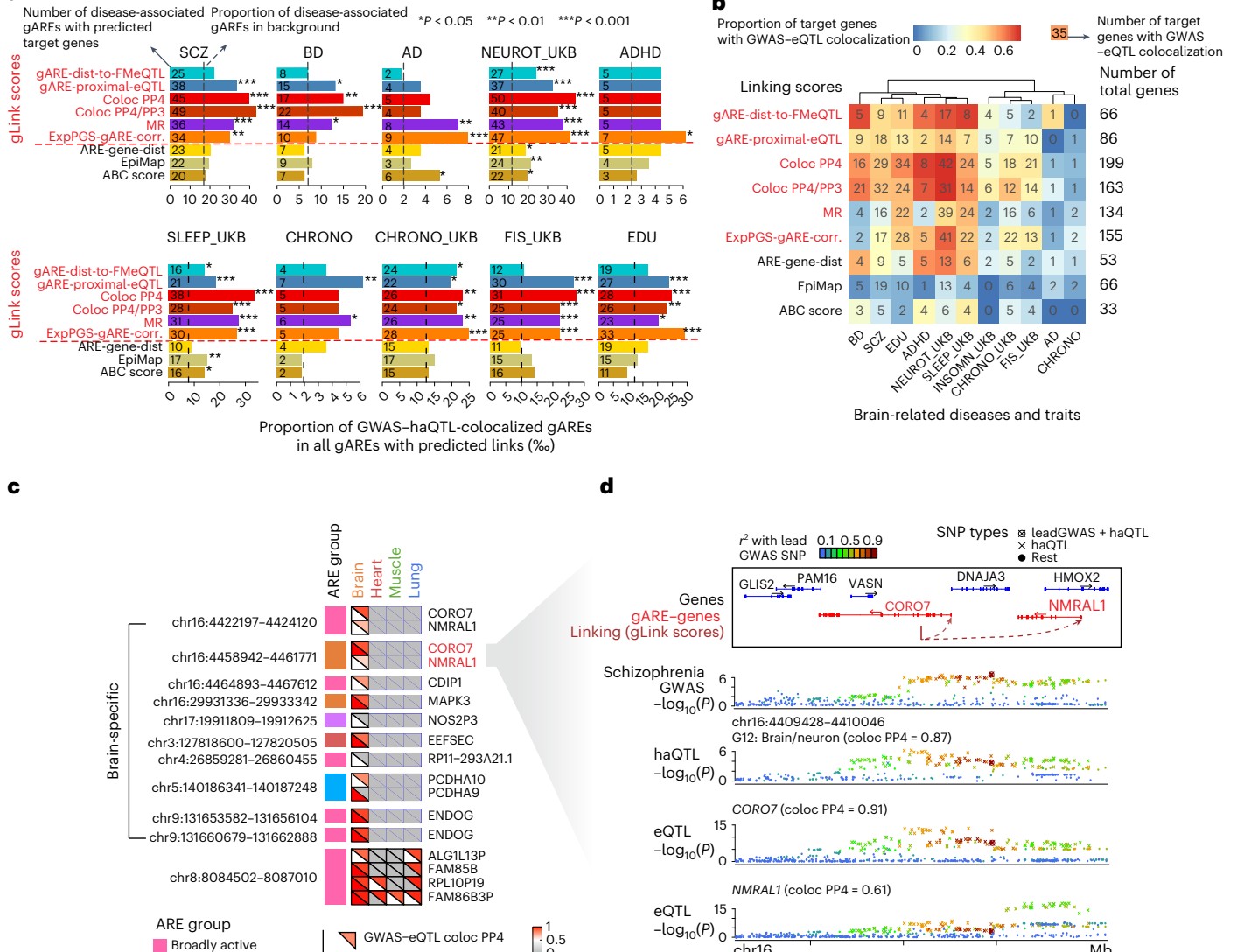

**Fig. 6 | gLink scores prioritize gARE–gene circuits for diseases and traits.**
**a**, Enrichment of GWAS–haQTL colocalized gAREs in gAREs with predicted links for brain. *X* axes, proportion of GWAS–haQTL-colocalized gAREs in gAREs with predicted ARE–gene links based on different linking scores (per mile); different rows in each panel denote different linking scores (gLink scores are shown in red text). Asterisk shows the significance level (one-sided proportion test); exact *P* value is shown in Supplementary Table 7. **b**, Comparison of the number of target genes from different linking scores with GWAS–eQTL colocalization. The heatmap shows the proportion of target genes prioritized by each linking score (by row) that are supported by GWAS–eQTL colocalization in brain tissue for each of ten diseases/traits in **a** (by column); the overlapped gene number is shown in each cell; the total gene number for each row is shown on the right. **c**, Schizophrenia GWAS–haQTL-colocalized gARE–gene circuits in the brain. The heatmap shows the genetic evidence of association between target gene and schizophrenia for each gARE–gene circuit (by row) in each tissue (by column); genomic position of ARE and ARE group is shown on the left; for each cell, the upper triangle shows the evidence supported by GWAS–eQTL colocalization (PP4), and the lower triangle shows the number of gLink scores that connect GWAS–haQTL-colocalized gARE to the same gene. **d**, Schizophrenia GWAS and brain QTL signals at *CORO7–NMRAL1* locus prioritized in **c**. From top to bottom, gene and gARE–genes linking annotation, schizophrenia GWAS signals, haQTL signals for gARE (chr16:4409428–4410046) and eQTL signals for *CORO7* and *NMRAL1* in brain with colocalization PP4 shown in parenthesis; *P* values are all nominal *P* (two-sided) for either GWAS (Chi-squared) or QTL (linear regression). AD, Alzheimer's disease; NEUROT_UKB, neuroticism; ADHD, attention-deficit/ hyperactivity disorder; SLEEP, sleep duration; CHRONO, chronotype; FIS, fluid intelligence score, EDU, education years; UKB, GWAS from UK biobank (detailed information in Supplementary Table 5);.

to others (Extended Data Fig. 10c). Overall, gLink scores tend to prioritize distal genes compared to ABC score and ARE-gene-dist (Extended Data Fig. 10d), suggesting that they provide a unique enhancer–gene link framework to understand *cis*-regulatory circuits associated with disease genetics.

We then identified 1,284 (brain), 1,005 (heart), 2,006 (muscle) and 734 (lung) high-confidence gARE–eGene circuits using a

unified score derived from gLink scores (Supplementary Table 8 and Supplementary Note) and looked into specific circuits for schizophrenia. They include 11 gAREs linking 14 genes, with 12 of those genes supported by GWAS–eQTL colocalization, suggesting their potential roles in schizophrenia etiology (Fig. 6c). Although most of these gAREs are tissue-shared, 13 out of 17 disease-associated gARE–eGene circuits are brain-specific. This underscores the influence

of multiple factors in tissue specificity of genetic regulation, including genetic regulation of an ARE, its association with GWAS signal and gene–ARE linking. For example, at a schizophrenia GWAS locus on chromosome 16, a brain/neuron (G12) gARE links to two target genes *CORO7* and *NMRAL1*, both previously genetically correlated to the disease[56] (Fig. 6d). Their regulation of vesicle transport[57] and cellular response to redox changes[58], respectively, and their links to the same gARE imply a coordinated dysregulation of both processes in schizophrenia etiology[59,60]. We also detected 11 regulatory circuits in the heart or muscle potentially contributing to schizophrenia genetics (Extended Data Fig. 10e), also supported by GWAS–eQTL colocalization analysis across multiple heart or muscle tissues in GTEx (Extended Data Fig. 10f). This implies that haQTL signals and gLink scores in muscle and heart may capture disease genetic signals in tissue-shared vascular cells. Two of these six genes, *WBP1L* and *MFHAS1*, were previously identified as marker genes for brain perivascular fibroblast subtype and capillary endothelial cell subtype[61], respectively, indicating their potential roles in vascular homeostasis in psychiatric disorders[62].

## Discussion

In this study, we generated a multitissue, multi-individual epigenomic dataset with 387 H3K27ac maps in brain, heart, muscle and lung GTEx samples. We characterized the association of 282 k AREs with tissues/cell types, sex and genetics and inferred enhancer–gene links using genetic, epigenomic and transcriptomic covariation. Finally, we used these integrative analyses to gain new insights into diseases/traits genetics.

We investigated the primary tissues/cell types of activity for AREs detected in this study (Fig. 2e), providing the epigenomic contexts for the overlapping GWAS variants and helping prioritize tissues/cell types of action for diseases and traits (Fig. 4a). For example, we identified smooth muscle cells for cardiac disease and lung functions and immune cells for autoimmune and psychiatric diseases. Our large sample size also enabled us to identify newly detected AREs (G14) missed by the reference epigenomes. Despite their low activity, G14 was substantially enriched for neuron-related pathways (Extended Data Fig. 2c) and GWAS signals from psychiatric or neurologic traits (Fig. 4a), suggesting their potential as proto-enhancers in neural development[63]. We also investigated the genetic regulation of AREs, revealing three types of gAREs, particularly 'ARE-specific' and 'haQTL-specific' gAREs. They show different mechanisms of tissue-specific genetic regulation and were reported to provide crucial context information for GWAS signals in previous[31,64] and current studies (Fig. 4a,c), highlighting an important direction for future research.

We found that most of the 'bulk-eQTL-missing GWAS loci' (Fig. 4e,f) could be captured by cell-type-level eQTL signals. However, we applied stringent and permissive cutoffs for GWAS–haQTL (PP4 < 0.5) and GWAS–eQTL (PP4 < 0.1) colocalization, respectively, to detect robust loci missed by eQTL signal. Using the same cutoff of 0.5 for both, we found 11 GWAS loci captured by haQTL but not by either level of eQTLs (Extended Data Fig. 8c and Supplementary Note). Weak GWAS–eQTL colocalization signals might be due to eQTLs from nondisease-relevant conditions missing condition-specific regulation. This, coupled with cell-type-specific factors, may affect eQTL detection, explaining the 'missing regulation' bias noted in a previous report[65]. These results also indicate that bulk epigenomic variation may capture the impact of genetic variants (as shown here by haQTL effects) with greater power than bulk eQTL for certain loci where the effect on gene expression may only become visible in specific cell types under specific conditions.

We proposed gLink scores, a framework of enhancer–gene linking based on genetics. Our contributions include the following factors: (1) distinguishing two existing enhancer–gene linking methods based on eQTL[66] and shared genetic regulation[33]; (2) refining the first one by focusing on gAREs; (3) providing a uniform framework of gLink scores to integrate the two and (4) discriminating gLink scores from other state-of-art approaches based on prediction consistency, distribution of distances between each pair and ability to pinpoint genes for GWAS–haQTL-colocalized gAREs. Unlike EpiMap, gLink scores prioritize enhancer–gene pairs based on interindividual genetic variation within the same tissue, rather than intertissue/cell type variation. gLink scores show stronger enrichment for GWAS–haQTL-colocalized gAREs compared to gene–ARE distance, EpiMap and ABC scores, likely due to their capacity to capture gARE-proximal genetic signals related to gene expression. Therefore, gLink scores offer a unique and complementary perspective on the enhancer–gene regulatory network.

Despite the unique opportunity provided to bridge genetic and phenotypic variation, our study has limitations. Our sample size and read depth (Extended Data Fig. 6b,c) were underpowered to detect haQTL signals, affecting haQTL sharing estimation (Fig. 3d,e), tissue specificity correlation between haQTL and eQTL for lung (Extended Data Fig. 7e), GWAS–haQTL colocalization (Extended Data Fig. 8c) and gLink scores' performance (especially in the lung; Fig. 5c). Future research could increase the power by augmenting sample sizes or utilizing deep models to borrow regulation information across the genome[67,68]. Secondly, our epigenomic signals lacked cell-type resolution. Although we developed a deconvolution strategy for bulk samples and mitigated confounding factors due to cell fraction changes, we could not accurately deconvolve ARE activity and estimate cell-type-level haQTL without cell-sorted H3K27ac ChIP-seq data. Future studies could address this by generating cell-type-level reference epigenomic profiles for deconvolution or by mapping cell-type-level haQTL from enough cell-sorted samples across individuals. Thirdly, the majority of GTEx samples was from healthy individuals, mostly of European descent. Future studies should incorporate samples from disease-relevant contexts and diverse ancestries to enhance power and accuracy in identifying disease-related and ancestry-relevant GWAS–haQTL-colocalized gAREs and regulatory links for precision medicine purpose[69,70].

Taken together, our dataset and integrative analyses, including ARE groups, enriched TFs binding sites in each group, sex-biased AREs, gAREs, haQTLs and their tissue sharing, gLink scores, GWAS–haQTL-colocalized gAREs and gARE–gene linking, provide a comprehensive view of ARE regulation in human primary tissues and their implications in disease genetics. Our analyses help elucidate functions of GWAS variants, boosting the efforts to accurately identify the context of function. Our dataset represents a unique resource, together with the GTEx dataset and other eGTEx projects, to explore the genetics of gene expression from different regulatory layers with samples of the same cohort.

## Online content

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

## Methods

### Samples for H3K27ac ChIP–seq

Samples were collected by the GTEx Consortium. The donor enrollment and consent, informed consent approval, histopathological review procedures, and biospecimen procurement methods and fixation were the same as previously described[22]. No compensation was provided to the families of participants. Massachusetts Institute of Technology Committee on the Use of Humans as Experimental Participants approved that this study does not involve human participants as defined by federal regulations. H3K27ac profiling was carried out across four human primary tissues, including brain prefrontal cortex ($n = 113$), heart left ventricle ($n = 100$), skeletal muscle ($n = 108$) and lung ($n = 66$; Supplementary Table 1).

### Statistics and reproducibility

No statistical method was used to predetermine sample size, which was mostly based on the availability of the samples in the GTEx cohort. The experiments were not randomized, and the investigators were not blinded to allocation during experiments and outcome assessment, as the study is not a randomized controlled trial study. We used data from 387 samples that passed the quality control as described in the section below in the downstream analysis. We carried out most statistical tests with R (v3.6.0) unless specified, including Fisher's exact test and proportion test for enrichment analysis with basic function fisher. test and prop.test in R, respectively, as well as test for sex-biased ARE identification with R packages limma voom[71] and sva[72]. We mapped haQTL with linear regression implemented in FastQTL[46]. Please refer to the corresponding sections in Methods or Supplementary Note for details on statistical tests.

### Nuclei isolation and crosslinking protocol

Tissues were sliced up in seven to eight small chunks while kept on ice at all times. The pieces were transferred to 2 ml Safe-Lock Eppendorf Tubes (round bottom). NF-1 buffer (1 ml; 0.5% Triton X-100, 0.1 M sucrose, 5 mM $MgCl_2$, 1 mM EDTA and 10 mM Tris–HCL pH 8.0) with proteinase inhibitors (Roche complete tablets), and sterile RNAse- and DNAse-free steel beads were added to the tube.

Qiagen Tissuelyser allowed for the processing of 48 samples at the time and was used to perform the following lysing of the tissue. Heart and muscle tissues were processed for 2–2.5 min at 30 Hz. Lung and brain tissues were processed for 2.5–3 min at 30 Hz. The samples were then transferred to 1.5 ml tubes and centrifuged at 40 °C at 1,000$g$ for 10 min. The supernatant was discarded, and 937.5 µl of PBS-containing proteinase inhibitor was added. Crosslinking was performed by adding 62.5 µl of 16% formaldehyde (final concentration of 1%) and put on an elliptical rotator for 15 min at room temperature. The formaldehyde crosslinking was stopped/quenched by adding 125 mM glycine (50 µl of 2.5 M glycine stock solution) and left at room temperature for 5 min. The samples were then spun at 2,200 rpm for 10 min at 4 °C. The supernatant was then discarded, and the pellet was resuspended in 1 ml PBS-containing proteinase inhibitor. The sample was spun again at 2,200 rpm for 10 min at 4 °C. The supernatant was discarded, and the pellet was resuspended in 200 µl of PBS-containing proteinase inhibitor. The samples were then transferred to a 2D barcoded tube, flash frozen, combined in 96 samples 2D barcoded sample racks and sent for ChIP processing.

### ChIP–seq protocol

Sample plates containing the cross-linked samples were completely thawed on ice. The cross-linked sample pellets were resuspended in 100–300 µl of 1% SDS lysis buffer + protease inhibitors for 10 min on ice (1% SDS, 10 mM EDTA and 50 mM Tris–HCL pH 8.1). The volume was then adjusted to 1 ml with ChIP dilution buffer + protease inhibitors (16.7 mM Tris–HCl pH 8.1, 167 mM NaCl, 0.01% SDS, 1.1% Triton X-100 and 1.2 mM EDTA). The samples were then sheared following the previously optimized protocol using a Covaris LE-200 sonicator.

The desired amount of sheared chromatin (equivalent to 1 million cells per ml) was normalized to the desired concentration and volume, using ChIP dilution buffer (16.7 mM Tris–HCl pH 8.1, 167 mM NaCl, 0.01% SDS, 1.1% Triton X-100 and 1.2 mM EDTA) keeping the final concentration of SDS to less than 0.1%. The desired antibody (1 µg; Cell Signaling Technology, 8173; 1:100 dilution) was added to each tube per well and incubated overnight at 4 °C.

Protein A/G (50 µl; 50/50) magnetic beads per reaction were independently added with blocking buffer (1 ml) containing proteinase inhibitor and incubated on a 360° rotator at 4 °C. The tubes were placed on a magnet, the supernatant was discarded and an additional 100 µl of blocking buffer was added to each tube per well. This step was repeated one more time. The tubes per well were incubated at 4 °C on head over tail rotator at 4 °C for 1 h. The ChIPs (chromatin + antibody complex) from rotation at 4 °C were combined with the blocked beads and incubated for 1 h at 4 °C. The ChIPs were removed from rotation and briefly spun to remove any sample from the caps and placed on a magnet. The supernatant was discarded. The ChIPs/beads complexes were removed from the magnet and washed with 175 µl of RIPA/140 mM NaCl buffer (cold; 0.1% DOC, 0.1% SDS, 1% Triton X-100, 140 mM NaCl, 1 mM EDTA and 20 mM Tris–HCl pH 8.1). The ChIPs/beads complexes were washed two more times with 200 µl of RIPA/500 mM NaCl Buffer (cold; 0.1% DOC, 0.1% SDS, 1% Triton X-100, 500 mM NaCl, 1 mM EDTA and 20 mM Tris–HCl pH 8.1). The ChIPs/beads complexes were washed twice with 200 µl LiCl Buffer (cold; 0.25 M LiCl, 1% NP40, 1% Na deoxycholate, 1 mM EDTA and 10 mM Tris–HCl pH 8.1). Then, washed twice with a TE buffer. ChIP Elution Buffer (50 µl; 10 mM Tris–HCl pH 8.0, 5 mM EDTA, 300 mM NaCl, 0.1% SDS and directly before use 5 mM DTT) and reverse crosslinking buffer (8 µl; 250 mM Tris–HCl pH 6.5, 1.25 M NaCl, 62.5 mM EDTA, 5 mg ml$^{-1}$ proteinase K, and 62.5 µg ml$^{-1}$ RNAse A) was added to each well. The samples were incubated at 65 °C for 3 h or overnight. Note that the negative control ChIP samples were parallelly reverse cross-linked at that step.

The reversed cross-linked samples were placed on a magnet. The supernatant was transferred to a new tube/plate. Each sample was successively 1.8× SPRI beads cleaned and 2.2× SPRI beads cleaned. Each sample was washed twice with 200 µl 70% ethanol. The ChIP material was eluted in 22 µl of 10 mM Tris–HCl pH 8.0, at room temperature for 3 min.

We quantified ChIP material using Qubit, constructed ChIP–seq library with Nextera XT DNA Library Prep Kit (Illumina) according to the manufacturer's instruction and sequenced for 101-bp paired-end reads by Illumina HiSeq 2500 at Broad Technology Labs (BTL) in the Broad Institute. BTL implemented ChIP–seq as a robust, standardized, low-variability, automatically optimized procedure. We have automated all steps of the ChIP–seq protocol, including shearing, immunoprecipitation and library preparation, with all steps after shearing handled in series on a single liquid-handling robot with very limited human intervention. This has allowed major sources of variability and failure to be minimized or eliminated, including overcoming lot-to-lot variability in antibody performance.

### Quality control of H3K27ac ChIP–seq

We carried out peak calling with the ENCODE pipeline (https://github.com/kundajelab/chipseq_pipeline)[18]. Briefly, we mapped reads to human genome assembly hg19 (bwa, v0.5.9 (ref. [73]), command: bwa aln Homo_sapiens_assembly19.fasta -q 5 -l 32 -k 2 -t $NSLOTS -o 1 -f), filtered low-quality and multiple-mapped reads (samtools, v1.3.1 (ref. [74])) and applied MACS2 (v2.1.1)[75] for peak calling. Samples were then filtered based on the following three QC metrics (Supplementary Table 1): (1) relative strand cross-correlation coefficient (RSC), based on the (shifted) agreement between forward-strand and reverse-strand reads, providing a measure of signal-to-noise enrichment ratio; (2) sequencing depth, counted as total reads; (3) H3K27ac signal correlation between our samples and the samples from Roadmap

epigenomes based on 405 k tissue-specific AREs identified from Roadmap. We defined 'Tier 1' samples as those with RSC score ≥ 0.8, total reads ≥ 10 M and for which their matched Roadmap tissue was in the top three most correlated Roadmap epigenomes. We used high-quality samples of 'Tier 1' to generate a peak set in each tissue. We defined 'Tier 2' samples with relatively permissive cutoffs—RSC score ≥ 0.6, total reads ≥ 5 M and the same criteria of matched-tissue correlation. Also, we specifically included lung samples with RSC score ≥ 0.8 and total reads ≥ 5 M in 'Tier 2' samples, regardless of whether they passed the criteria of matched-tissue correlation, considering the limited lung samples in Roadmap. These criteria resulted in 113 brain, 100 heart, 108 muscle and 66 lung samples, with median total unique reads of 10.8 M, 11.7 M, 9.9 M and 10.5 M for each tissue, respectively.

### ARE detection, reference ARE set generation, ARE activity quantification and normalization

We only included H3K27ac ChIP–seq peaks ($q$-value ≤ 0.01) present in at least two 'Tier 1' samples of each tissue in the tissue ARE set or overlapping with peaks from a tissue-matched reference sample from the ROADMAP project, resulting in 335 k, 363 k, 227 k and 148 k AREs in brain, heart, muscle and lung, respectively. We further filtered AREs based on their overlap with promoter- and enhancer-related chromatin states in tissue-matched samples in Roadmap (overlapping region ≥ 50 bp), resulting in 189,681, 131,952, 142,822 and 107,151 AREs for brain, heart, muscle and lung, respectively. AREs from the four tissues were merged as a reference set of 282 k AREs (BEDTools, v2.26.0 (ref. [76])).

We also carried out ARE replication analysis, by repeating ARE detection for each tissue with the same procedure in 80% of Tier 1 samples randomly selected 10 times, and checked the proportions of these AREs that are also detected in the rest of Tier 1 and all Tier 2 samples. We observed a replication rate of AREs at 84% in brain, 83% in heart, 88% in muscle and 78% in lung. For newly detected AREs (G14), these numbers are slightly lower, as expected, because these AREs vary more across samples and are thus less likely to replicate using our metrics—they are 80% in brain, 68% in heart, 77% in muscle and 55% in lung.

We quantified the activity of each ARE for each sample based on fragment coverage on a specific ARE divided by its length:

$$\text{Activity}_{i,k} = \sum_{j=1}^{N}(\text{overlap}(\text{Frag}_{j,k}, \text{ARE}_i))/\text{length}_i,$$

where $\text{Activity}_{i,k}$ represents activity of $\text{ARE}_i$ in sample $j$, $\text{Frag}_{j,k}$ represents a fragment $j$ extended from a read $j$ toward the 3' end to the same length as the estimated fragment length in sample $k$, overlap($\text{Frag}_{j,k}$, $\text{ARE}_i$) represents the length of the overlapping part between $\text{Frag}_{j,k}$ and $\text{ARE}_i$, and $\text{length}_i$ is the length of $\text{ARE}_i$. ARE activity was normalized for read depths of different samples, by a factor estimated for each sample based on geometry mean of read coverage of all AREs as implemented in DESeq2 package[77] and then corrected for bias due to GC-content[78].

### Identification of ARE modules, submodules and groups

We identified ARE modules based on a previous approach[19]. Briefly, we extracted -log10P signal from 240 EpiMap H3K27ac samples for each ARE in our reference set (bigWigAverageOverBed, v2 (ref. [79])), binarized the matrix by a cutoff of 2 and clustered by a k-centroids algorithm implemented in R package flexclust[80] (V1.4-0) with the Jaccard index as the distance metric and cluster number set as 141 (corresponding to 2,000 AREs per module on average). To check the robustness of our modules, we further clustered AREs from each module into submodules following the same approach above, with the cluster number set corresponding to 200 AREs per submodule on average. The resultant 1,413 submodules showed largely homogeneous ARE activities across submodules of the same module both for our samples and for the

reference epigenomes, indicating that further clustering might not lead to new biologically interpretable subclusters (Extended Data Fig. 2b).

We then classified ARE modules into 14 groups with multiple steps as follows: (1) defining ubiquitous score as the proportion of samples with average ARE activity ≥ 0.2 for each ARE module; (2) defining ARE modules as broadly active (G1) if their ubiquitous scores were above 0.5; (3) defining ARE modules as multitissue (G2) if their ubiquitous scores were between 0.1 and 0.5; (4) defining the ARE module with no activity in any of the samples as newly identified (G14); (5) clustering 240 reference samples into 11 clusters based on average ARE signals across 127 modules by hierarchical clustering with 1-Pearson correlation coefficient as the distance metric, naming each sample cluster manually based on the names of majority samples and (6) grouping the rest of the ARE modules based on the sample cluster where the ARE module shows the strongest signal. The ARE modules are visualized by R package ComplexHeatmap (v2.4.3, R-4.0)[81] in Fig. 2b.

### Annotation of ARE modules

We annotated ARE modules based on following different functional categories: (1) housekeeping gene[82]—enrichment calculated for the genes assigned for each ARE module (assign gene to an ARE if within 2 kb); (2) chromatin states—we applied the results from an 18 chromatin states model from matched tissue in EpiMap[19] to annotate ARE (overlap ≥ 50 bp); 'TssA' and 'TssBiv' are counted as promoter states, while 'EnhG1', 'EnhG2', 'EnhA1', 'EnhA2', 'EnhWk', and 'EnhBiv' are counted as enhancer states (Fig. 2b); (3) genomic regions—annotation calculated by R package annotatr (v1.6.0)[83] (Fig. 2b); (4) biological processes from GO—enrichment of nearby genes is calculated by R package rGREAT (v1.2.1)[84,85] (Extended Data Fig. 2c) and (5) TF binding sites—we used position weight matrix of binding sites for TF families collected in EpiMap and scanned ARE modules for enriched motifs by Homer (v4.11.1)[86] with background autoselected, we filtered TF motifs based on adjust $P$ value (cutoff of 0.05) and odds ratio (cutoff of 1.2) and we visualized the enrichment for each TF family (mean odds ratio) (Extended Data Fig. 3a).

We found that specifically broadly active AREs (G1) were enriched for TSS-proximal regions, promoter and enhancer chromatin states, and significantly associated with housekeeping genes ($P < 2.2 \times 10^{-16}$). Multitissue AREs (G2) were enriched for TSS-distal enhancer chromatin states and biological processes, including immune response, posttranscriptional regulation and epithelial tube morphogenesis. Blood/immune AREs (G3, red box) were enriched for immune functions and binding sites of immune-related TFs, including IRF5 and IRF8. Skeletal/cardiac muscle AREs (G9, green box) were enriched for muscle and heart enhancers and promoters, associated with cardiac muscle tissue development pathways and enriched for binding sites of skeletal muscle function-related TF, such as STAT3 (ref. [87]). Brain/neuron AREs (G12, orange box) were enriched for brain enhancers and promoters, associated with *trans*-synaptic signaling, and enriched for binding sites of neuronal-function TFs, such as NEUROG2 (ref. [29]). Lung-active AREs (modules 59 and 92, blue box) were enriched for the binding sites of lung homeostasis-related TFs, such as ERG[88], FOXA2 (ref. [89]) and FOXL1 (ref. [90]).

### Map haQTL

We inferred the latent factors for our samples in each of the four tissues by Peer[45] (R package v1.0) and identified the optimum number of Peer factors (5, 5, 10 and 2 for brain, heart, muscle and lung) to correct for during haQTL calling. Additionally, we took into account age, batches for H3K27ac ChIP–seq experiment, top five genotype PCs, and those technical effects related to platform and PCR. We found strong correlations between the top peer factors and the tissue-archetype fractions estimated, confirming the importance of their removal to discover *cis*-acting genetic variants (brain as an example in Extended Data Fig. 6a). We applied FastQTL (v2.184)[46] to map haQTLs within 100 kb

to the center of each ARE on autosomes for variants with minor allele frequency ≥ 0.05. An empirical $P$ value for each ARE was estimated based on the lead nominal $P$ value and permutation results (--permute 1000 10000) by fitting a β distribution to account for multiple variants tested. We used an empirical $P$ value threshold of 0.005 to identify gAREs as in a previous study[29], and then for each gARE, we applied the nominal $P$ value threshold corresponding to the empirical $P$ value of 0.005 for each locus to identify haQTLs (Fig. 3a).

We used two strategies to estimate and summarize the power of haQTL calling (Extended Data Fig. 6b,c). The first one is using powereQTL (v0.3.4)[91] with the parameters observed from our haQTLs, including sample size, variance of H3K27ac signal and haQTL mean effect size, as well as different values of minor allele frequency. The second strategy is to sequentially and randomly downsample the 109 brain samples to smaller sample sizes ten times each, and measure the fraction of haQTLs that can be recapitulated with these randomly selected samples. As expected, haQTLs peaked within 25 kb of their target gAREs (Extended Data Fig. 6d), validating our choice of 100-kb distance cutoff.

### Tissue sharing of ARE genetic regulation and identification of three types of gAREs

We defined the following three types of ARE/gARE based on their tissue-sharing: type I 'haQTL-shared', where both the ARE and gARE are shared between tissues; type II 'haQTL-specific', where the ARE is shared while the gARE is specific to a certain tissue and type III 'ARE-specific', where the ARE is specific to a certain tissue and thus the gARE is specific too. For type I/II AREs, we estimated haQTL tissue-sharing using a previously reported method which is based on directionality consistency and is less dependent on the discovery power[29]. The underlying reasoning is that the tissue-sharing haQTLs would show consistent directionality between tissues, even if they did not pass the $P$-value threshold due to the power. Specifically, for all the lead haQTLs that are significant in at least one tissue (discovery tissue), we tested whether they show consistent directionality of haQTL effect in each of the rest of the tissues (replication tissues). The directionality consistency was quantified as the percentage of gARE–haQTL pairs that show the same effect directionality (either positive or negative) between the discovery tissue and the replication tissues. We grouped the haQTLs into the following four bins based on their nominal $P$ value in the replication tissues: (1) strong-replicated ($P < 10^{-5}$), (2) medium-replicated ($10^{-5} < P < 10^{-3}$), (3) weak-replicated ($10^{-3} < P < 0.1$) and (4) no-effect ($P > 0.1$) and calculated the directionality consistency for each of these $P$-value bins. The proportion of tissue-sharing (Fig. 3d) haQTLs was quantified as follows: Tissue-sharing $= \sum_{(i=1)}^{nbin} \text{Perc}_i \times [DC_i - (1 - DC_i)]$, where $\text{Perc}_i$ represents the percentage of haQTLs of bin $i$, and $DC_i$ represents the haQTL directionality consistency of bin $i$.

To define haQTL-shared AREs between discovery and replication tissues, we set the nominal $P$-value threshold equal to 0.02 in the replication tissue, which makes the directionality consistency between the two tissues over 95% (Extended Data Fig. 7d).

As an additional approach to detect subthreshold haQTL tissue sharing, we quantified the similarity of haQTL effect size between the discovery tissue and the replication tissue in the same bins as above, defined as the regression coefficient of the effect size between the replication tissue and the discovery tissue. We observed that the haQTLs with stronger $P$ values in the replication tissue tend to show greater similarity in effect size with the discovery tissue. We also observed that the haQTLs from the weak-replicated bin ($10^{-3} < P < 0.1$) show greater similarities (0.3–0.6) than those from the no-effect bin (-0.05; Extended Data Fig. 7b,c). These results further confirm the tissue sharing of subthreshold haQTLs in the replication tissue.

### GWAS signal enrichment in ARE groups by LDSC

We applied LD score regression (v1.0.0)[9,92] with default parameters to estimate the enrichment of GWAS signals from 55 datasets

(Supplementary Table 5) for AREs from each group active in each of the four tissues as genomic annotations. We carried out Benjamini–Hochberg correction for testing multiple ARE groups of each tissue for each trait (adjust $P$ cutoff 0.05).

### Reporting summary

Further information on research design is available in the Nature Portfolio Reporting Summary linked to this article.

## Data availability

H3K27ac ChIP–seq profiles from this study, including the raw data in bam format and the processed bigwig format, are available on dbGaP with accession number phs000424.v8.p2. Additionally, the data can be accessed via AnVIL with authentication: https://anvil.terra.bio/#workspaces/anvil-datastorage/AnVIL_GTEx_V8_hg38. Because the raw sequencing data with genetic information are protected, application and authentication are needed before accessing the data. All nonprotected data of H3K27ac ChIP–seq can be visualized via the GTEx Portal (www.gtexportal.org) as part of eGTEx v8. The eQTL datasets are from GTEx v8, which can be accessed at https://gtexportal.org/home/datasets. Please refer to Supplementary Tables 1–8 for sample metadata, AREs, sex-biased AREs, haQTL, gAREs, GWAS, GWAS–haQTL colocalization and gLink scores. All the processed data in the study are available on Zenodo (https://doi.org/10.5281/zenodo.7992724).

## Code availability

All codes for this study are available on Zenodo[93].

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

## Acknowledgements

We thank all the donors and their families for their generosity in tissue donations to the GTEx research project. We thank the support from and insightful discussion with members of the GTEx Consortium. We thank A. Grayson, P. Purcell, A. Chapman, K. Kang and other Kellis Lab members from MIT for suggestions, and Jared Nedzel and Katherine Huang from the Broad Institute for H3K27ac data visualization at GTEx portal. This work was supported by NIH under grants HG007610, HG008155, MH109978, MH119509, AG058002, NS110453 and MH109978 (to M.K.).

## Author contributions

This study was designed by L.Hou, X.X. and M.K., and directed and coordinated by M.K. B.M. and N.V.W. carried out experiments overseen by S.S. and C.N. L.H. and X.X. carried out analyses and interpreted results with help from Y.P., C.B., B.J., N.S., L.He., A.P., Z.Z., F.A., and K.G.A. L.Hou, X.X. and M.K. wrote the manuscript. All authors participated in the discussion of the project.

## Competing interests

The authors declare no competing interests.

## Additional information

**Extended data** is available for this paper at https://doi.org/10.1038/s41588-023-01509-5.

**Correspondence and requests for materials** should be addressed to Manolis Kellis.

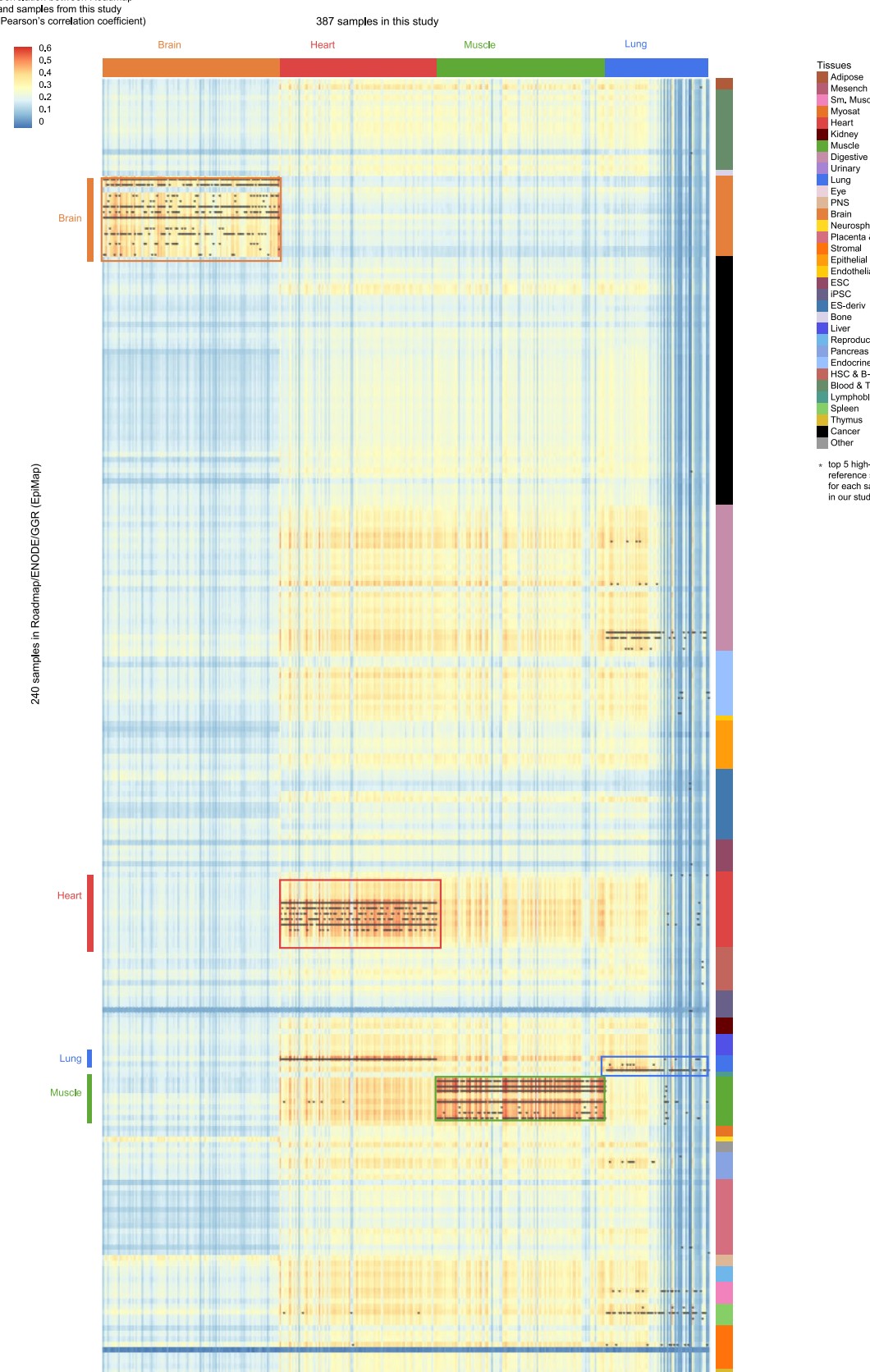

**correlation of H3K27ac profiles between samples in this study and those from reference epigenomes**

**Extended Data Fig. 1 | Correlation of H3K27ac profiles between samples in this study and those from the reference epigenomes.** Each column represents a sample in our study with tissue name on the top, and each row represents a sample from the reference epigenomes; for each sample in our study, the top five highly correlated reference samples are labeled with '*'; orange, red, green and blue boxes indicate tissue-matched pairs between our data and the reference data.

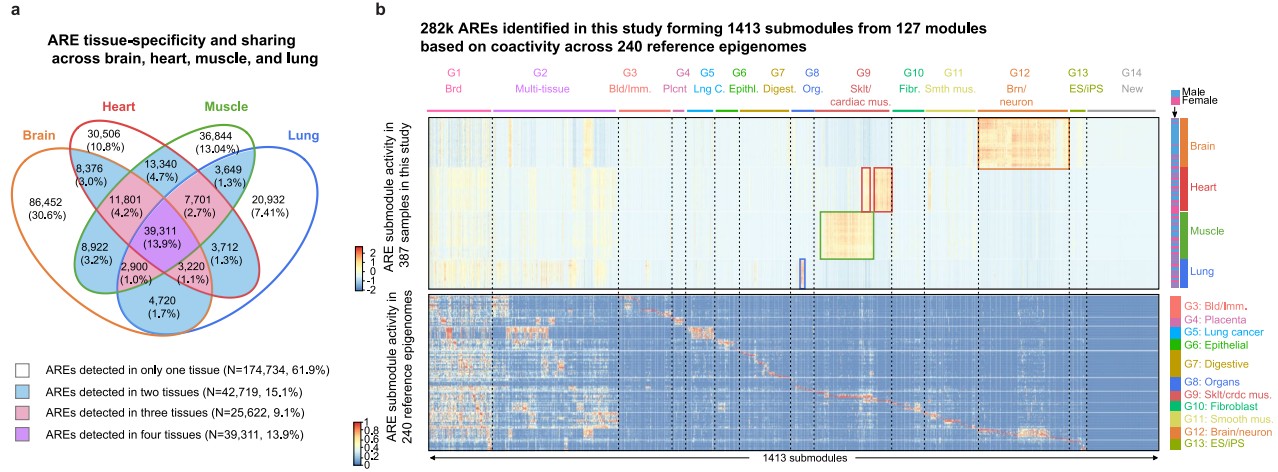

**a**

ARE tissue-specificity and sharing across brain, heart, muscle, and lung

**b**

282k AREs identified in this study forming 1413 submodules from 127 modules based on coactivity across 240 reference epigenomes

**c**

GO Biological Process enrichment for 127 ARE modules

Functional enrichment for
1. G3 Blood/immune
2. G9 Skeletal/cardiac mus.
3. G12 Brain/neuron
4. Lung specific modules

log enrichment

**Extended Data Fig. 2 | See next page for caption.**

**Extended Data Fig. 2 | Tissue specificity of AREs and functional annotations of ARE modules. a**, ARE tissue-specificity and sharing across brain, heart, muscle, and lung. The Venn diagram shows the numbers and proportions of AREs for different combinations of tissue-sharing across four tissues. **b**, 282k AREs identified in this study form 1413 submodules from 127 modules based on coactivity across 240 reference epigenomes. Upper panel: ARE activity of 1413 submodules (by column) across samples (by row) in our study; orange, red, green, and blue boxes showing tissue-specific modules for brain, heart, muscle, and lung, respectively; sex and tissue information are on the right. Lower panel: ARE activity of 1413 submodules in the reference epigenomes; sample clusters annotated on the right. **c**, GO biological processes enrichment for 127 ARE modules. Each row represents a GO term and each column represents an ARE module with ARE group labeled at the bottom; red, green, orange, and blue boxes indicate the enrichment for G3, G9, G12, and lung-specific modules.

**a**    motif enrichment for enhancer modules

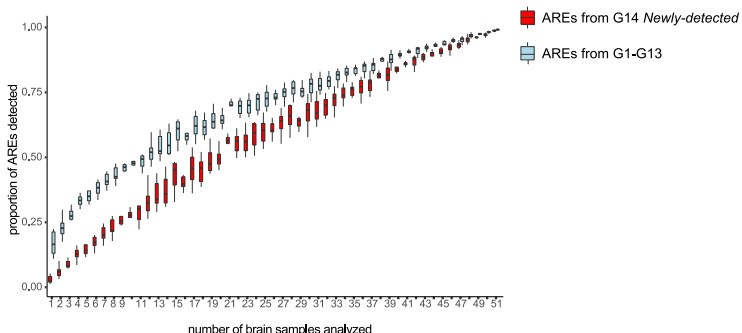

**b**

Comparison of detection rates between the AREs from G14 Newly-detect and the other groups (n=10).

**Extended Data Fig. 3 | See next page for caption.**

**Extended Data Fig. 3 | TF motif enrichment of ARE modules and ARE detection power. a**, Motif enrichment for enhancer modules. Each row denotes a TF family, represented by the TF labeled on the right having the strongest odds ratio across modules; each column represents an ARE module with ARE group labeled at the bottom; red, green, orange, blue, and purple boxes indicate enrichment for G3, G9, G12, and lung-specific modules. **b**, Comparison of ARE detection rates between *Newly-detected* ARE (G14) and the other groups. X-axis shows the number of brain samples randomly selected for each experiment; y-axis shows the proportion of AREs detected from each experiment; colors denote which groups AREs are from; n = 10 independent times of sampling for each box.

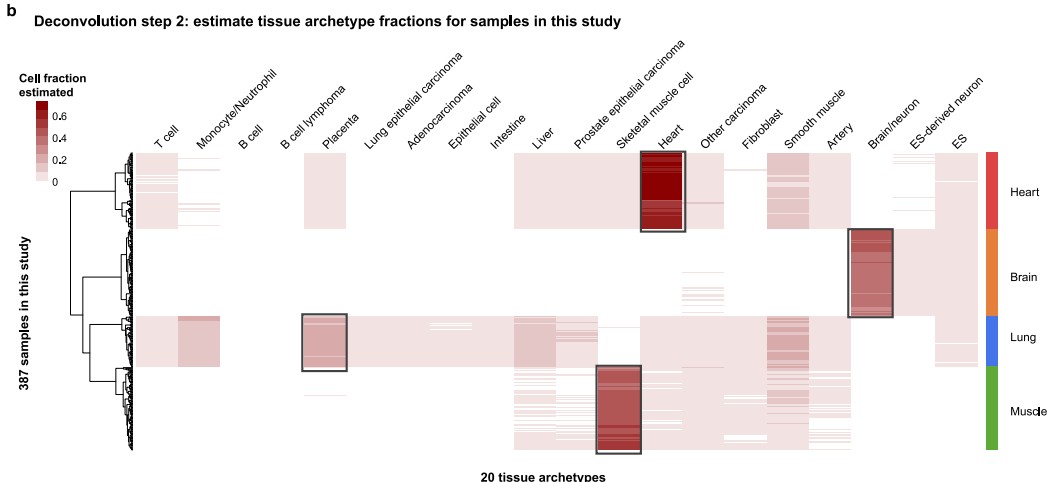

**Extended Data Fig. 4 | See next page for caption.**

**Extended Data Fig. 4 | Tissue-archetype fraction estimation. a**, Deconvolution step 1. The heatmap shows the correlation between the profiles of tissue-archetype (by column) and the profiles for the reference samples (by row) with strong tissue-archetype specific patterns. Typical sample names are shown on the right, four samples that are not clustered with other tissue-matched samples in Fig. 1b and mentioned in the section of 'Comparison of H3K27ac profiles across studies' are labeled on the left. **b**, Deconvolution step 2. The heatmap shows the fraction of each tissue-archetype (by column) estimated for samples (by row) in each of our tissues, with the primary tissue-archetypes indicated by gray boxes.

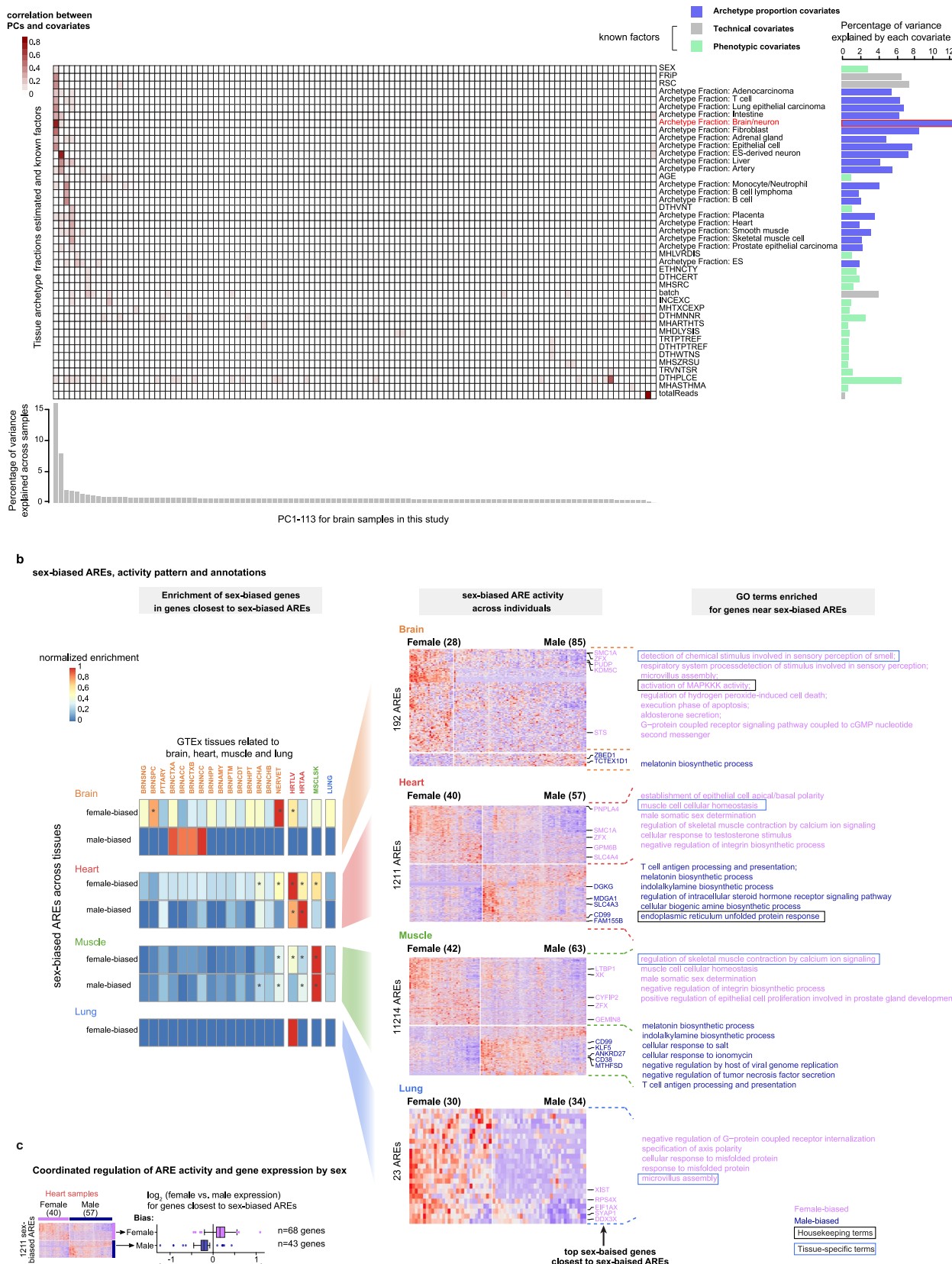

**Extended Data Fig. 5 | See next page for caption.**

**Extended Data Fig. 5 | Sex-biased ARE identification. a**, Comparison between principal components (PCs) and covariates including estimated tissue-archetype fractions and known factors for brain samples. Top left panel: heatmap shows the correlation between PCs (by column) and known factors (by row); top right panel: percentage of variation (x-axis) explained by the covariates (by row), with red highlight for the primary tissue-archetype identified in Extended Data Fig. 4a; bottom panel: the percentage of variation (y-axis) explained by each PC for brain samples (by column). **b**, Sex-biased AREs, activity pattern and annotations. Left panel: enrichment of sex-biased genes from matched GTEx tissue (by column) in genes closest to sex-biased AREs identified from this study; * denotes strong enrichment (adjusted $P < 0.1$, two-sided Fisher's exact test, BH correction

across multiple tissues tested, shown in Supplementary Table 2); middle panel: sex-biased ARE activity of each sample (by column) in each tissue, with top 5 sex-biased genes closest to any sex-biased ARE labeled on the right; right panel: GO biological processes enriched for genes near sex-biased AREs; purple and blue colors represent female-biased and male-biased genes and terms, respectively, for middle and right panels. **c**, Coordinated regulation of ARE activity and gene expression by sex. Left panel**:** ARE activity for the sex-biased AREs in heart samples; right panel: sex-differential signal for the genes closest to the sex-biased AREs in heart; boxes = 25th-75th percentile (that is inter-quartile range; IQR); line = median; whiskers = $1.5 \times$ IQR.

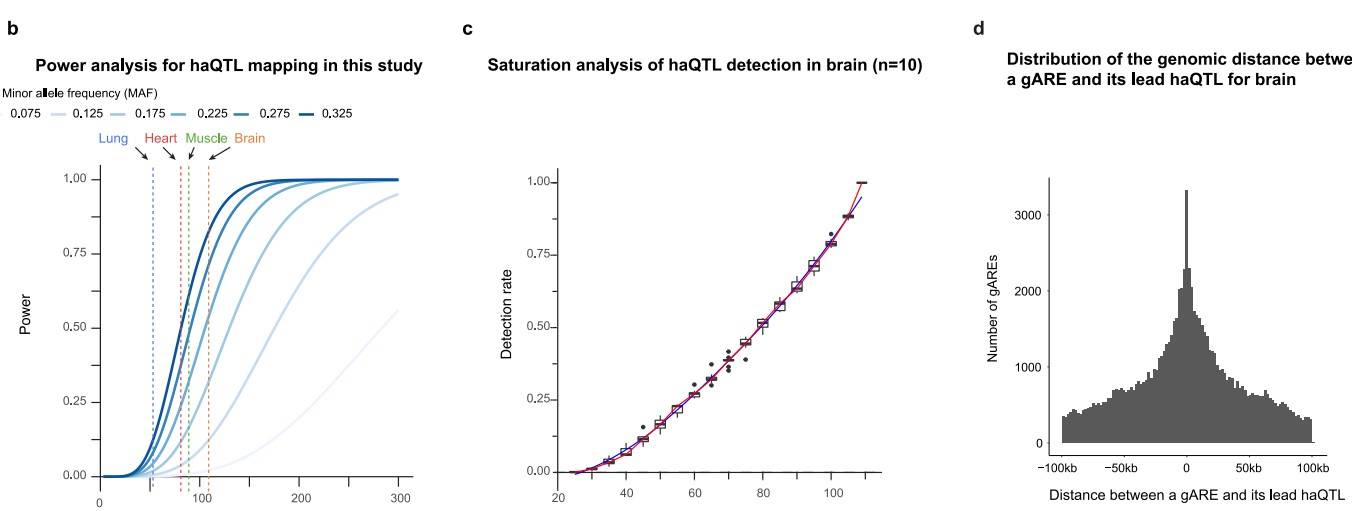

**a**

Comparison between Peer factors and covariates including estimated tissue archetype fraction and known factors in brain

**b**

Power analysis for haQTL mapping in this study

**c**

Saturation analysis of haQTL detection in brain (n=10)

**d**

Distribution of the genomic distance between a gARE and its lead haQTL for brain

**Extended Data Fig. 6 | Identification of haQTLs. a**, Comparison between Peer factors (by column) and covariates (by row) including estimated tissue-archetype fraction and known factors for brain. **b**, Power analysis for haQTL mapping. Colors indicate different minor allele frequencies, and vertical dashed lines denote current sample sizes for each tissue. **c**, Saturation analysis for haQTL detection in the brain based on down-sampling. For each sample size, 10 randomly down-sampling were performed. The x-axis denotes sample sizes after downsampling, while the y-axis denotes the detection rate of the haQTLs from the downsampled data relative to the haQTLs detected from the full data. The boxes show the 25th–75th percentile; the lines show the median; the whiskers show 1.5 × IQR. **d**, Distribution of the genomic distance between a gARE and its lead haQTL for brain.

**a**

**Quantification of haQTL pairwise tissue-sharing based on directionality consistency**

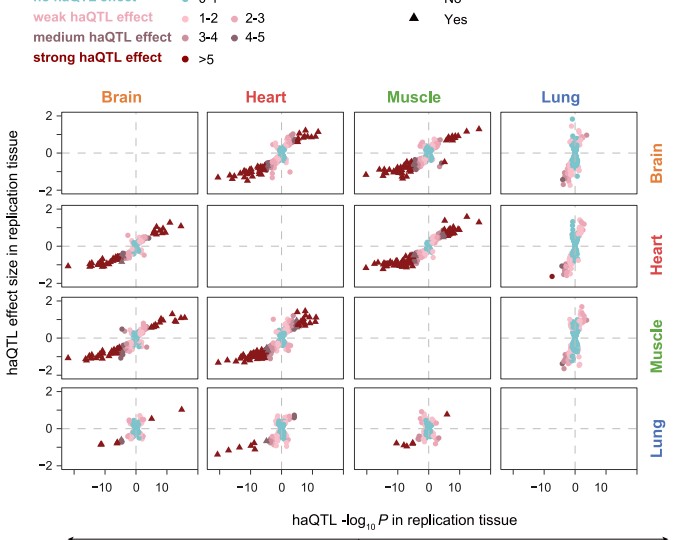

**b**

**Quantification of haQTL pairwise tissue-sharing based on effect size similarity**

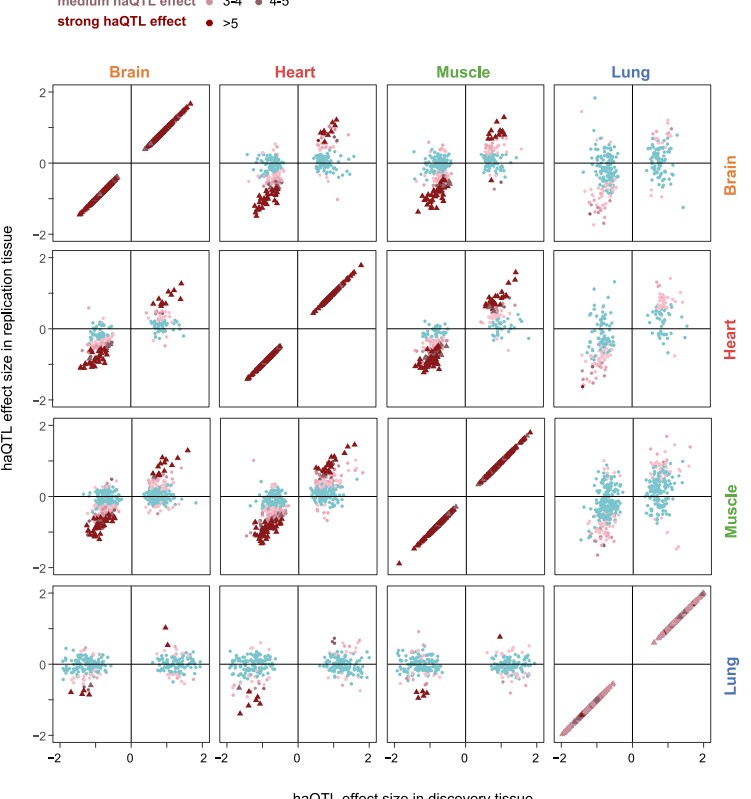

**c**

**haQTL effect size similarity between tissues aggregated for each haQTL effect bin**

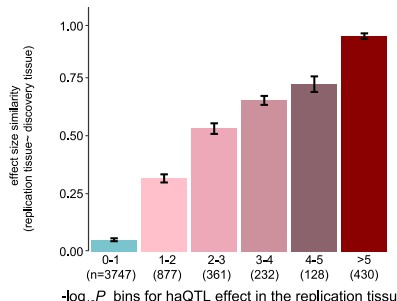

**d**

**Identification of Type-I haQTL-Shared gARE based on P-value in replication tissue**

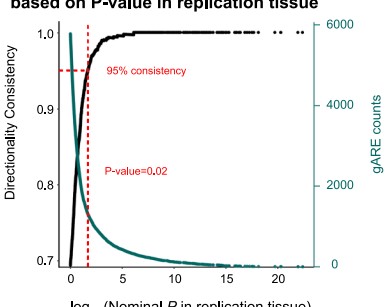

**e**

**gARE type explains eQTL tissue specificty**

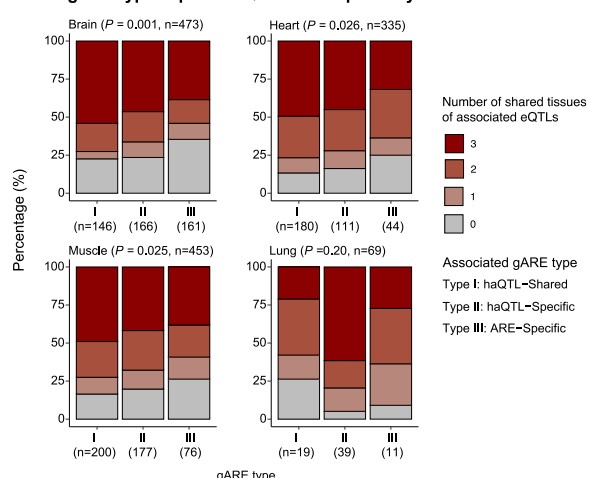

**Extended Data Fig. 7 | See next page for caption.**

**Extended Data Fig. 7 | haQTL tissue-specificity. a**, Quantification of haQTL pairwise tissue-sharing based on directionality consistency. The x-axes show the -$\log_{10}$($P$-value) of haQTLs in the replication tissue, separated by positive effect (right half-plane) and negative effect (left half-plane) in the discovery tissue; the y-axes show the haQTL effect sizes in the replication tissue; haQTLs $P$-values shown in panels **a-d** are all nominal $P$-values based on linear regression (two-sided test). **b**, Quantification of haQTL pairwise tissue-sharing based on similarity of effect size. The x-axes show the haQTL effect sizes in the discovery tissue, and the y-axes show the haQTL effect sizes in the replication tissue. **c**, The effect size similarity, defined as the coefficient of effect size between the replication tissue and discovery tissue, increases as the $P$-value significance increases in the replication tissue; the centers represent the estimated coefficient, and the error bars denote standard errors of the estimation. **d**, Identification of Type-I gAREs (haQTL-Shared) based on the nominal $P$-values in the replication tissue. The black curve shows directionality consistency (the y-axis on the left) of gAREs passing the $P$-value threshold on the x-axis; the green curve shows the count of gAREs (the y-axis on the right) passing the nominal $P$-value threshold on the x-axis; the nominal $P$-value threshold was set to 0.02 in the replication tissue to define a Type-I gARE (haQTL-Shared) between the discovery and replication tissues, which makes the directionality consistency between the two tissues be over 95%. **e**, gARE type explains eQTL tissue specificity. Different panels represent results for each tissue; x-axis represents the different type of gARE with increasing tissue specificity; y-axis represents eQTL tissue-specificity, the number of eQTL-sharing tissues; $P$-values testing the dependence of eQTL tissue-specificity on gARE tissue-specificity (linear regression, two-sided) are shown on top.

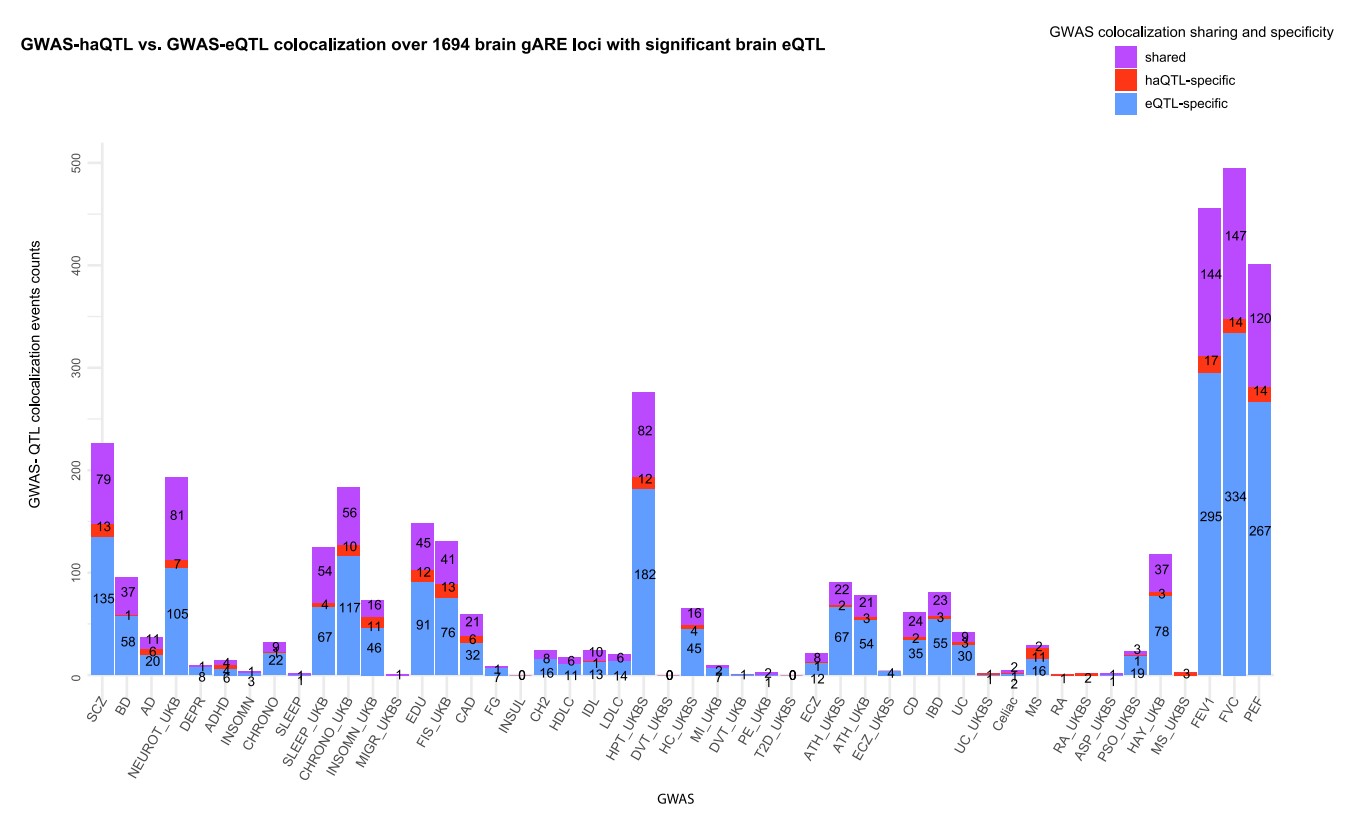

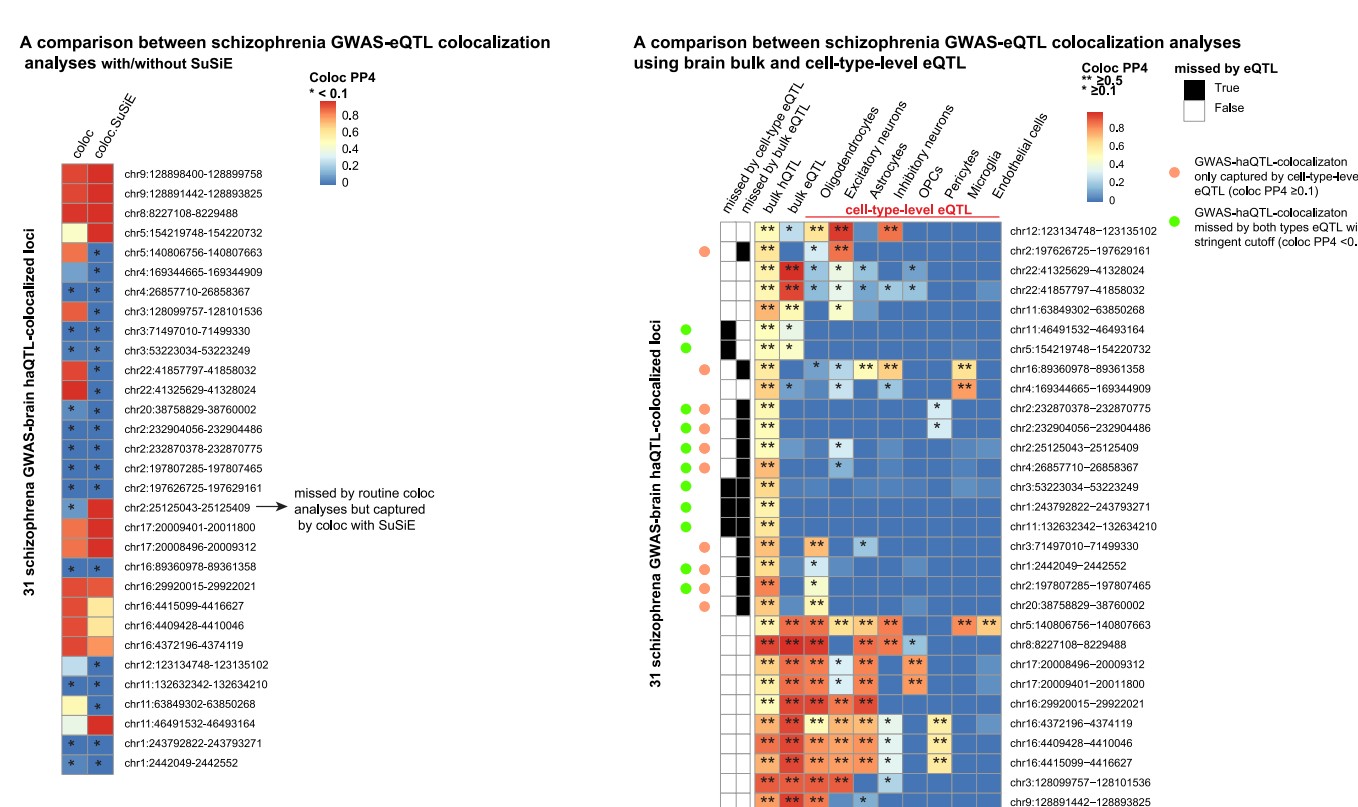

**Extended Data Fig. 8 | See next page for caption.**

**Extended Data Fig. 8 | GWAS-haQTL and -eQTL colocalization. a**, GWAS-haQTL vs. GWAS-eQTL colocalization over 1694 brain gARE loci with significant brain eQTL. The x-axis shows different GWAS traits, and the y-axis denotes the counts of GWAS-haQTL or GWAS-eQTL colocalization events from three types: shared colocalization (in red, both GWAS-eQTL and GWAS-haQTL coloc PP4 ≥ 0.1, at least one of them ≥0.5), haQTL-specific colocalization (in green, GWAS-haQTL coloc PP4 ≥ 0.5 and GWAS-eQTL coloc PP4 < 0.1), and eQTL-specific colocalization (in blue, GWAS-eQTL coloc PP4 ≥ 0.5 and GWAS-haQTL coloc PP4 < 0.1). **b**, A comparison between schizophrenia GWAS-eQTL colocalization analyses with/without SuSiE. The heatmap shows the coloc PP4 for each gARE locus (by row, labeled on the right, hg38 coordinates) with each method (by column, labeled on top). * marks blocks missing colocalization signal (coloc PP4 < 0.1), while arrow points out the only loci missed by coloc and captured by coloc with SuSiE. **c**, A comparison between schizophrenia GWAS-eQTL colocalization analyses using brain bulk and cell-type-level eQTL. The heatmap shows the coloc PP4 for each gARE locus (by row, labeled on the right, hg38 coordinates) based on bulk haQTL, bulk eQTL, and eQTL from eight cell types in the brain (by column). * marks blocks with weak colocalization signal (coloc PP4 ≥ 0.1), while ** marks blocks with strong colocalization signal (coloc PP4 ≥ 0.5). Orange dot on the right marks the loci only captured by cell-type-level eQTL (coloc PP4 ≥ 0.1), while green dot marks the loci missed by both types of eQTLs with stringent cutoff (coloc PP4 < 0.5). Three gARE loci with strong GWAS-haQTL colocalization signals (coloc PP4 ≥ 0.5) and missed by both types of eQTLs even at the permissive cutoff (coloc PP4 < 0.1) are shown in red box.

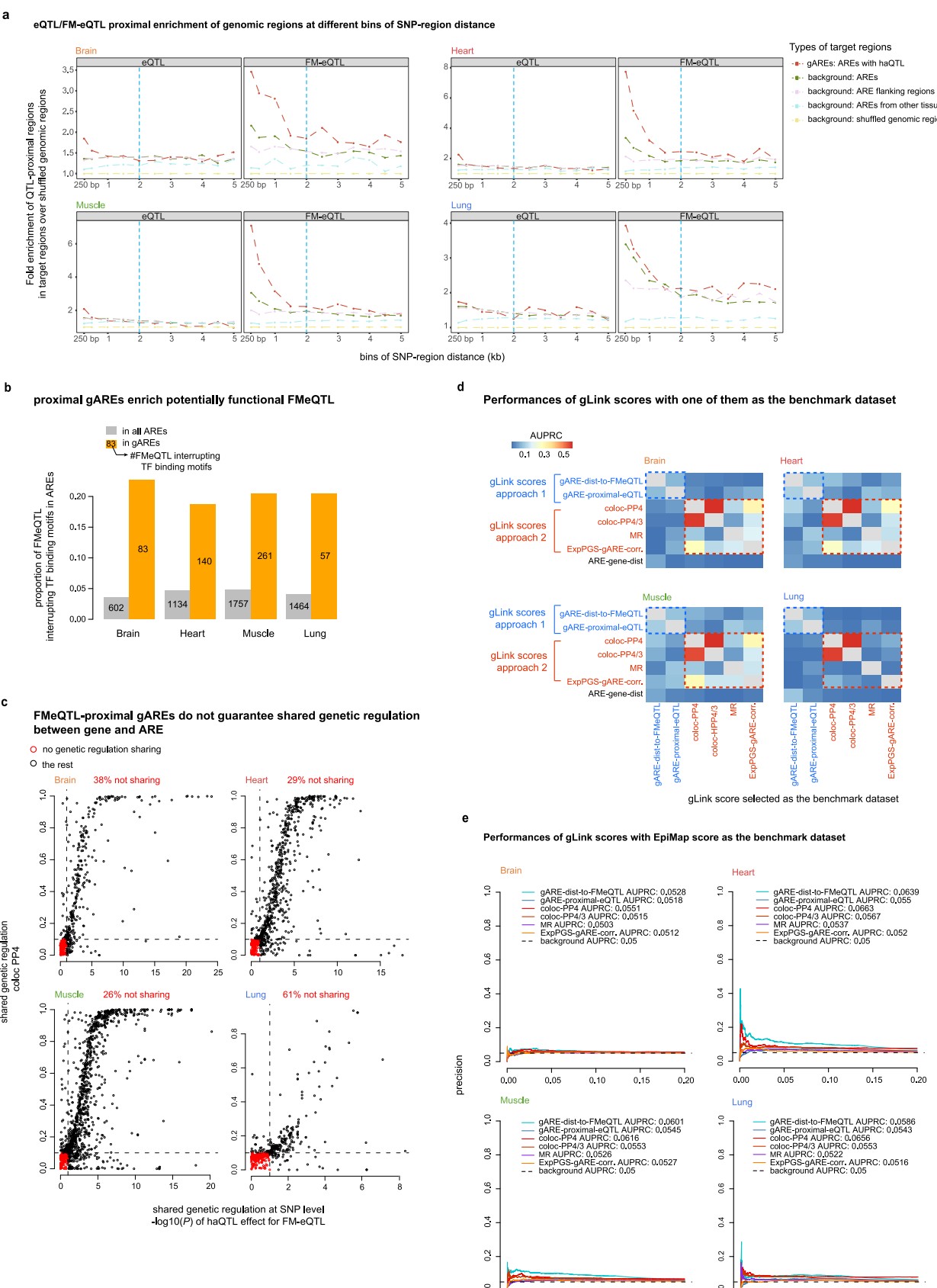

**Extended Data Fig. 9 | See next page for caption.**

**Extended Data Fig. 9 | Properties of gLink scores. a**, eQTL/FM-eQTL proximal enrichment of genomic regions at different bins of SNP-region distance. The plots show the fold enrichment of QTL-proximal regions in target regions (in different color) over shuffled genomic regions for eQTL (left) and FM-eQTL (right) for each tissue. The y-axis shows the enrichment, while the x-axis shows different SNP-region bins (0–250 bp, 250bp-500bp, 0.5–1, 1–1.5, 1.5–2, 2–2.5, 2.5–3, 3–3.5, 3.5–4, 4–4.5, 4.5–5 kb). The blue dashed line marks 2 kb which we chose as the cutoff to define the proximity. **b**, Proximal gAREs enrich FMeQTLs interrupting TF binding sites. The y-axis shows the proportion of FMeQTLs that interrupt TF binding sites from Extended Data Fig. 2c; orange bars represent the FMeQTLs located in gAREs, and gray bars represent FMeQTLs in all AREs. **c**, FMeQTL-proximal gAREs do not guarantee shared genetic regulation between gene and ARE. Each point: gARE-gene pair based on *gARE-dist-to-FMeQTL* score

(distance cutoff of 2 kb); x-axes denote the shared genetic regulation at the SNP-level (nominal *P* for haQTL, linear regression, two-sided); y-axes denote the shared genetic regulation at the locus-level; Red circles indicate pairs without evidence of shared genetic regulation at both the locus and SNP levels. Percentage of FmeQTL-proximal gAREs without shared genetic regulation in each tissue shown at the top of each graph. **d**, Performances of gLink scores with one of them as the benchmark dataset (AUPRC). The heatmaps show AUPRC for each gLink score (by row) with one of scores as the benchmark dataset (by column) for each tissue; red dashed boxes indicate results for gLink scores from approach 2, showing higher consistency between these scores. **e**, Performance of gLink scores with EpiMap score as the benchmark dataset. We showed PRC of gLink scores for each tissue, which is barely higher than that of background.

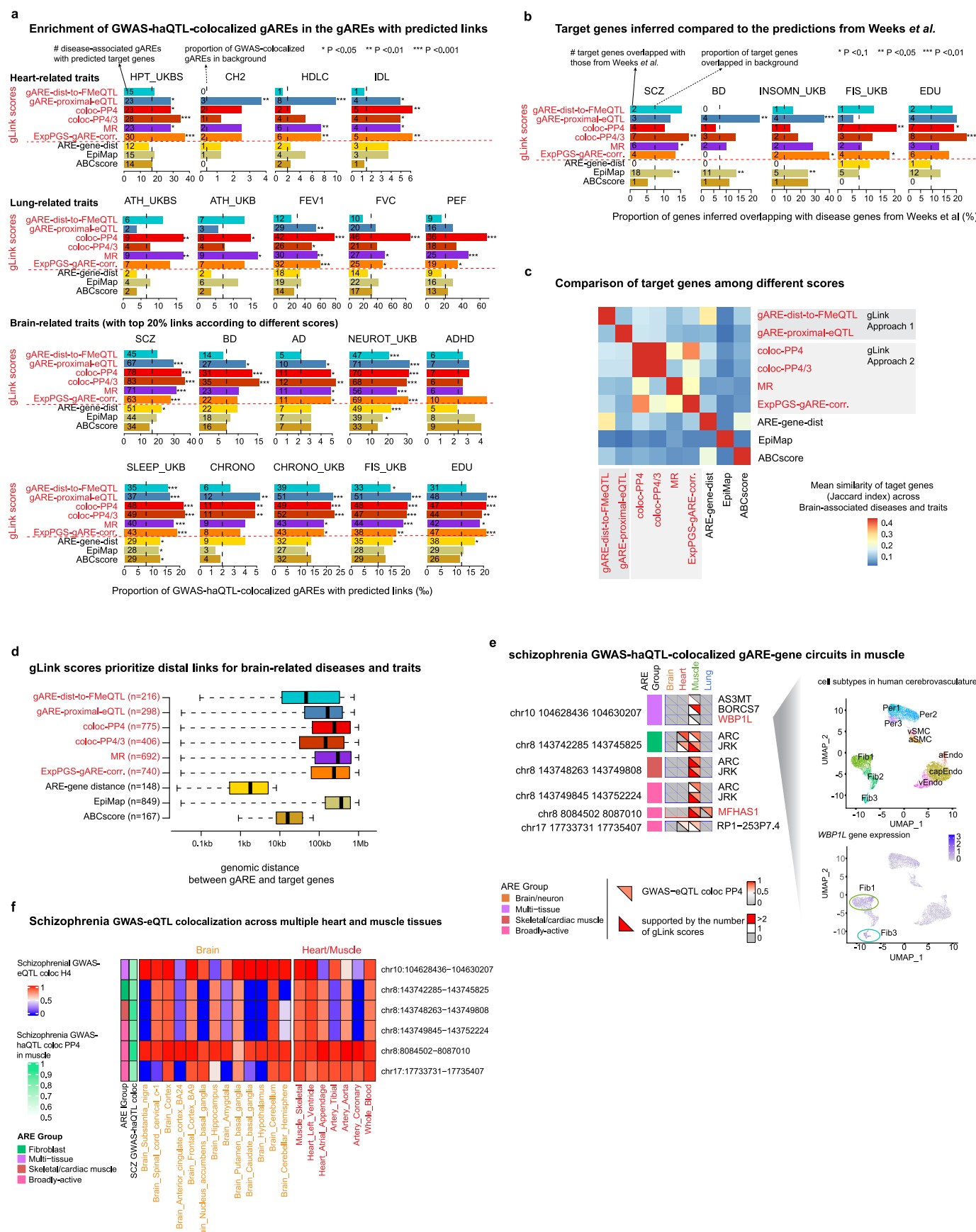

**Extended Data Fig. 10 | See next page for caption.**

**Extended Data Fig. 10 | gLink scores prioritize gARE-gene circuits for diseases and traits. a**, Enrichment of GWAS−haQTL-colocalized gAREs in the gAREs with predicted links for heart- and lung-related traits. Figure format as in Fig. 6a; * shows the significance levels (one-sided proportion test); HPT, hypertension; CH2, MAGNETIC_CH2.DB.ratio; HDLC, MAGNETIC_HDL.C; IDL, MAGNETIC_IDL. TG; ATH, asthma; FEV1, volume that has been exhaled at the end of the first second of forced expiration; FVC, Forced Vital Capacity; PEF, Peak expiratory flow; UKB, UK biobank; UKBS, self reported traits from UK biobank. **b**, Target genes inferred compared to the predictions from Weeks et al. The x-axes denote the percentage of genes inferred from each approach (by row) overlapping with the disease genes from a previous report for each disease (by panel); * shows the significance levels (two-sided Fisher's exact test). INSOMN, insomnia. **c**, Comparison between target genes from different linking scores. The heatmap shows the mean of the similarity of target genes (Jaccard index) across 10 brain-related traits in Fig. 6a for each pair of linking scores. **d**, Distribution of the distance between a GWAS−haQTL-colocalized gARE and its predicted target gene by each linking score for the 10 brain-related traits in Fig. 6a. Boxes = 25th–75th percentile (that is inter-quartile range; IQR); line = median; whiskers = 1.5 IQR; number of gARE−gene pairs shown. **e**, Schizophrenia GWAS−haQTL-colocalized gARE-gene circuits only in muscle or heart. Left panel: The heatmap shows the genetic evidence of association between target gene and schizophrenia for each gARE−gene circuit (by row) in each tissue (by column); genomic position of ARE and ARE group shown on the left; for each cell, upper triangle shows evidence based on GWAS−eQTL colocalization (PP4), and lower triangle shows number of gLink scores that connect GWAS−haQTL-colocalized gARE to the same gene; genes in red text on right side of heatmap identified as fibroblast subtype marker genes from a brain vasculature sc-RNA-seq study. Right panel: upper, UMAP result of sc-RNA-seq profiles with cell subtype labeled from a brain vasculature study, and lower, *WBP1L* expression level marked in the UMAP. **f**, Schizophrenia GWAS−eQTL colocalization for gAREs loci from panel **e**. The heatmap shows GWAS−eQTL PP4 across 13 brain-related tissues and 7 muscle/heart-related tissues (by column) for each gARE (by row); ARE group and GWAS−haQTL colocalization are annotated on left.

## Reporting Summary

## Statistics

For all statistical analyses, confirm that the following items are present in the figure legend, table legend, main text, or Methods section.

| n/a | Confirmed | |
|---|---|---|
| ☐ | ☒ | The exact sample size (*n*) for each experimental group/condition, given as a discrete number and unit of measurement |
| ☐ | ☒ | A statement on whether measurements were taken from distinct samples or whether the same sample was measured repeatedly |
| ☐ | ☒ | The statistical test(s) used AND whether they are one- or two-sided *Only common tests should be described solely by name; describe more complex techniques in the Methods section.* |
| ☐ | ☒ | A description of all covariates tested |
| ☐ | ☒ | A description of any assumptions or corrections, such as tests of normality and adjustment for multiple comparisons |
| ☐ | ☒ | A full description of the statistical parameters including central tendency (e.g. means) or other basic estimates (e.g. regression coefficient) AND variation (e.g. standard deviation) or associated estimates of uncertainty (e.g. confidence intervals) |
| ☐ | ☒ | For null hypothesis testing, the test statistic (e.g. *F*, *t*, *r*) with confidence intervals, effect sizes, degrees of freedom and *P* value noted *Give P values as exact values whenever suitable.* |
| ☒ | ☐ | For Bayesian analysis, information on the choice of priors and Markov chain Monte Carlo settings |
| ☒ | ☐ | For hierarchical and complex designs, identification of the appropriate level for tests and full reporting of outcomes |
| ☐ | ☒ | Estimates of effect sizes (e.g. Cohen's *d*, Pearson's *r*), indicating how they were calculated |

*Our web collection on statistics for biologists contains articles on many of the points above.*

## Software and code

Policy information about availability of computer code

| | |
|---|---|
| Data collection | The H3K27ac ChIP-seq profiling sequencing data was generated at the Genomics Platform and the Broad Technology Labs (BTL) using the Illumina HiSeq 2500 sequencer. Raw sequencing data was aligned and processed using bwa (v0.5.9). |
| Data analysis | We used samtools (v1.3.1), MACS2 (v2.1.1) for peak calling, bedtools (v2.26.0) for peak processing, FastQTL (v2.184) for QTL identification, LDSC (v1.0.0) for GWAS inheritability analysis, HOMER (v4.11.1) for motif analysis. We mainly used R (v3.6.0) unless specified, and used following R packages in the analysis: Seurat (v3.2.1), flexclust (v1.4-0), annotatr (v1.6.0), rGREAT(v1.2.1), NMF (v0.21.0), limma (v3.36.5), sva (v3.28.0), Peer (v1.0), MR (v0.4.1), coloc (v3.2-1), susieR (v0.12.27, R-4.0), Sushi (v1.18.0), powereQTL (v0.3.4), PRROC (v1.3.1), ComplexHeatmap (v2.4.3, R-4.0) . |

For manuscripts utilizing custom algorithms or software that are central to the research but not yet described in published literature, software must be made available to editors and reviewers. We strongly encourage code deposition in a community repository (e.g. GitHub). See the Nature Portfolio guidelines for submitting code & software for further information.

## Data

Policy information about availability of data

All manuscripts must include a data availability statement. This statement should provide the following information, where applicable:

- Accession codes, unique identifiers, or web links for publicly available datasets
- A description of any restrictions on data availability
- For clinical datasets or third party data, please ensure that the statement adheres to our policy

Supplementary info and datasets are linked from http://compbio.mit.edu/eGTEx-H3K27ac/. H3K27ac ChIP-seq profiles from this study are available on dbGaP with accession number phs000424.v8.p2. Additionally, the data can be accessed via AnVIL with authentication: https://anvil.terra.bio/#workspaces/anvil-datastorage/ AnVIL_GTEx_V8_hg38. Since the raw sequencing data with genetic information are protected, application and authentication are needed before accessing the data. All non-protected data of H3K27ac ChIP-seq can be visualized via the GTEx Portal (www.gtexportal.org) as part of eGTEx v8.

## Human research participants

Policy information about studies involving human research participants and Sex and Gender in Research.

| | |
|---|---|
| Reporting on sex and gender | We added sex and age in the supplementary table 1 as part of meta information |
| Population characteristics | This study does not involved human research participants, since samples are postmortem from the GTEx cohort and deidentified. |
| Recruitment | *Describe how participants were recruited. Outline any potential self-selection bias or other biases that may be present and how these are likely to impact results.* |
| Ethics oversight | Massachusetts Institute of Technology Committee on the Use of Humans as Experimental Subjects (COUHES) approved this study does not involve human subjects as defined by federal regulations. |

Note that full information on the approval of the study protocol must also be provided in the manuscript.

# Field-specific reporting

Please select the one below that is the best fit for your research. If you are not sure, read the appropriate sections before making your selection.

☒ Life sciences  ☐ Behavioural & social sciences  ☐ Ecological, evolutionary & environmental sciences

For a reference copy of the document with all sections, see nature.com/documents/nr-reporting-summary-flat.pdf

# Life sciences study design

All studies must disclose on these points even when the disclosure is negative.

| | |
|---|---|
| Sample size | Due to the limited availability of human primary tissue samples, we initially performed genome-wide H3K27ac profiling in 517 human samples across brain, heart, muscle, and lung, with more than 100 samples for each tissue. |
| Data exclusions | After stringent quality control based on both standard ENCODE pipeline and tissue-specificity criterion, 387 profiles were kept for analyses. |
| Replication | Since very few data are available for haQTLs in brain, muscle, heart, and lung, we did not carry out replication analysis. Instead, we checked the sharing of haQTL effect among our four tissues, and observed consistency between tissues. |
| Randomization | During haQTL mapping, individuals were grouped based on genotypes, and covariates such as sex, age, and inferred latent factors were taken into consideration. |
| Blinding | Individuals are grouped based on genotypes, which is blind to investigators. |

# Reporting for specific materials, systems and methods

We require information from authors about some types of materials, experimental systems and methods used in many studies. Here, indicate whether each material, system or method listed is relevant to your study. If you are not sure if a list item applies to your research, read the appropriate section before selecting a response.

## Materials & experimental systems

| n/a | Involved in the study |
|-----|----------------------|
| ☐ | ☒ Antibodies |
| ☒ | ☐ Eukaryotic cell lines |
| ☒ | ☐ Palaeontology and archaeology |
| ☒ | ☐ Animals and other organisms |
| ☒ | ☐ Clinical data |
| ☒ | ☐ Dual use research of concern |

## Methods

| n/a | Involved in the study |
|-----|----------------------|
| ☐ | ☒ ChIP-seq |
| ☒ | ☐ Flow cytometry |
| ☒ | ☐ MRI-based neuroimaging |

## Antibodies

| Antibodies used | Rabbit monoclonal antibody to the H3K27ac epitope (Cell Signaling Technology, Cat#8173), with 1:100 dilution. |
|---|---|
| Validation | This antibody has been validated by the manufacturer using SimpleChIP® Enzymatic Chromatin IP Kits. It was also validated in ENOCDE project (https://www.encodeproject.org/antibodies/ENCAB502OHI/), and has been cited in more than 300 other papers. |

## ChIP-seq

### Data deposition

☒ Confirm that both raw and final processed data have been deposited in a public database such as GEO.

☒ Confirm that you have deposited or provided access to graph files (e.g. BED files) for the called peaks.

| Data access links *May remain private before publication.* | https://doi.org/10.5281/zenodo.7992724 |
|---|---|
| Files in database submission | H3K27ac ChIP-seq profiles from this study are available on dbGaP with accession number phs000424.v8.p2. Additionally, the data can be accessed via AnVIL with authentication: https://anvil.terra.bio/#workspaces/anvil-datastorage/ AnVIL_GTEx_V8_hg38; All non-protected data of H3K27ac ChIP-seq can be visualized via the GTEx Portal (www.gtexportal.org) as part of eGTEx v8. Proccessed data could be found on Zenodo (https://doi.org/10.5281/ zenodo.7992724). |
| Genome browser session (e.g. UCSC) | https://gtexportal.org/home/. Check IGV browser for detail. |

### Methodology

| Replicates | We do not have biological replicates for each individual. |
|---|---|
| Sequencing depth | pair-end data, reads length 101bp. Please refer to supplementary Table 1 for the total reads of each of 387 experiments. |
| Antibodies | monoclonal antibody to the H3K27ac epitope (Cell Signaling Technology, #8173) |
| Peak calling parameters | macs2 callpeak -t ${REP1_TA_FILE}.tagAlign.gz -f BED -n ${PEAK_OUTPUT_DIR}/${CHIP_TA_PREFIX} -g ${GENOMESIZE} -p 1e-2 -- nomodel --shift 0 --extsize ${FRAGLEN} --keep-dup all -B --SPMR |
| Data quality | Samples were QCed based on three metrics: (1) Relative Strand Cross-correlation coefficient (RSC), based on the (shifted) agreement between forward-strand and reverse-strand reads, providing a measure of signal-to-noise enrichment ratio; (2) sequencing depth, counted as total reads; (3) H3K27ac signal correlation between our samples and the samples from Roadmap epigenomes based on tissue-specific AREs identified from Roadmap. Please refer to Supplementary Table 1 for details, including number of peaks for each sample. |
| Software | Raw sequencing data was aligned and processed using bwa and picard, followed by AQUAS pipeline (https://github.com/kundajelab/ chipseq_pipeline) |

