## [Peer Review File · Nature Genetics]

Peer Review Information

Manuscript Title: Multi-tissue H3K27ac profiling of GTEx samples links epigenomic variation to disease

Corresponding author name(s): Professor Manolis Kellis

Reviewer Comments & Decisions:

Decision Letter, initial version:

29th Aug 2022

Dear Manolis,

Your Article, entitled "Multi-tissue epigenomic variation atlas across 387 GTEx samples interprets disease genetics in brain, heart, muscle, and lung", has now been seen by 4 referees. I apologize for the slow review process.

You will see from their comments below that while they find your work of interest, some important points are raised.

Reviewer #1 is impressed by the resource value of the dataset and by the analyses. Their main suggestion is about extending the MR analysis, and modifying some claims.

Reviewer #2 also says that dataset constitutes an important resource but notes that more details about data and code availability are needed. They think you should also provide some clarification on the gLink analysis.

Reviewer #3 equally highlights the value of the dataset. Their suggestions are mainly aimed at improving/extended the current analyses. There are no major concerns.

Reviewer #4 echoes what the other reviewers say about the dataset. They have some minor critiques related to the text, and the level of conceptual advance. However, the latter point is not a major editorial concern on this occasion.

We are interested in the possibility of publishing your study in Nature Genetics, but would like to consider your response to these concerns in the form of a revised manuscript before we make a final decision on publication.

We therefore invite you to revise your manuscript taking into account all reviewer and editor comments. Please highlight all changes in the manuscript text file. At this stage we will need you to upload a copy of the manuscript in MS Word .docx or similar editable format.

We are committed to providing a fair and constructive peer-review process. Do not hesitate to contact me if there are specific requests from the reviewers that you believe are technically impossible or unlikely to yield a meaningful outcome.

*2) If you have not done so already please begin to revise your manuscript so that it conforms to our Article format instructions, available [here](http://www.nature.com/ng/authors/article_types/index.html). Refer also to any guidelines provided in this letter.

[redacted]

We hope to receive your revised manuscript within eight weeks. If you cannot send it within this time, please let us know.

Sincerely,

Tiago

Tiago Faial, PhD
Chief Editor
Nature Genetics
<https://orcid.org/0000-0003-0864-1200>

Reviewers' Comments:

Reviewer #1:

Remarks to the Author:

The authors describe how genetic variation is affecting active regulatory elements (ARE) in brain, heart, muscle and lung. They observe a large number of genetic variants that affect AREs that are shared between tissues and that are specific to certain (combinations of) tissues. The authors carefully study this sharing: they first discovery gAREs (genetically affected AREs) and then attempt replication in another tissue using liberal threshold, and ascertaining allelic direction concordance, observing, even for the weakly replicating gAREs (replication $1e-3 < P < 0.1$) that 82% share the same allelic direction.

Subsequently the authors ascertain to what extent a diverse collection of traits, on which GWAS has been conducted, are showing enriched association signal in regions that have been identified as gARE. Indeed, the authors observe this to be the case, and find enrichments for the expected trait - tissue combinations.

Then the authors use colocalization to identify 456 GWAS-haQTL gAREs for 43 traits. What is exciting to observe is that for several of these co-localizing gAREs no cis-eQTLs are found, underscoring the relevance and importance of studying H3K27 acetylation in different tissues of a large number of samples.

I am impressed by the dataset, and the analyses that are conducted, and I believe this paper and resource is a very important contribution to the field.

The last part of the paper, where eQTLs and gAREs are integrated, to me is however somewhat speculative. For instance, The authors state in the discussion: "gLink scores outperform gene-ARE distance, EpiMap and ABC scores on linking GWAS-haQTL-colocalized gAREs to target genes.". For me it is not immediately clear that gLink indeed is better than ABC: for instance, figure 5c does not seem to include results on ABC. In my opinion, the paper would actually improve further if this section

would be rephrased somewhat, e.g. as a novel way to generate interesting hypotheses for follow-up research.

Major question:

Colocalization is used to ascertain overlap between GWAS traits and gAREs. In the field, colocalization is often combined with Mendelian randomisation (MR) analyses. The authors report on 456 GWAS-haQTL gAREs for 43 traits. Can the authors also perform MR to study whether there is evidence to suggest these effects on diseases are likely mediated through these AREs? The authors already use MR in a later section of the paper, so I suppose this is something they could do here as well I believe it would be valuable to use it here as well.

The authors state that for schizophrenia there are 20 co-localized GREs for which no eQTLs are known. Could it be that the GTEx sample-size is a limiting factor, and some of these GRE SNPs actually do give a small but significant or cell-type specific cis-eQTL effect in other larger bulk brain datasets or single-cell brain datasets? E.g. one way to address this question might be to use the 'Bryois et al, Nature Neuroscience 2022' paper that studied single-nuclei or the 'de Klein et al, BioRxiv 2021' preprint that studied >2,000 bulk samples.

The authors study whether known GWAS variants are in LD with variants that affect GREs that are sex-biased. The authors highlight one example: "For example, SNP rs7090118 is a subthreshold GWAS hit ($P=2.3 \times 10^{-6}$) for hypertension in moderate LD ($r^2=0.54$) with the lead SNP (rs10764326, $P=1.5 \times 10^{-11}$).". I must admit that I am a bit reserved on this example, because of the sub-threshold significance and the limited LD. I believe the paper on itself is already very valuable and does not necessarily need this section. Maybe the authors could consider removing this paragraph from the paper?

Minor questions:

The authors state: "We found that our H3K27ac profiles from the same tissue clustered together, and importantly co-clustered with tissue-matched samples from Roadmap/ENCODE/GGR (Fig. 1b), including 14 of 14 previously-profiled brain samples, 13 of 14 heart samples, 6 of 9 muscle samples, 2 of 3 healthy lung samples, and 18 of 19 lung tumor samples.". Can the authors speculate what is the reason 6 samples do not cluster as expected?

There seems to be a discrepancy between the number of GWAS studies used in the LD score regression part and the subsequent colocalization part. Can the authors explain what is the reason this number is different? Can this be made consistent?

Can the authors correct a typo in the discussion? "First, although we profied over".

Reviewer #2:

Remarks to the Author:

This study produces a valuable resource of H3K27ac ChIP-seq datasets in 387 brain, heart, muscle, and lung tissue samples from the GTEx cohort. The authors systematically identify modules of H3K27ac peaks, call H3K27ac QTLs, and colocalize these QTLs with GWAS variants. The authors also

develop a “gLink” analysis that uses QTL evidence to link H3K27ac peaks to target genes, and apply this to interpret GWAS variants.

Overall, I am enthusiastic about the resource, which represents one of the largest H3K27ac ChIP-seq QTL studies and would be generally useful for the community. Improved availability or documentation of data and metadata is essential to make this resource useful for the genomics community. Clearer justification and explanation of the gLink analysis would also improve the study.

Major Comments:

1. Data availability: Availability of data, metadata, and processed intermediate analysis files is essential to enable use of this resource by the community.

a. Data: The data is deposited on dbGap, and so I was not able to confirm availability. Please report exactly what files (raw data, processed data) are available there.

b. Metadata: Is there other eGTEx data from the same/similar samples that can be integrated? e.g. ATAC-seq would be particularly valuable for identifying regulatory elements at higher resolution. This paper should at least provide the needed metadata to integrate and explain how the samples used here overlap with other eGTEx efforts. Presumably this metadata could be reported without access control.

c. Intermediate analysis files: The supplementary tables have only a small number of the intermediate analyses, and appear to include only significant hits (e.g., GWAS-QTL colocalizations about some threshold). Please instead provide processed data files with full intermediate analysis results, modified as needed to enable open access rather than access control. Many similar files are available open access from the GTEx portal for the main eQTL component of the project and so I do not believe that there would be access restrictions on similar files from this study.

(<https://gtexportal.org/home/datasets>) Examples of intermediate analysis files that would be useful:

- i. Full QTL results for all tested variant-peak pairs
- ii. Data matrices used for each heatmap figure
- iii. List of peaks and their counts + normalized counts across all samples
- iv. List of peaks and their mean normalized counts across tissues
- v. Full colocalization results

2. Code availability: The code is not available at the link provided

3. The authors should conduct a power analysis to help readers understand this datasets, and to inform future studies. This should include (1) reporting power for each tested variant-ARE pair, the PEER model beta, the estimated fold-change effect in H3K27ac counts; and (2) summarizing power across the study

4. The analysis about GWAS signals that colocalize with haQTLs but not eQTLs (20 of 59 SCZ loci) is potentially very interesting. The authors propose a 2-step model in Fig 4f. Could the authors address to what extent any of the following effects could provide alternative explanations for this category of GWAS signals:

- a. Lack of power in the eQTL analysis or colocalization analysis
- b. Presence of multiple independent eQTL signals that confound analysis with coloc
- c. Ability of haQTLs to pick up QTL signals at cell-type specific enhancers for rarer cell types in a tissue (e.g. microglia in brain), whereas eQTL signals would be washed out because the gene is expressed in multiple cell types in the tissue

5. Could the authors simplify the presentation of the gLink analysis? I am a bit lost, and have various questions about the analysis:
- Could the authors explain the utility of the gLink analysis? It seems to me that it will necessarily be a very incomplete list of enhancers for any given target gene, because it depends on there being eQTL variants for the gene (which is underpowered, and where natural genetic variation might not exist). It certainly is helpful to annotate fine-mapped eQTLs that directly overlap gARE peaks — thereby implicating the peak in regulation of the target gene ... Beyond this simple analysis, I am not understanding the utility of the more complex analysis presented
 - There are 6 different gLink scores, but then on page 10 line 1 what is the “unified score system for gLink scores”?
 - Page 8 — 2kb distance between eQTL and gARE is a fairly generous window for linking gAREs to variants and target genes. eQTL variants and GWAS variants much more highly enriched in chromatin accessible regions than in H3K27ac peaks, which tend to include large sequences adjacent to chromatin accessible regions. If using H3K27ac peaks directly for variant overlaps, I would suggest a smaller distance threshold, like 250bp from the peak which would capture a chromatin accessible site adjacent to the H3K27ac peak. Alternatively, it could be even better to infer the location of the chromatin accessible site through analysis of dips in the H3K27ac signal, and use those regions for variant overlap.
 - Fig 5c “Validation of gLink scores with ABC score” should probably be better titled “Comparison of gLink scores to ABC score”
 - Fig 6a. dotted line— How is the background defined?
 - Fig 6a. dotted line— is it correct that this dotted line defines the background for one of these methods, but not all? E.g. if you wanted to calculate an enrichment over background for each method, does each method need its own background calculation or is the background the same for all methods?
 - Is the validation analysis in Fig 6b circular, because eQTL signals were used to define the gLink scores?
 - For the comparison of gLink scores vs EpiMap and ABC, could the authors explain why these scores perform better (at least, have better recall in Fig 6a)? Is it because of the threshold on the scores chosen? Is it because the gLink scores are based on eQTL signals in a tissue that can be cell-type specific, whereas cell-type specific signals are harder to detect in EpiMap and ABC?
 - For the gLink-GWAS analysis on page 10, given that enhancers are linked to genes based on fine-mapped eQTLs, does gLink analysis give something substantially different than simply doing eQTL colocalization? The fact that 10 of 12 gLink predictions are also colocalized eQTL-GWAS signals would suggest this might be the case. What about the reverse? Does colocalized eQTL-GWAS signals find many more target genes than gLink predictions?

Minor Comments:

- Please add page numbers for review
- Please report the range of unique reads per ChIP sample at top of page 2
- “found primarily at active enhancers and to a lower degree at active promoters”. I believe it would be more correct to say it is found at both active enhancers and promoters.
- “our four tissues sampled here might not be their primarily tissues of action”. I am not sure what

this means, with regards to an enhancer module. Please clarify. Is another possibility that they reflect enhancers active in certain cell types e.g. vascular or immune that are present in many tissues?

5. Fig 4c — The p-values for SCZ GWAS look to be not genome-wide significant. Could the authors clarify how they chose sub-threshold significant SCZ GWAS loci to include in the analysis?

6. Page 6 — please specify in the main text the PP4 probability threshold used to call “colocalization”

Reviewer #3:

Remarks to the Author:

Hou et al. present results from an enhancing GTEx (eGTEx) study that profiles epigenomic variation (H3K27ac) in 387 brain, heart, muscle, and lung samples from GTEx. They identify 282K active regulatory elements (AREs) and characterize their tissue-specific activity patterns. They then proceed to show some proof-of-principle analyses to illustrate the utility of these AREs in understanding the molecular basis of complex traits, including haQTL analyses, GWAS-haQTL colocalization analyses, gene-ARE linking scores to prioritize disease target genes. Overall, the paper is clearly written, the analyses are well described, detailed and use mostly well-established analytical tools from the GTEx and the Roadmap Epigenomics projects. The paper provides a complementary data resource to the eQTL data commonly used in follow-up GWAS studies that geneticists may find very helpful. A weakness of the paper is that some of the proof-of-principle analyses were underdeveloped and could be improved. Detailed comments are below.

1. Can the authors provide a measure of false discovery rate among the detected AREs? In particular, among the newly discovered AREs can we get an estimate of FPR?
2. The correlations between the H3K27ac profiles in this study and previous studies (Sup Fig 1) seem low, can the authors provide additional comments on this?
3. The authors perform NMF to remove unwanted variation from the H3K27ac profiles. Why not perform this step from the beginning?
4. On page 5: the authors state: “We also distinguish Type-I “haQTL-Shared” and Type-II “haQTL-Specific” gAREs based on the nominal P-value threshold ($P < 0.02$) in the replication tissues”. How is this threshold for nominal significance (0.02) selected? Seems arbitrary.
5. Overall, I think it would be helpful to have a more thorough comparison of eQTLs and haQTLs for example in terms of co-localization results. Although it is clear that haQTLs can provide additional information on top of eQTLs and partially address the missing regulation problem with eQTLs, it is not clear from these comparisons how useful eQTLs and haQTLs are on their own. For example, how many GWAS-eQTL colocalizations are missed by GWAS-haQTL colocalizations?
6. Although the two-step regulation may be consistent with the observed data, it does not in any case imply that is the only true underlying model ... this part needs to be toned down.
7. I find the part on the gLINK score underdeveloped. I find the comparisons among the different scores not convincing because the scores being used as benchmark are in reality not that accurate.

The authors should identify better benchmarks. For example, previous papers linking loci/variants to genes might be helpful here (PMID: 35668300; PMID: 35147782, PMID: 34711957).

8. The authors show a comparison of gLINK scores with other scores e.g. EpiMap and ABC score to GWAS- eQTL colocalization but it is hard to say whether gLINK scores perform better simply because (as the authors acknowledge) they “tend to capture gARE-proximal genetic signals associated with gene expression”.

9. Also related to the gLINK scores, there are six different scores being proposed. Each of them is pretty simple. Is there a way to combine them as one score? since given the issues mentioned at the previous point, it is not clear which score one can use.

Reviewer #4:

Remarks to the Author:

Hou et al. profile histone acetylation marks in 387 GTEx samples covering brain, heart, muscle, and lung. They identify peaks termed active regulatory elements (ARE), compute their association with genetic variation and finally attempt to explain disease-associated loci in 78 GWAS, identify tissues of action and driver SNPs. They also propose a method to link active regulatory elements to genes.

The 387 ChIP-seq data paired with WGS and RNA-seq already available for the same samples represent an important resource for the scientific community. However, the data does not seem to be yet available on ANVIL. Publication should be contingent on their availability.

The figures are very helpful and the analysis is sound.

I found the manuscript hard to read with details that get in the way of the narrative. Some sections could be simplified (for example the explanation of glink method), the main idea explained up front and provide details later.

The application of the resource did not seem very compelling compared to what was published before by ENCODE and Roadmap epigenome projects. What did we learn with this new data that we didn't already know before?

The authors combined analysis with ENCODE and Roadmap data which facilitates a more interpretable grouping of the clusters. It is also reassuring to see that the new data falls within expected clusters.

Below are some additional comments to the authors:

- In my opinion, defining new types of shared or tissue-specific QTLs (type I, II, and III) adds unnecessary cognitive load to the reader.
- Calling alternative causal SNPs based on the haQTL could be misleading. If the top QTL and the top GWAS variants are different, the more obvious conclusion would be that they are not colocalized, i.e. other mechanism needs to be found rather than claiming that the top GWAS association was “wrong”.
- “CORO7 and NMRAL1 proteins regulate vesicle transport and cellular response to redox changes respectively. The observation that both of these genes link to the same GWAS-haQTL-colocalized gARE indicates a coordinated dysregulation of both processes in the schizophrenia etiology” This conclusion

seems a bit speculative given the imperfect linking of genes.

- The two step regulation hypothesis is appealing but the evidence provided in the manuscript is consistent but I don't agree that it supports the "two-step regulation" model.

Author Rebuttal to Initial comments

Response to reviewers' Comments:

We thank the editorial team and the reviewers for their very valuable comments and constructive suggestions, which we believe have helped to strengthen our manuscript. We **addressed all the editorial and reviewer comments** in our revised manuscript with **edits in red**, and provided a detailed point-by-point response as below.

Reviewer #1:

Editor summary: Reviewer #1 is impressed by the resource value of the dataset and by the analyses. Their main suggestion is about **extending the Mendelian Randomization (MR) analysis**, and **modifying some claims**.

Remarks to the Author:

The authors describe how genetic variation is affecting active regulatory elements (ARE) in brain, heart, muscle and lung. They observe a large number of genetic variants that affect AREs that are shared between tissues and that are specific to certain (combinations of) tissues. The authors **carefully** study this sharing: they first discover gAREs (genetically affected AREs) and then attempt replication in another tissue using liberal threshold, and ascertaining allelic direction concordance, observing, even for the weakly replicating gAREs (replication $1e-3 < P < 0.1$) that 82% share the same allelic direction.

Subsequently the authors ascertain to what extent a diverse collection of traits, on which GWAS has been conducted, are showing enriched association signal in regions that have been identified as gARE. Indeed, the authors observe this to be the case, and find enrichments for the expected trait - tissue combinations.

Then the authors use colocalization to identify 456 GWAS-haQTL gAREs for 43 traits.

What is exciting to observe is that for several of these co-localizing gAREs no cis-eQTLs are found, underscoring the relevance and importance of studying H3K27 acetylation in different tissues of a large number of samples.

I am impressed by the **dataset**, and the **analyses** that are conducted, and I believe this **paper** and **resource** is a very **important contribution to the field**.

We thank the reviewer for the very positive comments on our manuscript!

The last part of the paper, where eQTLs and gAREs are integrated, to me is however somewhat **speculative**. For instance, The authors state in the discussion: "gLink scores outperform gene-ARE distance, EpiMap and ABC scores on linking GWAS-haQTL-colocalized gAREs to target genes.". **For me it is not immediately clear that**

gLink indeed is better than ABC: for instance, figure 5c does not seem to include results on ABC. In my opinion, the paper would actually improve further if this section would be rephrased somewhat, e.g. as a novel way to generate interesting hypotheses for follow-up research.

We thank the reviewer for the suggestion, which we now adopt. To clarify, by “outperform” we mean that gLink scores show better performance from the perspective of linking GWAS-haQTL- colocalized gAREs to target genes based on Figure 6a. ABC is indeed a state-of-the-art strategy, which is why we used it as the **gold standard** for our comparison of the **other methods** relative to ABC in Figure 5c. (This is why no ABC results are shown in the PRC plot, and we now point out by the dash line the background of positive links based on ABC score, mentioned also in the Figure Caption).

To avoid confusion, we rephrased it as “gLink scores also show stronger enrichment for GWAS- haQTL-colocalized gAREs compared to gene-ARE distance, EpiMap, and ABC score”. We also now clarified that we see gLink as **complementary** rather than alternative to the existing methods (**page 10, line 483**). We have expanded our explanation of how these methods capture different types of information, and thus should all be considered when finding target genes of regulatory elements and genetic variants.

Major question:

Colocalization is used to ascertain overlap between GWAS traits and gAREs. In the field, colocalization is often combined with Mendelian randomisation (MR) analyses. The authors report on 456 GWAS-haQTL gAREs for 43 traits. Can the authors also **perform MR to study whether there is evidence to suggest these effects on diseases are likely mediated through these AREs?** The authors already use MR in a later section of the paper, so I suppose this is something they could do here as well. I believe it would be valuable to use it here as well.

We thank the reviewer for the important suggestion. We have reorganized the GWAS datasets used for enrichment analysis and coloc/MR analysis, following the reviewer’s suggestion in the minor questions. We report 1070 GWAS-haQTL colocalized gAREs (614 unique AREs across tissues) for 46 traits. We also performed MR and found 403 out of 1070 of them (38%) of them show causal effects at the FDR level of 0.2 (Benjamini-Hochberg multiple test). We added this result in the main text (**page 6, line 283**) and also updated the supplementary table (Table S6).

The authors state that for schizophrenia there are 20 co-localized GREs for which no eQTLs are known. Could it be that the GTEX sample-size is a limiting factor, and some of these GRE SNPs actually do give a small but significant or **cell-type specific cis-eQTL** effect in other larger bulk brain datasets or single- cell brain datasets? E.g. one way to address this question might be to use the 'Bryois et al, Nature Neuroscience

2022' paper that studied single-nuclei or the 'de Klein et al, BioRxiv 2021' preprint that studied >2,000 bulk samples.

We thank the reviewer for the important suggestion. It is indeed possible that bulk eQTLs do not have enough power to detect QTL effects at cell-type level. We checked the cell-type level eQTL data suggested by the reviewer

(Bryois et al, Nature Neuroscience 2022) for eQTL-GWAS colocalization.

As shown on the right, 18 out of 31 Schizophrenia GWAS-haQTL colocalization events in the brain (coloc PP4 ≥ 0.5), are captured by bulk eQTL at a permissive cutoff (coloc PP4 ≥ 0.1), shown as white blocks in the 2nd column of annotations labeled as "missed by bulk eQTL". 16 of these 18 loci could also be captured by eQTL signal at a specific cell type at the same permissive cutoff (0.1), shown as white blocks in the 1st column of annotations labeled as "missed by cell-type eQTL".

Interestingly, 10 out of 13 GWAS

could be explained by eQTL from a certain cell type (orange dots on the left), showing a more cell type-specific pattern compared to those loci captured both by bulk and cell-type eQTL.

The results support the reviewer's hypothesis that the missing eQTL regulation may be due to lack of power to detect cell type specific eQTL signals. We also note that there are still 3 loci that are captured by haQTL but not by either bulk or cell-type eQTL (in red box).

Furthermore, to detect robust loci missed by eQTL signal, we used a stringent coloc cutoff for GWAS-haQTL (PP4=0.5) and a permissive one for GWS-eQTL (PP4=0.1). If we used the same stringent cutoff of 0.5 for both analyses, there are 11 GWAS loci captured by haQTL but not by either bulk eQTL or single cell eQTL (green dots on the left). This is even more noteworthy given that the sample size of haQTL detection (109) is smaller than bulk eQTL (175) and single cell RNA-seq (192). One possible explanation for the discrepancy in loci detection is that both the cell-type and bulk eQTLs are detected in non-disease-relevant conditions, whereas the haQTLs may reflect epigenomic changes in disease-relevant conditions. Thus, eQTL regulation in cell type-specific and condition-specific manners may jointly account for the missing eQTL regulation we observed for bulk eQTL

data. These results also indicate that bulk epigenomic variation may capture the impact of genetic variants (QTL effects) with greater power than bulk eQTL for those loci where the effect may become visible only in specific cell types or specific conditions. We have now added these results as Supplementary Fig. 4c, and interpretation in the main text (page 7, line 304) and the discussion (page 9, line 442).

The authors study whether known GWAS variants are in LD with variants that affect GREs that are sex- biased. The authors highlight one example: "For example, SNP rs7090118 is a subthreshold GWAS hit ($P=2.3 \times 10^{-6}$) for hypertension in moderate LD ($r^2=0.54$) with the lead SNP (rs10764326, $P=1.5 \times 10^{-11}$).". I must admit that I **am a bit reserved on this example**, because of the sub-threshold significance and the limited LD. I believe the paper in itself is already very valuable and does not necessarily need this section. Maybe the authors could consider removing this paragraph from the paper?

We thank the reviewer for the very positive comments on our paper. We have implemented the suggestion of de-emphasizing this example, and moved the results related to sex-biased AREs to Supplementary Fig. 2h-i.

Minor questions:

The authors state: "We found that our H3K27ac profiles from the same tissue clustered together, and importantly co-clustered with tissue-matched samples from Roadmap/ENCODE/GGR (Fig. 1b), including 14 of 14 previously-profiled brain samples, 13 of 14 heart samples, 6 of 9 muscle samples, 2 of 3 healthy lung samples, and 18 of 19 lung tumor samples.". Can the authors speculate what is the reason 6 samples do not cluster as expected?

Thanks for this important point! We identified these six samples from reference epigenomes (BSS00080_AORTA, BSS01319_LEG MUSCLE, BSS01332_TRUNK MUSCLE, BSS01463_PSOAS MUSCLE, BSS01201_Lung, BSS01415_LUNG ADENOCARCINOMA) whose epigenomic profiles were far away from other tissue-matched samples (Fig. 1b). There are three possible reasons why the samples do not cluster as expected: (1) sample quality issues; (2) technical: the distance between samples may be skewed during the integration processes of the two datasets based on canonical correlation analysis (CCA) in R package Seurat due to unbalanced tissue/cell types between this study (four tissues) and Epimap (31 tissues); and (3) biological: profiles of bulk samples from different tissues may be similar due to dominant cell types shared. We first confirmed no sample showed any sign of a quality issue (1) based on preprocessing QC metrics and imputation

accuracy in a previous report¹. We then went back to the tissue archetype analysis for Epimap samples only (**Supplementary Fig. 2e**), where they were studied alone, to test for the technical effect (2) above. We found four of the six above samples (BSS00080_AORTA, BSS01319_LEG MUSCLE, BSS01332_TRUNK MUSCLE, BSS01463_PSOAS MUSCLE) are well clustered with the matched tissue samples before integration with the data from our study, suggesting that the technical effect (2) is the reason why these samples did not cluster as expected. BSS01201_Lung sample, is clustered with heart samples, with strong signal in heart archetype, indicating that the biological effect (3) could be the cause of why this sample did not cluster as expected. This may be partially due to the fact that this is a lung sample from a child. Finally, the BSS01415_LUNG ADENOCARCINOMA sample is clustered with

other adenocarcinoma samples instead of other lung tumors, which were all lung epithelial carcinoma. Thus the disparity in clustering for this sample is likely biological (3): it is from a different type of tumor from the other lung tumor samples.

We now removed “including 14 of 14 previously-profiled brain samples, 13 of 14 heart samples, 6 of 9 muscle samples, 2 of 3 healthy lung samples, and 18 of 19 lung tumor samples” to avoid distraction (**page 3, line 102**), and discussed these 6 samples in the

method section named “Quality control of H3K27ac ChIP-seq and lower-dimensional representation of H3K27ac profiles” to provide potential explanations (page 13, line 633).

There seems to be a discrepancy between the number of GWAS studies used in the LD score regression part and the subsequent colocalization part. Can the authors explain what is the reason this number is different? Can this be made consistent?

We thank the reviewer for the suggestion! We have now used 55 GWAS data for both analyses, mainly adding bipolar disorder and three lung related traits and removing the blood cell counts related traits, which are not directly related to our four tissues (brain, heart, muscle and lung). We updated the LD score regression results in Fig. 4a (shown below). In addition, we now updated the Table S4 that includes the information about these GWAS datasets.

Can the authors correct a typo in the discussion? "First, although we profied over". We've corrected the typo (page 10, line 496). Thank you!

Reviewer #2:

Editor summary: Reviewer #2 also says that dataset constitutes an important resource but notes that more details about **data and code availability** are needed. They think you should also provide some **clarification on the gLink analysis**.

Remarks to the Author:

This study produces a **valuable resource** of H3K27ac ChIP-seq datasets in 387 brain, heart, muscle, and lung tissue samples from the GTEx cohort. The authors **systematically identify** modules of H3K27ac peaks, call H3K27ac QTLs, and colocalize these QTLs with GWAS variants. The authors also **develop** a “gLink” analysis that uses QTL evidence to link H3K27ac peaks to target genes, and apply this to interpret GWAS variants.

Overall, I am **enthusiastic** about the **resource**, which represents **one of the largest** H3K27ac ChIP-seq

QTL studies and would be **generally useful** for the community. **Improved availability** or documentation of data and metadata is **essential** to make this resource **useful** for the **genomics community**. Clearer **justification** and **explanation** of the **gLink** analysis would also improve the study.

We thank the reviewer for the very positive comments on our manuscript!

We agree that the availability of all the data is essential, and with the suggestion of better justifying and explaining the gLink method, which we have now implemented in our revised manuscript.

Major Comments:

1. **Data availability:** Availability of data, metadata, and processed intermediate analysis files is essential to enable use of this resource by the community.

We agree with the reviewer, and have uploaded the sample metadata and the processed intermediate analysis files, including AREs in each tissue, merged AREs, raw/normalized ARE count matrices, ARE modules, haQTL summary statistics, colocalization results, ARE-gene linkages, etc., to the website: <https://data.broadinstitute.org/compbio1/eGTEx-H3K27ac/>.

- a. **Data:** The data is deposited on dbGap, and so I was not able to confirm availability. Please report exactly what files (raw data, processed data) are available there.

We uploaded the raw data (bam files) and the processed data (H3K27ac signal in bigwig format, and H3K27 peaks) for all 387 samples to dbGaP. We expect that readers will be able to access this data after applying for permission. Please check below about other data types. We now state in the “Data availability” section which data are available on dbGap as suggested by the reviewer.

- b. **Metadata:** Is there other eGTEx data from the same/similar samples that can be integrated? e.g. ATAC-seq would be particularly valuable for identifying regulatory elements at higher resolution. This paper should at least provide the needed metadata to integrate and explain how the samples used here overlap with other eGTEx efforts. Presumably this metadata could be reported without access control.

Thanks for the suggestion! We agree with the reviewer that the metadata would be very helpful for the community to integrate different resources. The metadata is provided as Table S1 now available at <https://data.broadinstitute.org/compbio1/eGTEx->

H3K27ac/supplementaryTables without access control.

c. **Intermediate analysis files:** The supplementary tables have only a small number of the intermediate analyses, and appear to include only significant hits (e.g., GWAS-QTL colocalizations about some threshold). Please instead provide **processed data files** with **full intermediate** analysis results, modified as needed to enable **open access** rather than access control. Many similar files are available open access from the GTEx portal for the main eQTL component of the project and so I do not believe that there would be access restrictions on similar files from this study.

(<https://gtexportal.org/home/datasets>) Examples of intermediate analysis files that would be useful:

- i. Full QTL results for all tested variant-peak pairs
- ii. Data matrices used for each heatmap figure
- iii. List of peaks and their counts + normalized counts across all samples
- iv. List of peaks and their mean normalized counts across tissues
- v. Full colocalization results

We thank the reviewer for this important suggestion, and we now provide these intermediate datasets here (https://data.broadinstitute.org/compbio1/eGTEx-H3K27ac/Processed_files/), including all the reviewer mentioned and others

- i. QTL (./haQTL/)
- ii. Data matrices for heatmap (./heatmaps/)
- iii. Peak and peak counts (./ARE_matrices)
- iv. peaks and their mean normalized counts (./ARE_matrices)
- v. colocalization and MR (./colocalizationAndMR/)
- vi. Modules of AREs and motif (./ARE_modules)
- vii. gLink scores (./gLinkScores_hg38)
- viii. GWAS summary statistics we used (./GWAS)
- ix. Peaks detected from each individual sample (./H3K27ac.peak.individual)
- x. sample metadata (./sampleMeta)
- xi. Sex biased ARE (./sex-ARE_hg19)
- xii. All the detected AREs (./tissueAREsAndmergedAREs)

2. Code availability: The code is not available at the link provided

The code is now fully available <https://data.broadinstitute.org/compbio1/eGTEx-H3K27ac/code>.

3. The authors should conduct a **power analysis** to help readers understand this datasets, and to inform future studies. This should include (1) reporting power for each tested variant-ARE pair, the PEER model beta, the estimated fold-change effect

in H3K27ac counts; and (2) summarizing power across the study

We agree with the reviewer that a power analysis would be very helpful and instructive for understanding this dataset and for guiding future study designs.

(1) We have now reported the power for the variant-gARE pairs based on `powerQTL`², which takes sample size, minor allele frequency, standard deviation of H3K27ac signal, and slope into consideration (please see the updated Table S3). We used `FastQTL` for haQTL calling, which does not provide beta value for PEER factors, but we instead provide the PEER covariate matrix (available at https://data.broadinstitute.org/compbio1/eGTEx-H3K27ac/Processed_files/ARE_matrices/covariatesForQTL/). The estimated fold-change effect in H3K27ac counts can be found as “beta” in Table S2.

(2) We then used two strategies to summarize the power across the study, including a) using `powerQTL` with the parameters observed from the haQTLs, and b) performing down-sampling with the brain samples from our studies, which was the largest sample group. For the first approach, we calculated the mean effect size for the haQTLs identified in each tissue. In addition, we partitioned the power estimation based on different minor allele frequencies (MAF) from 7.5% to 32.5% to demonstrate its impact on the power of QTL calling. As expected, the haQTLs with higher minor allele frequencies have better power to be detected (left panel in the figure below). For the down-sampling analysis, we sequentially and randomly down-sampled the 109 brain samples to different sample sizes ten times, and measured the detection rate of haQTLs that can be recapitulated after downsampling. We found that the haQTL detection power did not yet show a trend of saturation (right panel in the figure below), suggesting that more samples would further increase the number of haQTLs being detected. Altogether, the results indicate our samples are well powered for QTL with higher MAF (at least a power of 0.75 with 100 samples for SNPs with MAF>32.5%), and that approximately 175 samples are needed to reach at least a power of 0.5 for SNPs with MAF >12.5%. We now add these results as supplementary Fig. 3b-c, and add the

analysis in the discussion (page 10, line 498) and Methods (page 15, line 737).

4. The analysis about GWAS signals that colocalize with haQTLs but not eQTLs (20 of 59 SCZ loci) is **potentially very interesting**. The authors propose a 2-step model in Fig 4f. Could the authors address to what extent any of the following effects could provide alternative explanations for this category of GWAS signals:

a. Lack of power in the eQTL analysis or colocalization analysis

Given that the eQTL analysis was performed on larger numbers of samples than the haQTL analysis (175, 706, 386, and 515 samples for eQTL mapping compared to 109, 81, 89, 53 samples for haQTL mapping in brain, muscle, heart and lung, respectively), it is unlikely that the eQTL analysis is underpowered. We discuss the power to detect cell type events specifically in section c below. In addition, when we define eQTL-missing regulation loci, we use a stringent cutoff for GWAS-haQTL colocalization (coloc PP4 ≥ 0.5), and a permissive cutoff for GWAS-eQTL colocalization (coloc PP ≥ 0.1). This leads to identification of more robust detection of eQTL-missing loci even if eQTL detection power is limited at these loci.

b. Presence of multiple independent eQTL signals that confound analysis with coloc

We agree with the reviewer's point that multiple independent eQTL signals may confound analysis with coloc.

To directly address the reviewer's concern, we also tested how much these results are affected by multiple independent eQTL signals. We used the version of coloc allowing for multiple causal signals for each locus by running SuSiE first to identify independent signals. The results

shown on the right compare results for bulk eQTL colocalization by coloc and coloc-SuSiE for 31 Schizophrenia GWAS-haQTL colocalization loci. The two results are quite consistent:

only one out of 13 bulk-eQTL-missing GWAS loci from the brain could be due to independent eQTL signals confounding the analysis (shown with a black arrow).

Besides, in our analysis, we already used permissive cutoff for GWAS-eQTL colocalization to identify robust eQTL-missing GWAS loci. Therefore, the eQTL colocalization signals are more likely to be detectable even if they are affected by the existence of multiple independent eQTL signals.

In summary, it seems that multiple independent eQTL signals may not be the major reason for the loci that we detected as eQTL-missing. We have added this analysis as Supplementary Fig. 4b, added our interpretation in the main text (page 7, line 303), and updated the Methods accordingly (page 16, line 797).

A comparison between schizophrenia GWAS-eQTL colocalization analyses with/without SuSiE

c. Ability of haQTLs to pick up QTL signals at cell-type specific enhancers for rarer cell types in a tissue (e.g. microglia in brain), whereas eQTL signals would be washed out because the gene is expressed in multiple cell types in the tissue

We thank the reviewer for this important insight! It is indeed possible that bulk eQTLs do not have enough power to detect QTL effects at cell-type level. We checked the cell-type level eQTL data suggested by the reviewer (Bryois et al, Nature Neuroscience 2022)³ for eQTL-GWAS colocalization. As shown on the right, 18 out of 31 Schizophrenia GWAS-haQTL colocalization

events in the brain (coloc PP4 ≥ 0.5), are captured by bulk eQTL at a permissive cutoff (coloc PP4 ≥ 0.1), shown as white blocks in the 2nd column of annotations labeled as “missed by bulk eQTL”. 16 of these 18 loci could also be captured by eQTL signal at a specific cell type at the same permissive cutoff (0.1), shown as white blocks in the 1st column of annotations labeled as “missed by cell-type eQTL”. Interestingly, 10 out of 13 GWAS-haQTL colocalized gARE loci missed by bulk eQTL could be explained by eQTL from a certain cell type (orange dots on the left), showing a more cell type specific

pattern compared to those loci captured both by bulk and cell- type eQTL.

The results support the reviewer’s hypothesis that haQTL may be better to capture cell type specific QTL effect than bulk eQTL, which is an alternative explanation to our two-step model.

We also note that there are still 3 loci that are captured by haQTL but not by either bulk or cell-type eQTL (in red box). Furthermore, to detect robustness loci missed by eQTL signal, we used a stringent coloc cutoff for haQTL (PP4=0.5) and a permissive one for eQTL (PP4=0.1). If using the same cutoff of 0.5, there are 11 loci (green dots on the left),

captured by haQTL but neither by bulk eQTL nor single cell eQTL. This is even more noteworthy given that the sample size of haQTL detection (109) is smaller than bulk eQTL (175) and single cell RNA-seq

(192). One of the possible explanations is that both types of eQTLs are detected in non-disease-relevant conditions. Thus, eQTL regulation in cell type-specific and condition-specific manners may jointly account for the missing eQTL regulation we observed for bulk eQTL data. These results also indicate that bulk epigenomic variation may capture the impact of genetic variants (QTL effects) with greater power than bulk eQTL for those loci where the effect may become visible only in specific cell types or specific conditions.

We have now removed our two- step model, and added this result as Supplementary Fig. 4c and interpretation in the main text (page 7, line 304) and the discussion (page 9,

A comparison between schizophrenia GWAS-eQTL colocalization analyses using brain bulk and cell-type-level eQTL

line 442).

5. Could the authors simplify the **presentation of the gLink analysis**? I am a bit lost, and have various questions about the analysis:

We thank the reviewer for the comments. We've now simplified the presentation of the gLink analysis, and made additional changes in response to the comment and questions as detailed below.

a. Could the authors explain the **utility of the gLink analysis**? It seems to me that it will necessarily be a very incomplete list of enhancers for any given target gene, because it depends on there being eQTL variants for the gene (which is underpowered, and where natural genetic variation might not exist). It certainly is helpful to annotate fine-mapped eQTLs that directly overlap gARE peaks — thereby implicating the peak in regulation of the target gene ... Beyond this simple analysis, I am not understanding the utility of the more complex analysis presented

We thank the reviewer for the comments. We appreciate that the reviewer agrees that “Annotate fine-mapped eQTLs that directly overlap gARE peaks” is helpful to implicate “the peak in regulation of the target gene.” This is covered in gLink approach 1 (**Fig. 5a**). We also note that 26-61% of FM- eQTLs do not function through regulating nearby gARE activity even at a permissive cutoff (**Supplementary Fig. 5b**). This observation indicates that overlapping FM-eQTL with gAREs, the traditional approach in the field, may not be the only way to link gAREs to genes. We thus apply the gLink approach 2 to complement the first approach, namely linking gAREs to genes based on shared genetic regulation, i.e., genetic variants regulating both gARE activity and gene expression.

Altogether, our approach is complementary to the other approaches as it comprehensively prioritizes enhancer-gene linkage based on genetic evidence, providing: (1) the ability to capture genetically-driven links that may be missed by other approaches; and (2) the ability to track the impact of disease-associated genetic variants on enhancers and genes in a tissue-specific manner.

We agree with the reviewer that the analysis of gLink scores seems complex, and therefore we have now simplified it in the main text (**page 7, line 320**). We also added additional text in the discussion to clarify the important differences and utilities of the different gLink scores (**page 10, line 486**).

b. There are 6 different gLink scores, but then on page 10 line 1 what is the “unified score system for gLink scores”?

We described how we integrated the six gLink scores to define a “unified score” in the Method section “gLink scores”. We've now also clarified it in the main text, as “under a

unified score system, or the maximum transformed score among six gLink scores, across four tissues” (**page 8, line 391**), to avoid potential confusion based on the reviewer’s comment.

c. Page 8 — 2kb distance between eQTL and gARE is a fairly generous window for linking gAREs to variants and target genes. eQTL variants and GWAS variants much more highly enriched in chromatin accessible regions than in H3K27ac peaks, which tend to include large sequences adjacent to chromatin accessible regions. If using H3K27ac peaks directly for variant overlaps, I would suggest a smaller distance threshold, like 250bp from the peak which would capture a chromatin accessible site adjacent to the H3K27ac peak. Alternatively, it could be even better to infer the location of the chromatin accessible site through analysis of dips in the H3K27ac signal, and use those regions for variant overlap.

We thank the reviewer for the very helpful and detailed suggestions, which we now address in our revised manuscript. We agree that dips in the H3K27ac peaks could potentially give an even higher signal⁴, and our 2kb threshold was initially motivated to be able to capture the ‘correct’ dip that might be responsible for the signal, or the correct ‘set of dips’ in the case of multiple driver enhancers and potentially multiple variants in the same LD block⁵. We now include a new analysis showing how the enrichment changes at different thresholds, and we do indeed see that thresholds up to 2kb show the highest enrichment, after which the enrichment curves flatten out for fine-mapped eQTLs (new Supplementary Fig. 5a). Indeed, as the reviewer expected, the highest enrichment is at 250bp, leading to the highest specificity, but unfortunately at the cost of missing out on many potentially valid candidates (decreased sensitivity). Thus, we feel that the 2kb window is appropriate to maximize sensitivity, which was our goal here.

We have now added this result as Supplementary Fig. 5a, updated the main text (**page 7, line 326**), and included the discussion regarding peaks, multiple enhancer variants, and maximizing sensitivity in the Methods (**page 17, line 802**) accordingly.

eQTL/FM-eQTL Proximal enrichment of genomic regions at different bins of SNP-region distance

d. Fig 5c “Validation of gLink scores with ABC score” should probably be better titled “Comparison of gLink scores to ABC score”

We agree and have changed the title to “Comparison of gLink scores and other scores to the ABC score”.

e. Fig 6a. dotted line— How is the background defined?

The background is defined as the proportion of the gAREs with candidate gARE-gene pairs (gARE- TSS distance within 2kb-1Mb) that are captured by GWAS-haQTL colocalization (Methods, Page 18, line 854).

f. Fig 6a. dotted line— is it correct that this dotted line defines the background for one of these methods, but not all? E.g. if you wanted to calculate an enrichment over background for each method, does each method need its own background calculation or is the background the same for all methods?

Since all the linking methods are scoring the same set of candidate gARE-gene pairs, as mentioned above, we used the same background for the evaluation of all the methods. We have now clarified this in the method section (Page 18, line 854).

g. Is the validation analysis in Fig 6b circular, because eQTL signals were used to define the gLink scores?

We thank the reviewer for this important comment, which we address in our revised manuscript. Yes, we agree about the circularity, and we now state: “We confirmed that for brain-related traits, gLink scores are more likely than the other scores to identify disease target genes supported by GWAS-eQTL-colocalization (Fig. 6b), which is as expected since eQTLs were used as one of the information sources in our gLink

method” (page 8, line 376).

To provide a proper validation, we now also bring in a new dataset, namely the target genes predicted for brain-related traits based on GWAS summary statistics and gene features independent of eQTL⁶. For the five brain-related traits where any of the scores show enriched overlap with the previous prediction, we found that at least one of our gLink scores showed significant and higher enrichment of benchmark genes than ABC and Epimap scores (see figure below), confirming the ability of our gLink scores to detect disease relevant circuits. We have now added this result as Supplementary Fig. 6b, and updated the main text (page 8, line 378)

and methods accordingly (page 18, line 858).

h. For the comparison of gLink scores vs EpiMap and ABC, could the authors explain why these scores perform better (at least, have better recall in Fig 6a)? Is it because of the threshold on the scores chosen?

Fig 6a shows that gAREs linked to genes based on gLink scores are more likely to be associated with disease GWAS signal than other scores. There are two possible reasons: (1) a genetic signal regulating gARE activity may be more likely to regulate phenotypic variation if it also regulates gene expression; (2) gLink scores may capture disease relevant tissue-specific effects, which may be missing from Epimap and ABC. We now stress these explanations in the discussion (page 10, line 486)

To compare fairly across different scores, we use the same threshold of the top 10% for all scores. The result is robust if we use other cutoffs, as shown below for a cutoff of the top 20%. We have now included this analysis in the bottom two rows of Supplementary Fig. 6a and updated the Methods (page 18, line 856).

Brain-related traits (with top 20% links according to different scores)

i. For the gLink-GWAS analysis on page 10, given that enhancers are linked to genes based on fine-mapped eQTLs, does gLink analysis give something substantially different than simply doing eQTL colocalization? The fact that 10 of 12 gLink predictions are also colocalized eQTL-GWAS signals would suggest this might be the case. What about the reverse? Does colocalized eQTL-GWAS signals find

many more target genes than gLink predictions?

We expect that colocalization of GWAS-eQTL may find more target genes than gLink predictions given that (1) the majority of the non-coding regulatory elements impact phenotype through gene expression, therefore eQTLs more directly explain phenotypic variation than haQTLs do by nature; and (2) eQTLs in GTEx are discovered with higher power than haQTL in our cohort (175, 706, 386, and 515 samples for eQTL mapping vs 109, 81, 89, and 53 samples for haQTL mapping in brain, muscle, heart and lung, respectively). Indeed, we showed the global pictures of sharing and specificity of haQTL and eQTL colocalization with GWAS signals in the brain across multiple traits. eQTL-specific GWAS colocalization loci (GWAS-eQTL coloc PP4≥0.5 and GWAS-haQTL coloc PP4≤0.1) outnumber haQTL-specific GWAS colocalization loci (GWAS-haQTL coloc PP4≥0.5 and GWAS-eQTL coloc PP4≤0.1) as expected, with a median ratio of 17.4:1 as shown below. We have added the result as Supplementary Fig. 4a, and added this point into the discussion (page 10, line 460) to avoid any confusion, and updated the Methods accordingly (page 16, line 790).

GWAS-haQTL vs. GWAS-eQTL colocalization over 1694 brain gARE loci with significant brain eQTL

However, regardless of the simple number comparison, we show that the gLink-GWAS analysis can identify links between biological mechanisms that eQTLs-GWAS colocalization cannot, because gLink-GWAS analysis provides an intermediate layer of regulation that fills the gap between disease associated genetic signals and gene expression, which also serves as a confirmation of the genetic signal.

Minor Comments:

1. Please add page numbers for review

Thanks for the reminder! We've added page numbers in the revised manuscript.

2. Please report the range of unique reads per ChIP sample at top of page 2

We added "387 samples in brain prefrontal cortex, heart left ventricle, skeletal muscle, and lung, with median total unique reads of 10.8M, 11.7M, 9.9M, and 10.5M, respectively, from 256 GTEx subjects" in the main text as the reviewer suggested (page 2, line 69), and also added "These criteria resulted in 113 brain, 100 heart, 108 muscle, and 66 lung samples, with total unique reads ranging 6.2-11.7M, 5.0- 44.2M, 5.1-45.8M, 6.2-10.5M for each tissue, respectively" in the Methods (page 13, line 624).

3. "found primarily at active enhancers and to a lower degree at active promoters". I believe it would be more correct to say it is found at both active enhancers and promoters.

We agree and have made the change (page 2, line 59).

4. "our four tissues sampled here might not be their primarily tissues of action". I am not sure what this means, with regards to an enhancer module. Please clarify.

Is another possibility that they reflect enhancers active in certain cell types e.g. vascular or immune that are present in many tissues?

We thank the reviewer for pointing this out. By “primary tissue of action”, we meant a certain tissue where a specific module of enhancers shows the most activity. Now we rephrase it as “primary tissue of activity” to avoid any confusion (page 3, line 36). With our analysis, we potentially capture a more comprehensive set of AREs that are detected in our four tissues even though they have the strongest signal in tissues. As the reviewer pointed out, the activity of an enhancer module in a tissue may reflect the activity of certain cell types. Thus, the “primary tissue of activity” may be the tissue with the highest abundance of the cell type showing the highest enhancer module activity. We believe that applying the same analysis to more cell-type level epigenomic data in the future will lead to results of higher resolution, allowing identification of the “primary cell type of activity” for enhancer modules. We now also discuss this important point in the revised discussion section (page 10, line 506).

5. Fig 4c — The p-values for SCZ GWAS look to be not genome-wide significant. Could the authors clarify how they chose sub-threshold significant SCZ GWAS loci to include in the analysis?

We thank the reviewer for this question. We chose $1e-5$ as a nominal *P*-value cutoff for SCZ GWAS signal, and this is now included in the Methods. Previous studies have indicated that sub-threshold GWAS also contribute to heritability⁷, which is why we include these loci in our analysis.

6. Page 6 — please specify in the main text the PP4 probability threshold used to call “colocalization” We thank the reviewer for this comment and have now added in the main text that we used

$PP4 \geq 0.5$ to identify high-confidence GWAS-haQTL colocalized gAREs (Fig. 4b-f, Supplementary Fig.

4a-c) (page 6, line 284), and applied a permissive cutoff ($PP4 \geq 0.1$) for disease target genes identification (page 8, line 375).

Reviewer #3:

Editor summary: Reviewer #3 equally **highlights the value of the dataset**. Their suggestions are mainly aimed at improving/extended the current analyses. There are **no major concerns**.

Remarks to the Author:

Hou et al. present results from an enhancing GTEx (eGTEx) study that profiles epigenomic variation (H3K27ac) in 387 brain, heart, muscle, and lung samples from GTEx. They identify 282K active regulatory elements (AREs) and characterize their tissue-specific activity patterns. They then proceed to show some proof-of-principle

analyses to illustrate the **utility** of these AREs in understanding the **molecular basis** of **complex traits**, including haQTL analyses, GWAS-haQTL colocalization analysis, gene-ARE linking scores to prioritize disease target genes. Overall, the paper is **clearly written**, the analyses are **well described, detailed** and use mostly **well-established** analytical tools from the GTEx and the Roadmap Epigenomics projects. The paper provides a **complementary data resource** to the eQTL data commonly used in follow-up GWAS studies that geneticists may find very helpful. A **weakness** of the paper is that some of the proof-of-principle analyses were **underdeveloped** and could be **improved**.

Detailed comments are below.

1. Can the authors provide a measure of **false discovery rate among the detected AREs**? In particular, among the newly discovered AREs can we get an estimate of FPR?

We thank the reviewer for the question about the false positive rate of ARE detection, which we now address in our revised manuscript. Indeed, we had initially relied on the enrichment of newly- detected AREs to show their biological relevance, including: (1) enrichment of biological processes (Supplementary Fig. 2b last column); (2) enrichment of disease GWAS signal (Fig. 3a); (3) enrichment of tissue ARE-specific gAREs (Fig. 3F), confirming that our approach captures weak but reliable regulatory element signals.

We now also include a replication analysis for ARE detection, and show a **replication rate of AREs at 84% in brain, 83% in heart, 88% in muscle, and 78% in lung**. For *Newly-detected* AREs (G14), these numbers are slightly lower as expected, since these AREs vary more across

samples and are thus less likely to replicate using our metrics – they are: 80% in brain, 68% in heart, 77% in muscle, and 55% in lung. To calculate these numbers, we repeated ARE detection for each tissue with the same procedure in 80% of Tier1 samples randomly selected for 10 times and checked the proportions of these AREs that are also detected in the rest Tier 1 and all Tier 2 samples. We have now added this results in Methods (**page 14, line 667**).

For each sample, we also provided FDR for ARE detection (peak bed files for each sample in our website https://data.broadinstitute.org/compbio1/eGTEx-H3K27ac/Processed_files/H3K27ac.peak.individual/). We note that FDR calculations for ARE detected at the population-level are particularly difficult, since we do not expect each ARE to be detected in every sample of the same tissue due to genetic and environmental factors. To ensure the valid detection, we followed previously published studies on the variation of epigenomic signal at the population level^{8,9}. We have made three adjustments: (1) we only detect ARE in samples with Tier 1 good quality; (2) we also allowed peaks to be detected in our dataset that is overlapped with the tissue-matched samples in the ROADMAP project, resulting in all peaks being detected in at least two individuals among samples from our cohort and from the ROADMAP cohort

from the same tissue, and (3) for a more robust signal, we also performed a further filtration based on promoter/enhancer associated chromatin states from tissue-matched ROADMAP samples (Methods, **page 14, line 661**).

2. The correlations between the H3K27ac profiles in this study and previous studies (Sup Fig 1) seem low, can the authors provide additional comments on this?

We thank the reviewer for the observation. We believe the low correlation (~ 0.03) was due to the fact that we used 405k tissue/cell type-specific H3K27ac peaks from the Roadmap project, identified from 98 tissue/cell samples including many tissues besides the four target tissues in our study. We have now repeated the correlation analysis using the intersected peak set (250k) between the peak set (282k) in our study and the previous peak set (405k), and found high correlations (~ 0.6) between our samples and the biologically-relevant samples from EpiMap.

Please see the figure below, and we have now updated Supplementary Fig. 1a and the Methods (**page 14, line 653**) accordingly.

3. The authors perform NMF to remove unwanted variation from the H3K27ac profiles. Why not perform this step from the beginning?

In Fig. 2, we specifically seek to capture this signal, which represents variation across

different tissues, and thus its removal would be premature. We remove it at the beginning of Fig. 3, where we focus on the variation across individuals and such removal is essential.

4. On page 5: the authors state: “We also distinguish Type-I “haQTL-Shared” and Type-II “haQTL- Specific” gAREs based on the nominal P-value threshold ($P < 0.02$) in the replication tissues”. How is this **threshold** for nominal significance (0.02) **selected**? Seems arbitrary.

We stated in the legend of Supplementary Fig. 3h that “the nominal P-value threshold was set to

0.02 in the replication tissue to define a Type-I gARE (haQTL-Shared) between the discovery and replication tissues, which makes the directionality consistency between the two tissues be over 95%.” To make it more visible to the readers, we have now also stated it in the Methods (**page 16, line 761**).

5. Overall, I think it would be helpful to have a more **thorough comparison of eQTLs and haQTLs** for example in terms of co-localization results. Although it is clear that haQTLs can provide additional information on top of eQTLs and partially address the **missing regulation problem** with eQTLs, it is not clear from these comparisons how useful eQTLs and haQTLs are on their own. For example, how many GWAS-eQTL colocalizations are missed by GWAS-haQTL colocalizations?

We agree with the reviewer’s point. We expect that eQTL could capture much more GWAS signal than haQTL, given that: (1) the majority of the non-coding regulatory elements impact phenotype through gene expression, therefore eQTLs more directly explain phenotypic variation than haQTLs do by nature; and (2) eQTLs in GTEx are discovered with higher power than haQTL in our cohort (175, 706, 386, and 515 samples for eQTL mapping vs 109, 81, 89, 53 samples for haQTL mapping in brain, muscle, heart and lung, respectively). Indeed, we showed the global pictures of sharing and specificity of haQTL and eQTL colocalization with GWAS signals in the brain across multiple traits as below. eQTL-specific GWAS colocalization loci (GWAS-eQTL coloc $PP4 \geq 0.5$ and GWAS-haQTL coloc $PP4 \leq 0.1$) outnumber haQTL-specific GWAS colocalization loci (GWAS-haQTL coloc $PP4 \geq 0.5$ and GWAS-eQTL coloc $PP4 \leq 0.1$) as expected, with a median ratio of 17.4:1 as shown below, and they also have substantial shared loci. We have now added this result as Supplementary Fig. 4a, and included this very important point in the discussion (**page 10, line 460**).

GWAS-haQTL vs. GWAS-eQTL colocalization over 1694 brain gARE loci with significant brain eQTL

6. Although the two-step regulation may be consistent with the observed data, it does not in any case imply that is the only true underlying model ... this part needs to be toned down.

We agree. We carried out colocalization with cell-type-level eQTLs from sn-RNAseq data, and found that most of the bulk-eQTL-missing GWAS loci can be explained by cell-type-level eQTLs (**Supplementary Fig. 4c**). We have now removed the two-step hypothesis entirely, and updated the main text (**page 7, line 304**) and the discussion (**page 9, line 442**).

7. I find the part on the gLINK score underdeveloped. I find the comparisons among the different scores not convincing because the scores being used as benchmark are in reality not that accurate. The authors should identify better benchmarks. For example, previous papers linking loci/variants to genes might be helpful here (PMID: 35668300; PMID: 35147782, PMID: 34711957).

We thank the reviewer for the suggestion of these additional benchmarks, including cS2G (PMID: 35668300), effector index for target gene prediction (PMID: 35147782), and L2G from openTarget (PMID: 34711957). We now incorporate it in our revised manuscript. Specifically, we used

target/critical genes previously predicted⁶, which are used in the cS2G paper¹⁰, to confirm the target genes predicted by gLink scores vs. EpiMap, ABC, or distance. We

found that at least one of our gLink scores showed significant and higher enrichment of benchmark genes than ABC and Epimap scores in five brain-related traits (see figure below), supporting the power of our gLink scores to detect disease relevant circuits. We now add this result as Supplementary Fig. 6c and in the main text (page 8, line 378).

We agree with the reviewer that the benchmark scores we used (ABC score and EpiMap) may not be ideal. Indeed, enhancer-gene linkage has been an important and open question in the field during the past 10 years^{11–15}, and we believe no perfect golden standard data is available, considering that different methods capture different aspects and that biologically the links are dynamic. Similarly, as stated in the cS2G study (PMID: 35668300)¹⁰ mentioned by the reviewer, they also find very low concordance across different SNP-to-gene strategies: “We observed low concordance between the S2G strategies: the average overlap fraction is 0.14 across the 34 S2G strategies, 0.19 across the 13 main strategies, and 0.08 across the 10 main functionally informed strategies; similarly, the average correlation is 0.10 across the 34 S2G strategies, 0.09 across the 13 main strategies, and 0.05 across the 10 main functionally informed strategies”.

We selected ABC and EpiMap as benchmark scores as they represent two major types of strategies when inferring the linking between an ARE and a gene, namely, based on physical chromatin interaction and based on correlation across different tissues/cell types, respectively, and because they encompass the largest diversity of tissues thus providing a broader benchmark that can be relevant for other GTEx tissues as well. As we discuss in our paper, our gLink scores provide a **third complementary strategy**, which is based on genetics. One of the significant contributions of our work is the idea that different strategies can prioritize different sets of enhancer-gene links that may

function under different contexts (during differentiation, or during disease progression), and thus the linkages need to be considered in a combinatorial way, which is also the strategy used in the cS2G paper.

We also noted that for the three benchmark datasets the reviewer mentioned, none of them provide enhancer-gene linking scores in any of the four tissues we studied (brain, heart, muscle and lung), and thus could not be used as direct benchmark data.

8. The authors show a comparison of gLINK scores with other scores e.g. EpiMap and ABC score to GWAS- eQTL colocalization but it is hard to say whether gLINK scores perform better simply because (as the authors acknowledge) they “tend to capture gARE-proximal genetic signals associated with gene expression”.

We agree with the reviewer that “perform better” is unspecific and misleading. We have now removed “perform better”, and specifically stated that “gLink scores also show stronger enrichment for GWAS-haQTL-colocalized gAREs” to avoid potential confusion. The stronger enrichment may be due to the fact that they “tend to capture gARE-proximal genetic signals associated with gene expression, which are more likely to be associated with GWAS signals” (**page 10, line 483**).

The central point for the section of gLink scores is **not to demonstrate** that they “perform better” than other scores. Instead, we aim to propose that gLink scores, compared to other linking scores, provide a different perspective and show different properties in terms of GWAS-haQTL colocalized gARE enrichment (Fig. 6a, Supplementary Fig. 6a), target genes (Fig. 6b, Supplementary Fig. 6b-c), and distribution of distance between predicted pairs (Supplementary Fig. 6d), and thus should be considered as a **complementary** approach. We have now added this in the discussion (**Page 10, line 481**).

9. Also related to the gLINK scores, there are six different scores being proposed. Each of them is pretty simple. Is there a way to combine them as one score? since given the issues mentioned at the previous point, it is not clear which score one can use.

We thank the reviewer for the comment regarding the usage of gLink score. We included in the Methods section a description of how we standardized the gLink scores so that they are comparable, mainly based on ABC scores as a benchmark dataset, as well as how we calculated a maximum transformed score across six gLink scores as a unified gLink score. We have now added in the main text a brief clarification of the unified score system as “the maximum transformed score among six gLink scores” (**page 8, line 391**) to provide a potential way to apply gLink scores.

We are also aware of the limitations of this unified score system: (1) ABC may not be the ideal benchmark dataset; and (2) a unified score may ignore the fact that the performance of each score at specific loci depends on the power of different QTLs which derive gLink scores. Specifically, approach 1 gLink scores (*gARE-dist-to-*

FMeQTL and *gARE-proximal-eQTL*) rely on eQTL power, while approach 2 gLink scores (*coloc-PP4*, *coloc-PP4/3*, *MR*, and *ExpPGS-gARE-corr.*) rely on both eQTL and haQTL powers. Thus, we highly recommend that users select gLink scores or build a unified score based on their own data and context. We now put this point in the Methods for better usage (page 17, line 837).

Reviewer #4:

Editor summary: Reviewer #4 echoes what the other reviewers say about the **dataset**. They have some **minor critiques** related to the text, and the **level of conceptual advance**. However, the latter point is **not a major editorial concern** on this occasion.

Remarks to the Author:

Hou et al. profile histone acetylation marks in 387 GTEx samples covering brain, heart, muscle, and lung. They identify peaks termed active regulatory elements (ARE), compute their association with genetic variation and finally attempt to explain disease-associated loci in 78 GWAS, identify tissues of action and driver SNPs. They also propose a method to link active regulatory elements to genes.

The 387 ChIP-seq data paired with WGS and RNA-seq already available for the same samples represent an important resource for the scientific community. However, the data does not seem to be yet available on ANVIL. Publication should be contingent on their availability.

The figures are very helpful and the analysis is sound.

I found the manuscript **hard to read** with details that get in the way of the narrative. Some sections could be **simplified** (for example the explanation of **glink** method), the main idea explained up front and provide details later.

We thank the reviewer for the comment! We have rephrased the manuscript to make it concise and straightforward (page 7, line 320). We moved the technical details of gLink to the Methods and simplified the description in the main text. Based on the reviewer's comment below, we also renamed the Type I, II, and III gARE groups based on their biological implications, to make them more intuitive to the readers (page 4, line 193).

..

The **application of the resource** did not seem very compelling compared to what was published before by **ENCODE and Roadmap epigenome** projects. What did we learn with this new data that we didn't already know before?

We thank the reviewer for the question. We believe there are at least two novel aspects of our analyses that have not been achieved by the ENCODE or Roadmap projects:

- 1) The samples used by ENCODE and Roadmap covered a wide range of

tissues and cell lines, but lacked variation across individuals and genetics. Our eGTEx samples provide a resource to understand the variation across sex and due to genetic variants and to interpret disease loci from the epigenomic layer.

- 2) We provide the first multi-tissue QTL analysis of H3K27ac, which allows us to investigate the sharing and specificity of haQTL and their genetic association between tissues.

We have highlighted these points into the first paragraph of the revised discussion to help readers realize the potential utility of this resource (**page 9, line 415**).

The authors **combined** analysis with ENCODE and Roadmap data which facilitates a more interpretable grouping of the clusters. It is also **reassuring** to see that the new data falls within expected clusters.

We thank the reviewer for the positive comment!

Below are some additional comments to the authors:

- In my opinion, defining new types of shared or tissue-specific QTLs (**type I, II, and III**) adds unnecessary cognitive load to the reader.

We thank the reviewer for this comment. Understanding tissue/cell type/context specific regulation is one of the fundamental questions for the fields of both epigenetic regulation¹⁶ and disease genetics¹⁷. Our approach to define different types of haQTLs is a framework to decouple two different factors that contribute to genetic regulation of ARE activity: 1) whether a specific ARE is active in a certain tissue in general, and 2) whether it is impacted by a genetic variant at a certain loci. This framework could be applied to other QTLs, or to compare QTLs at different levels (e.g., cell type level). Based on the results, researchers could then investigate different transcription factors or chromatin remodelers that participate in the regulation of such tissue-specificity or tissue- sharing.

We are thankful for the reviewer's concern, and agree to simplify the nomenclature. We now simply name these gARE types based on their actual biological meanings. Specifically, we used "haQTL- Shared gARE", "haQTL-Specific gARE", and "ARE-specific gARE" to replace Type I, II, and III gARE groups in the main text and figures.

- Calling **alternative causal SNPs** based on the haQTL could be misleading. If the top QTL and the top GWAS variants are different, the more obvious conclusion would be that they are not colocalized, i.e. other mechanisms need to be found rather than claiming that the top GWAS association was "wrong".

We agree with the reviewer that our description of the alternative causal SNPs could be misleading. We have now changed it to "causal SNPs different from the lead SNP in a specific GWAS locus". in the main text (**page 7, line 312**).

We also emphasize there may be different modes of genetic regulation as “Collectively, the colocalization across haQTL, eQTL, and GWAS revealed different modes of regulation, suggesting the complexity of regulatory circuits in different loci and genetic traits.” in the discussion (page 10, line 468)

- “CORO7 and NMRAL1 proteins regulate vesicle transport and cellular response to redox changes respectively. The observation that both of these genes link to the same GWAS-haQTL-colocalized gARE indicates a coordinated dysregulation of both processes in the schizophrenia etiology” This conclusion seems **a bit speculative** given the imperfect linking of genes.

We agree and have now revised the sentence to “The observation that both of these genes linking to the same GWAS-haQTL-colocalized gARE make us speculate a potential coordinated dysregulation of both processes in the schizophrenia etiology” (page 8, line 404).

- The two-step regulation hypothesis is appealing but the evidence provided in the manuscript is consistent but I don’t agree that it supports the “two-step regulation” model.

We thank the reviewer for this comment. We carried out colocalization with cell-type-level eQTLs from sn-RNAseq data, and found that most of the bulk-eQTL-missing GWAS loci can be explained by cell-type-level eQTLs (**Supplementary Fig. 4c**). We now removed the two-step regulation model, and added our new results in the main text (page 7, line 304) and the discussion (page 9, line 442).

Reference

1. Boix, C. A., James, B. T., Park, Y. P., Meuleman, W. & Kellis, M. Regulatory genomic circuitry of human disease loci by integrative epigenomics. *Nature* **590**, 300–307 (2021).
2. Dong, X., Li, X., Chang, T.-W., Weiss, S. T. & Qiu, W. powerEQTL: An R package and shiny application for sample size and power calculation of bulk tissue and single-cell eQTL analysis. Preprint at <https://doi.org/10.1101/2020.12.15.422954>.
3. Bryois, J. *et al.* Cell-type-specific cis-eQTLs in eight human brain cell types identify novel risk genes for psychiatric and neurological disorders. *Nat. Neurosci.* **25**, 1104–1112 (2022).

4. Ernst, J. *et al.* Genome-scale high-resolution mapping of activating and repressive nucleotides in regulatory regions. *Nat. Biotechnol.* **34**, 1180–1190 (2016).
5. Corradin, O. *et al.* Combinatorial effects of multiple enhancer variants in linkage disequilibrium dictate levels of gene expression to confer susceptibility to common traits. *Genome Res.* (2013) doi:10.1101/gr.164079.113.
6. Weeks, E. M. *et al.* Leveraging polygenic enrichments of gene features to predict genes underlying complex traits and diseases. Preprint at <https://doi.org/10.1101/2020.09.08.20190561>.
7. Wang, X. *et al.* Discovery and validation of sub-threshold genome-wide association study loci using epigenomic signatures. (2016) doi:10.7554/eLife.10557.
8. Grubert, F. *et al.* Genetic Control of Chromatin States in Humans Involves Local and Distal Chromosomal Interactions. *Cell* **162**, 1051–1065 (2015).
9. Sun, W. *et al.* Histone Acetylome-wide Association Study of Autism Spectrum Disorder. *Cell* **167**, 1385–1397.e11 (2016).
10. Gazal, S. *et al.* Combining SNP-to-gene linking strategies to identify disease genes and assess disease omnigenicity. *Nat. Genet.* **54**, 827–836 (2022).
11. Ernst, J. *et al.* Mapping and analysis of chromatin state dynamics in nine human cell types. *Nature* **473**, 43–49 (2011).
12. He, B., Chen, C., Teng, L. & Tan, K. Global view of enhancer-promoter interactome in human cells. *Proc. Natl. Acad. Sci. U. S. A.* **111**, E2191–9 (2014).

13. Whalen, S., Truty, R. M. & Pollard, K. S. Enhancer-promoter interactions are encoded by complex genomic signatures on looping chromatin. *Nat. Genet.* **48**, 488–496 (2016).
14. Cao, Q. *et al.* Reconstruction of enhancer-target networks in 935 samples of human primary cells, tissues and cell lines. *Nat. Genet.* **49**, 1428–1436 (2017).
15. Wang, H., Huang, B. & Wang, J. Predict long-range enhancer regulation based on protein-protein interactions between transcription factors. *Nucleic Acids Res.* **49**, 10347–10368 (2021).
16. Kundaje, A. *et al.* Integrative analysis of 111 reference human epigenomes. *Nature* **518**, 317–330 (2015).
17. Uffelmann, E. *et al.* Genome-wide association studies. *Nature Reviews Methods Primers* **1**, 1–21 (2021).

Decision Letter, first revision:

9th Mar 2023

Dear Manolis,

Thank you for submitting your revised manuscript "Multi-tissue epigenomic variation atlas across 387 GTEx samples interprets disease genetics in brain, heart, muscle, and lung" (NG-A60452R). It has now been seen by the original referees and their comments are below. I apologize for the slow process.

The reviewers find that the paper has improved in revision, and therefore we'll be happy in principle to publish it in Nature Genetics, pending minor revisions to comply with our editorial and formatting guidelines.

Thank you again for your interest in Nature Genetics. Please do not hesitate to contact me if you have

any questions.

Congratulations!

Sincerely,

Tiago

Tiago Faial, PhD
Chief Editor
Nature Genetics
<https://orcid.org/0000-0003-0864-1200>

Reviewer #1 (Remarks to the Author):

The authors have adequately addressed the comments that I raised. I would like to thank the authors for providing honest answers and for incorporating the results that were a result of these questions.

Reviewer #2 (Remarks to the Author):

The authors have addressed most of my suggestions. It's quite interesting that the bulk H3K27ac-QTL analysis is finding many colocalizations that bulk eQTLs did not, but that are present in the cell-type-specific eQTL datasets.

One remaining request: Thanks for sharing all of the data, metadata, code, and intermediate analysis through the web site. To make these accessible to the community, the authors should add README files to explain the source and format of each set of data. Minimally, explain what the columns in each file represent.

Reviewer #3 (Remarks to the Author):

The authors have done a good job in addressing my comments. The additional analyses and clarifications do provide a better picture of the potential of these data in downstream analyses.

Reviewer #4 (Remarks to the Author):

The authors have addressed my suggestions satisfactorily.

Author Rebuttal, first revision:

We would like to express our sincere gratitude to the editorial team and the reviewers for evaluating the revised manuscript. We are delighted to learn that all the reviewers have acknowledged and appreciated the revisions we made. Their valuable comments have enhanced the solidity and comprehensiveness of our analyses, and also made the manuscript more accessible to a broader readership.

Reviewer #1 (Remarks to the Author):

The authors have adequately addressed the comments that I raised. I would like to thank the authors for providing honest answers and for incorporating the results that were a result of these questions.

Thanks for the reviewer's comments and thanks for the valuable suggestions to improve the manuscript!

Reviewer #2 (Remarks to the Author):

The authors have addressed most of my suggestions. It's quite interesting that the bulk H3K27ac-QTL analysis is finding many colocalizations that bulk eQTLs did not, but that are present in the cell-type-specific eQTL datasets.

One remaining request: Thanks for sharing all of the data, metadata, code, and intermediate analysis through the web site. To make these accessible to the community, the authors should add README files to explain the source and format of each set of data. Minimally, explain what the columns in each file represent.

Thanks for the reviewer's important comments and thanks for the valuable suggestions to improve the manuscript! We have provided a readme file in each folder to explain the format of each data.

Reviewer #3 (Remarks to the Author):

The authors have done a good job in addressing my comments. The additional analyses and clarifications do provide a better picture of the potential of these data in downstream analyses.

Thanks for the reviewer's comments and thanks for the valuable suggestions to improve the manuscript!

Reviewer #4 (Remarks to the Author):

The authors have addressed my suggestions satisfactorily.

Thanks for the reviewer's comments and thanks for the valuable suggestions to improve the manuscript!

Final Decision Letter:

22nd Aug 2023

Dear Dr. Kellis,

I am delighted to say that your manuscript "Multi-tissue H3K27ac profiling of GTEx samples links epigenomic variation to disease" has been accepted for publication in an upcoming issue of Nature Genetics.

Your paper will be published online after we receive your corrections and will appear in print in the next available issue. You can find out your date of online publication by contacting the Nature Press Office (press@nature.com) after sending your e-proof corrections. Now is the time to inform your Public Relations or Press Office about your paper, as they might be interested in promoting its publication. This will allow them time to prepare an accurate and satisfactory press release. Include your manuscript tracking number (NG-A60452R1) and the name of the journal, which they will need when they contact our Press Office.

Before your paper is published online, we shall be distributing a press release to news organizations worldwide, which may very well include details of your work. We are happy for your institution or funding agency to prepare its own press release, but it must mention the embargo date and Nature

Genetics. Our Press Office may contact you closer to the time of publication, but if you or your Press Office have any enquiries in the meantime, please contact press@nature.com.

Please note that *Nature Genetics* is a Transformative Journal (TJ). Authors may publish their research with us through the traditional subscription access route or make their paper immediately open access through payment of an article-processing charge (APC). Authors will not be required to make a final decision about access to their article until it has been accepted. [Find out more about Transformative Journals](https://www.springernature.com/gp/open-research/transformative-journals)

Authors may need to take specific actions to achieve [compliance](https://www.springernature.com/gp/open-research/funding/policy-compliance-faqs) with funder and institutional open access mandates. If your research is supported by a funder that requires immediate open access (e.g. according to [Plan S principles](https://www.springernature.com/gp/open-research/plan-s-compliance)) then you should select the gold OA route, and we will direct you to the compliant route where possible. For authors selecting the subscription publication route, the journal's standard licensing terms will need to be accepted, including [self-archiving-and-license-to-publish](https://www.nature.com/nature-portfolio/editorial-policies/self-archiving-and-license-to-publish). Those licensing terms will supersede any other terms that the author or any third party may assert apply to any version of the manuscript.

An online order form for reprints of your paper is available at <https://www.nature.com/reprints/author-reprints.html>. Please let your coauthors and your institutions' public affairs office know that they are also welcome to order reprints by this

method.

If you have not already done so, we invite you to upload the step-by-step protocols used in this manuscript to the Protocols Exchange, part of our on-line web resource, natureprotocols.com. If you complete the upload by the time you receive your manuscript proofs, we can insert links in your article that lead directly to the protocol details. Your protocol will be made freely available upon publication of your paper. By participating in natureprotocols.com, you are enabling researchers to more readily reproduce or adapt the methodology you use. [Natureprotocols.com](https://natureprotocols.com) is fully searchable, providing your protocols and paper with increased utility and visibility. Please submit your protocol to <https://protocolexchange.researchsquare.com/>. After entering your nature.com username and password you will need to enter your manuscript number (NG-A60452R1). Further information can be found at <https://www.nature.com/nature-portfolio/editorial-policies/reporting-standards#protocols>

Thank you.

Sincerely,
Chiara

Chiara Anania, PhD
Associate Editor
Nature Genetics
<https://orcid.org/0000-0003-1549-4157>